# CDK4/6 inhibition triggers ICAM1-driven immune response and sensitizes *LKB1* mutant lung cancer to immunotherapy

Xue Bai [1,8], Ze-Qin Guo [1,8], Yan-Pei Zhang [1,2,8], Zhen-zhen Fan[3,4,8], Li-Juan Liu[5,8], Li Liu[2,6], Li-Li Long[1], Si-Cong Ma [1,2], Jian Wang[1], Yuan Fang [1], Xin-Ran Tang[1], Yu-Jie Zeng [7], Xinghua Pan [5] ✉, De-Hua Wu [1] ✉ & Zhong-Yi Dong [1] ✉

Liver kinase B1 (*LKB1*) mutation is prevalent and a driver of resistance to immune checkpoint blockade (ICB) therapy for lung adenocarcinoma. Here leveraging single cell RNA sequencing data, we demonstrate that trafficking and adhesion process of activated T cells are defected in genetically engineered *Kras*-driven mouse model with *Lkb1* conditional knockout. *LKB1* mutant cancer cells result in marked suppression of intercellular adhesion molecule-1 (ICAM1). Ectopic expression of *Icam1* in *Lkb1*-deficient tumor increases homing and activation of adoptively transferred SIINFEKL-specific CD8+ T cells, reactivates tumor-effector cell interactions and re-sensitises tumors to ICB. Further discovery proves that CDK4/6 inhibitors upregulate *ICAM1* transcription by inhibiting phosphorylation of retinoblastoma protein RB in *LKB1* deficient cancer cells. Finally, a tailored combination strategy using CDK4/6 inhibitors and anti-PD-1 antibodies promotes ICAM1-triggered immune response in multiple *Lkb1*-deficient murine models. Our findings renovate that ICAM1 on tumor cells orchestrates anti-tumor immune response, especially for adaptive immunity.

Developments in immune checkpoint inhibitors (ICIs) are revolutionizing treatment options for non-small cell lung cancer (NSCLC)[1–3]. Although ICIs represent a set of powerful new tools, their clinical benefits have mostly been observed in patients with lung cancer without driver mutations[3–5]. Notably, the status of driver genes is closely associated with the immune profile of the tumor microenvironment (TME) in NSCLC as well as the response to immunotherapy, especially in lung adenocarcinoma (LUAD)[1,6–8]. Patients with epidermal growth factor receptor (*EGFR*) mutations or *EML4-ALK* fusion gene tend to have a diminished response to immunotherapy[9,10]. Additionally, liver kinase B1 (*LKB1*) mutations in LUAD lead to primary resistance to ICIs, regardless of the presence of *KRAS* co-mutations[11]. Therefore, tackling

[1]Department of Radiation Oncology, Nanfang Hospital, Southern Medical University, 1838 North Guangzhou Avenue, Guangzhou 510515, China. [2]Information Management and Big Data Center, Nanfang Hospital, Southern Medical University, Guangzhou 510515, China. [3]CAS Key Laboratory of Genomics and Precision Medicine, Beijing Institute of Genomics, University of Chinese Academy of Sciences, Chinese Academy of Sciences, China National Center for Bioinformation, Beijing 100101, China. [4]Key Laboratory of Functional Protein Research of Guangdong Higher Education Institutes and MOE Key Laboratory of Tumor Molecular Biology, Institute of Life and Health Engineering, College of Life Science and Technology, Jinan University, Guangzhou, China. [5]Department of Biochemistry and Molecular Biology, School of Basic Medical Sciences, Southern Medical University, and Guangdong Provincial Key Laboratory of Single Cell Technology and Application, Guangzhou 510515 Guangdong Province, China. [6]Department of Medical Quality Management, Nanfang Hospital, Southern Medical University, Guangzhou 510515, China. [7]The First Clinical Medical College, Southern Medical University, 1838 North Guangzhou Avenue, Guangzhou 510515, China. [8]These authors contributed equally: Xue Bai, Ze-Qin Guo, Yan-Pei Zhang, Zhen-zhen Fan, Li-Juan Liu. ✉e-mail: PanVictor@SMU.edu.cn; 18602062748@163.com; dongzy1317@foxmail.com

immunotherapy resistance from the perspective of driver mutations holds great promise to uncover the mechanisms linking cancer genetics to immunotherapy responsiveness.

*LKB1* is the third most frequently mutated gene in LUAD[12]. Patients with lung cancer harboring an oncogenic *LKB1* mutation, are refractory to almost all currently available therapies[13,14]. Recently, *LKB1* mutations in LUAD have attracted substantial attention as a potential driver of primary ICI resistance[15], although mutation of this gene increases tumor mutational burden[16]. Genetic alterations of *LKB1* define a subtype of LUAD, characterized by a T-cell-excluded TME and low programmed death-ligand 1 (PD-L1) expression on tumor cells (type II cancers)[6,17]. However, few studies have depicted it at single cell resolution, while bulk cell sequencing hides the cellular heterogeneity and likely misses the specific subset of cells and cell interactions that may be vital. We therefore leveraged single cell RNA sequencing to establish the atlas for *Kras/Lkb1* driven lung tumors in CRISPR/Cas9-edited genetically engineered mouse models (GEMM). These unbiased profiles offer insights into cell–cell interactions through ligand-receptor signaling network as well as the precise cell clusters and their molecular scenarios, which might uncover the molecular underpinnings that drive the immunotherapy resistance in *LKB1* deficient tumor ecosystems. In such cases, combination immunotherapy that targets the vulnerable point, switching immunologically "cold" tumors to "hot" tumors, was effective for these patients[18].

A pilot analysis showed that *LKB1* alterations were correlated with inferior clinical outcomes in patients with NSCLC who received combination therapy of chemotherapy and pembrolizumab[19]. Loss of *LKB1* leads to the suppression of stimulator of interferon genes (STING), suggesting that therapies that reactivate LKB1 or the STING pathway may reinvigorate the anticancer immune response and reverse the resistance to ICI therapy[20]. Therefore, establishing personalized and targeted combination approaches for *LKB1* mutant lung cancer are warranted. Accumulating data suggest that targeted inhibition of certain oncogenic pathways in combination with programmed cell death protein-1 (PD-1) axis blockade promoted reactivation of the immune microenvironment, which might be more suitable for immune-cold tumors[21]. Therefore, identifying an evidence-driven targeted combination immunotherapy may offer a breakthrough to reverse the immunotherapy resistance of type II tumors.

In this work, we obtain mechanistic insights into how LKB1 sculpts the tumor immune microenvironment and influences immune response at single cell resolution. Moreover, we propose a combination strategy to overcome the resistant state, hopefully paving the way for clinical trials and guiding clinical practice for LUAD.

## Results

### Activated T-cell trafficking and adhesion are impaired in mutant *Kras/Lkb1* driven lung cancer

To obtain a comprehensive view of the immune atlas for *LKB1* deficient lung cancer, we utilized CRISPR/Cas9-mediated gene knockout in the genetically engineered *Kras*-driven mouse model to generate $Kras^{G12D/+}$ mice with conditional knockout of *Lkb1*. These animals were treated with nasal inhalation of lentiviruses targeting *Lkb1* (KL) or *Tomato* (K, referred to as negative control) (Fig. 1a). The tumors forming at twelve weeks after lentiviruses infection were comparable and confirmed by micro-CT (Fig. 1b), and conditional deletion of *Lkb1* in the lung tumors was verified (Fig. 1c, Supplementary Fig. 1A–C). The tumors were then analyzed by single cell RNA sequencing (scRNA-seq) (10X Genomics) to profile the cell atlas for *Kras/Lkb1* versus *Kras*-driven lung tumor. After quality control and cell filtering, we cataloged 14,260 cells into eleven distinct cell lineages annotated with canonical feature cell markers, thus identifying cancer cell types 1, cancer cell types 2, endothelial cells, and immune cells types (including activated T cells, exhausted T cells, B cells, neutrophils, macrophages cells, NK cells, dendritic cells and plasmacytoid dendritic cells) (Fig. 1d–f,

Supplementary Data 1). Cell composition analysis showed reduced activated T cells and increased exhausted T cells in tumors from KL mice compared with those from K mice (Fig. 1f, Supplementary Fig. 1D). Activated T cells in KL group were characterized by significantly lower expression of *Tcf7* relative to K group ($p = 0.001$, Fig. 1g, Supplementary Fig. 1E); while exhausted T cells displayed significantly higher expression of *Ctla4* in KL mice versus those of K mice ($p < 0.0001$, Fig. 1g, Supplementary Fig. 1E). To further evaluate the general effect of *LKB1* deficiency on different immune cell clusters, we also calculated gene signature-based scores at the single cell level in different cell types using single sample gene set enrichment analysis (ssGSEA)[22]. Cancer cells in KL group showed significantly higher LKB1 loss signature scores[23], confirming *Lkb1* deficiency in KL mice (Supplementary Fig. 1F). And we found that activated T cells isolated from KL tumors showed significantly lower "T cells homing on tumor" signature scores; while exhausted T cells isolated from KL tumors showed significantly higher "PD-1 pathway", "T-cell anergy" signature scores compared with T cells from K tumors (Supplementary Fig. 2A, B). However, no significant difference in signature scores was observed among the other cell clusters, including B cells (Supplementary Fig. 2C–F). These data suggest that T-cell activation status is the main determinant downstream signaling of LKB1 in dictating the tumor immune microenvironment.

Besides, we also investigated the immune profiles of LUAD patients from The Cancer Genome Atlas (TCGA) database. Tumors with loss of or mutated *LKB1* had significantly lower infiltration of immune-competent cells (CD4, CD8-effector, and NK cells), tumor-associated macrophages (TAMs), and T regulatory cells than tumors harboring WT *LKB1* (Supplementary Fig. 3). Notably, there was no significant difference in the infiltration pattern among tumors with *LKB1* mutations and those with *LKB1* loss (Supplementary Fig. 3), suggesting that such effects on immunocyte infiltration were equivalent. These data demonstrated that *LKB1* deficiency disturbed the immune infiltration landscape regardless of the specific mutation type.

Next, we focused our attention on $Tcf7^+$ activated T-cell clusters. Pathway enrichment analysis among these activated T cells indicated that T-cell adhesion and immune effector process significantly downregulated in KL versus K group (Fig. 1h, Supplementary Fig. 1E). We deduced that the underlying reason for paucity of activated T cells in *LKB1*-deficient tumors might lie in their impaired trafficking and adhesion process.

### LKB1 modulates intercellular adhesion molecule-1 (ICAM1) expression in non-small cell lung cancer

To elucidate the chief culprit leading to impaired trafficking and adhesion of T cells in *LKB1* deficient lung cancer, we exploited inch-by-inch search among different cell types and compared their differential biological functions between KL and K samples using scRNA-seq data (Supplementary Fig. 4A–F). Enrichment analysis targeting cancer cell clusters demonstrated that cancer cell adhesion process was also defected in KL groups (Fig. 2a), which aroused our attention to cancer cells themselves and cancer-T-cell interactions.

In the cancer cell dimension, we firstly established lung cancer cell lines expressing WT (*LKB1*-WT), mutated (kinase-dead mutation; *LKB1*-MUT), or no LKB1 (*LKB1* control) (A549 and H460 cell lines, both of which exhibit the complete loss of *LKB1*) and performed RNA sequencing. We also assessed the transcriptomes of human LUAD samples (GSE72094). A series of differentially expressed genes after *LKB1* deficiency in both cells and tissues were identified. Subsequent overrepresentation analysis based on these genes revealed the top 20 immune-specific signatures associated with *LKB1* alteration (Fig. 2b), which could be summarized into two main categories, (1) cytotoxic cell activation and immune response, and (2) cell adhesion and leukocyte trafficking, which was consistent with the results of our scRNA data and pathway enrichment analysis (Supplementary Fig. 5A–E). Genes among

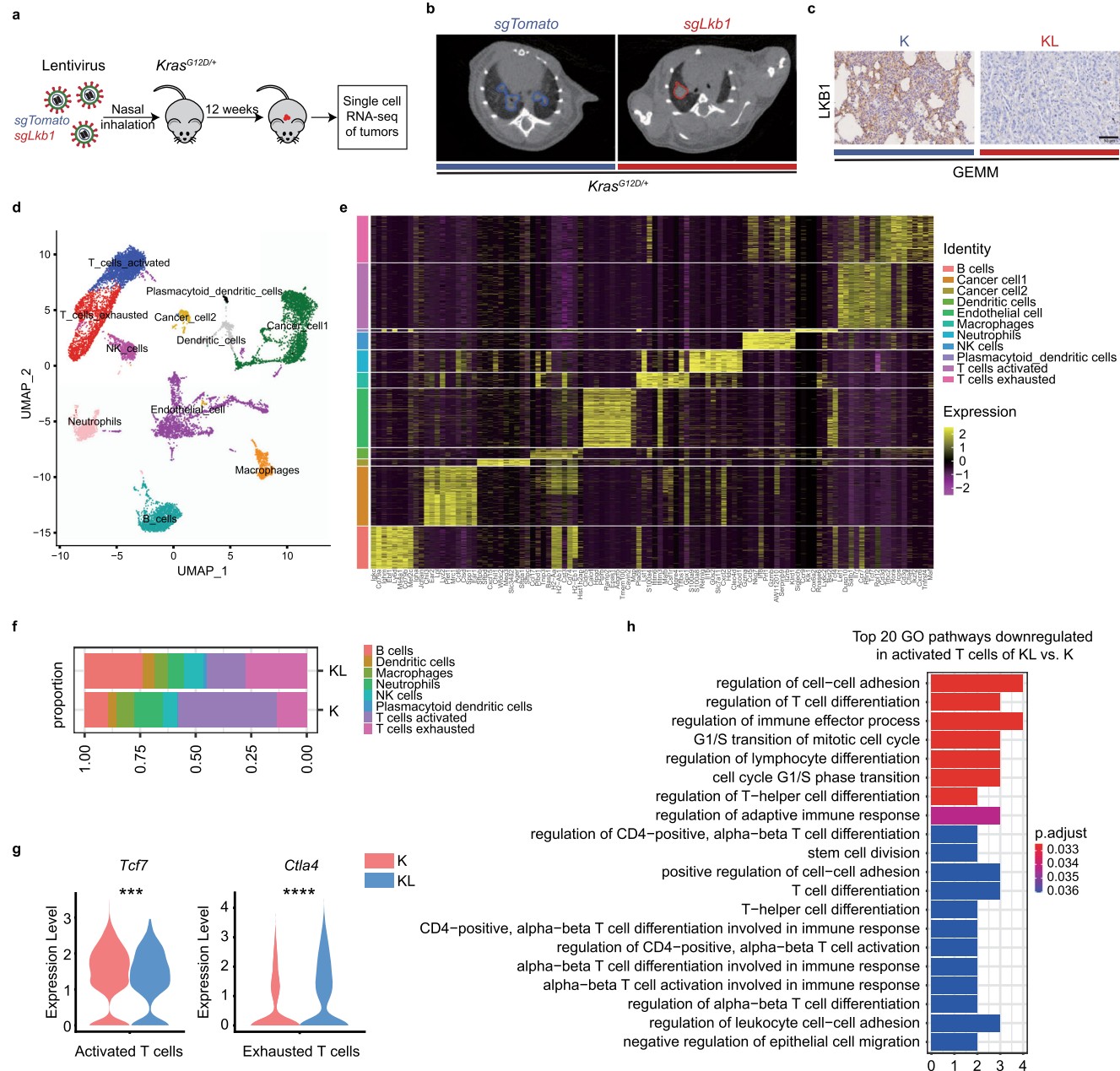

**Fig. 1 | Mutant *Kras/Lkb1* driven lung cancer cell atlas was established.**
**a** Schematic illustration for generation of the genetically engineered *Kras*-driven mouse model to establish *Kras*^G12D/+ mice with conditional knockout of *Lkb1*.
**b** Micro-CT scan of the mouse lung. Left: *Kras*^G12D/+ sg*Tomato* (K) mouse. Right: *Kras*^G12D/+ sg*Lkb1* (KL) mouse. **c** IHC staining of LKB1 in mouse lung tumor sections in K and KL mouse. Scale bar, 50 μm (40×). *n* = 1 experiment with *n* = 5 mice. **d** UMAP plot of 14,260 cells from lung tumor samples of K (*n* = 1) and KL (*n* = 1) mouse, colored by their 11 major cell types. **e** Heatmap of canonical cell-type markers of 11 major cell types. **f** Immune cell-type composition of each sample. **g** Expression level of *Tcf7* in activated T cells and that of *Ctla4* in exhausted T cells from lung tumor samples of K and KL mouse. ****p* = 0.001, *****p* < 0.0001. **h** Enrichment analysis on differentially expressed genes (DEGs) in KL versus K along the activated T cells using clusterProfiler for gene sets in GO terms. (Data were presented as violin plot (median, 25–75%, range). Statistical significance was tested with a two-tailed Mann–Whitney *U* test. ns, not significant; ****p* < 0.001; *****p* < 0.0001. Source data are provided as a Source Data file.).

these enriched signatures were assessed and ranked according to their frequency. *ICAM1* emerged as the most frequently differentially expressed gene (Fig. 2b), indicating that ICAM1 in cancer cells might play a critical role in regulating cell–cell adhesion with T cells.

In order to investigate into the interplay between cancer and T cells, we analyzed receptor–ligand-interactions using Cellphone DB. Cell–cell interaction analysis for K and KL demonstrated a salient interaction between cancer cells and endothelial cells (Supplementary Fig. 6A, B). Besides, we found fewer interaction possibilities in KL than K group, between cancer cells and T cells (Supplementary Fig. 6C). Particularly, cancer cells in KL versus K showed impaired interaction

with T cells relating to ICAM1-mediated T-cell adhesion and recruitment (Supplementary Fig. 6D). This provoked us to seek regulatory mechanism of ICAM1 in *LKB1* deficient cancer cells.

Gene profiles of patients with LUAD from TCGA were analyzed. The tumors with mutated *LKB1* displayed lower levels of *ICAM1* than those with WT *LKB1* (Fig. 2c), regardless of *KRAS* co-mutation and *LKB1* mutation types (Supplementary Fig. 7A, B). We then assessed the association between LKB1 and ICAM1 in clinical samples by immunohistochemistry (IHC). Decreased ICAM1 was observed in more than 80% of *LKB1*-mutated tumor samples examined, whereas 80% of *LKB1*-WT specimens showed intensified ICAM1 staining (Fig. 2d). Together,

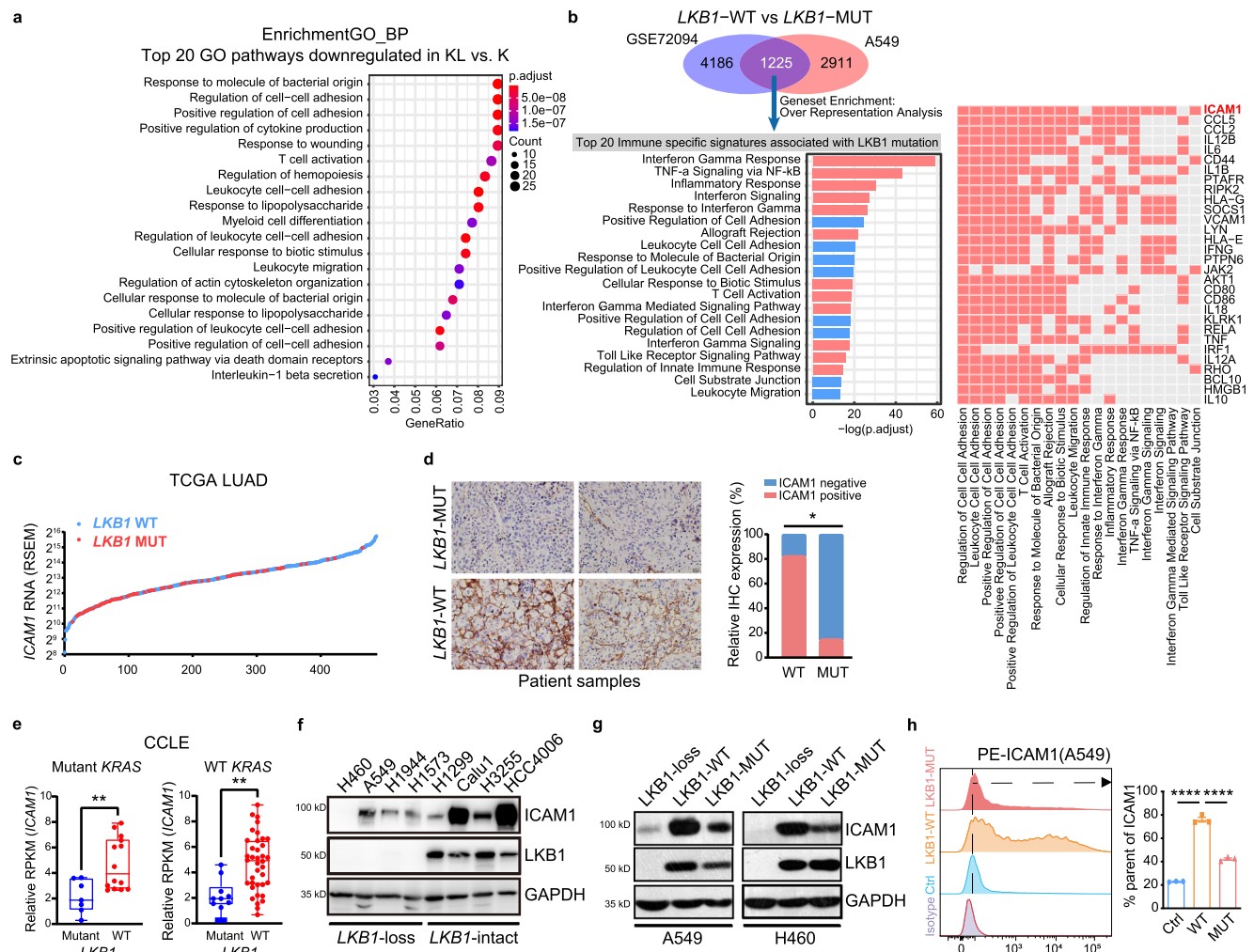

**Fig. 2 | LKB1 regulates ICAM1 expression in non-small cell lung cancer.**
**a** Enrichment analysis on DEGs in the cancer cells between KL (*n* = 701 cells) versus K (*n* = 662 cells) using clusterProfiler for gene sets in GO terms. **b** Venn diagram showing the overlap of DEGs between *LKB1* wild-type and mutated groups both in tissues (GSE72094, *n* = 374 for *LKB1*-WT, *n* = 68 for *LKB1*-MUT) and cells (A549 cells, *n* = 3 for *LKB1*-WT, *n* = 3 for *LKB1*-MUT). Overrepresentation analysis was performed using these overlap genes, and the top 20 immune-specific signatures were revealed. Ranking of the genes among these enriched signatures according to their frequency. **c** Plot displays *ICAM1* expression level in lung adenocarcinoma patient samples with (*n* = 73) and without (*n* = 437) *LKB1* mutation. **d** Representative images and corresponding quantification of IHC staining of ICAM1 in human LUAD tumor sections in *LKB1* wild-type and mutated groups. Scale bar, 50 μm (40×). *n* = 11 (WT) or *n* = 7 (MUT) biologically independent samples examined over 1 independent experiment. **p = 0.0128. **e** Relative RPKM values of *ICAM1* in *LKB1* mutant versus *LKB1* wild-type lung cancer cells from CCLE, with or without *KRAS* co-mutations.

Mutant *KRAS*: *n* = 7 with mutant LKB1, *n* = 14 with WT LKB1. WT *KRAS*: *n* = 10 with mutant LKB1, *n* = 38 with WT LKB1. Middle line: median; box edges: 25th and 75th percentiles; whiskers: the upper and lower ends of the whiskers signify the maxima and minima, respectively. Most extreme points that do not exceed ± interquartile range × 1.5; further outliers are marked individually. **p = 0.0066, **p = 0.0011. **f** Immunoblot (IB) of the indicated proteins in lung cancer cells with *LKB1*-loss or *LKB1*-intact. *n* = 3 experiments. **g**, **h** A549 and H460 lung cancer cells were transfected with lentivirus expressing the indicated genes (Ctrl, *LKB1*-WT, and *LKB1*-Mut). ICAM1 expression was analyzed by immunoblot (**g**, *n* = 3 experiments.) and flow cytometry (**h**; left, representative images; right, quantification of results. *n* = 3 biologically independent samples examined over 1 independent experiment. ****p < 0.0001, ****p < 0.0001.) (Results are presented as mean ± SEM. One-way ANOVA followed by Tukey's multiple comparisons test, two-tailed Student's t-test or two-tailed Fisher's exact test was used to analyze the data. **p < 0.01; ***p < 0.001; ****p < 0.0001. Source data are provided as a Source Data file.).

these data confirmed that *LKB1* deficiency is associated with suppression of ICAM1 in lung cancer. To demonstrate this finding further, we examined ICAM1 expression across a panel of *LKB1* mutant versus *LKB1* wild-type lung cancer cell lines. ICAM1 mRNA and protein levels were either significantly downregulated or completely undetectable in *LKB1* mutant cell lines, regardless of *KRAS* co-mutation (Fig. 2e, f).

We subsequently investigated the causal relationship between LKB1 and ICAM1. Small interference RNA or short hairpin RNA mediated knocking down of *LKB1* decreased ICAM1 expression in vitro and in vivo (Supplementary Fig. 7C, D). Finally, *LKB1* reconstitution in ICAM1[lo] KL cell lines rescued ICAM1 expression, which also required LKB1 kinase activity (Fig. 2g, h and Supplementary Fig. 7E–G). Therefore, LKB1 positively regulates ICAM1 expression.

## ICAM1 may be prerequisite for the homing and activation of tumor-specific T cells in *LKB1* deficient lung cancer

To fully demonstrate that ICAM1-mediated cancer cell-T-cell adhesion and interactions affects trafficking and activation of effector immune cells in *LKB1*-deficient NSCLC, we adoptively transferred congenitally-marked (APC[+]) stimulated CD8[+] T cells expressing the OT-I, which is specific for OVA, into mice bearing *Lkb1* proficient/deficient orthotopic lung tumors (LLC1-control-*Icam1*[WT]-OVA; LLC1-sh*Lkb1-Icam1*[WT]-OVA; LLC1-sh*Lkb1-Icam1*[OE]-OVA) and assessed OT-I cell infiltration and activation status in lung tumors (Fig. 3a). On day 5 after T cells injection, we observed a sharp decrease of OT-I cells in *Lkb1*-deficient tumors than that in *Lkb1* proficient tumors, indicating the role of LKB1 in influencing cytotoxic T-lymphocyte (CTL) infiltration; furthermore,

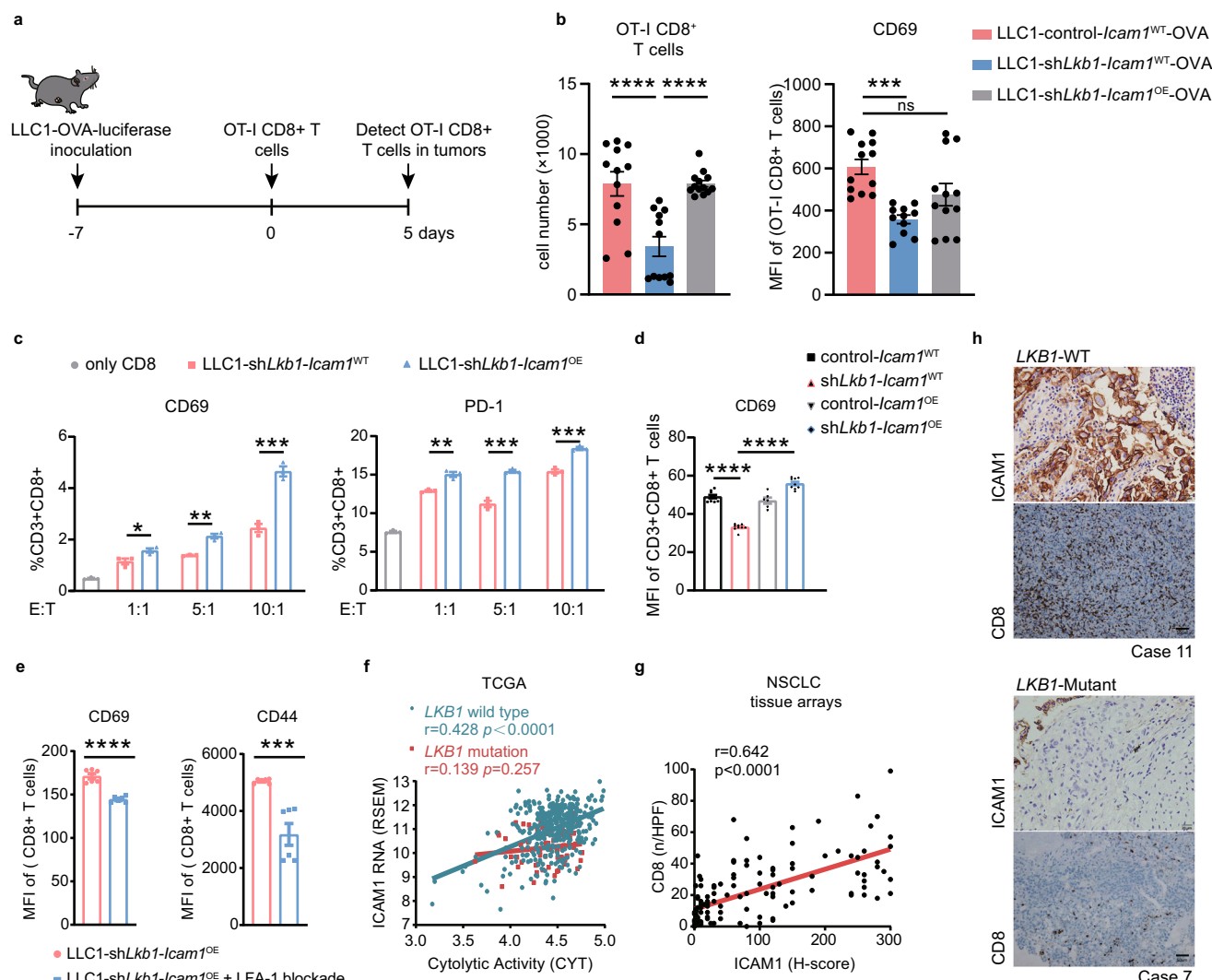

**Fig. 3 | ICAM1 is important for the homing and activation of tumor-specific T cells in *LKB1* deficient lung cancer. a** LLC1-OVA tumor cells (LLC1-control-*Icam1*WT-OVA; LLC1-sh*Lkb1*-*Icam1*WT-OVA; LLC1-sh*Lkb1*-*Icam1*OE-OVA) were implanted. Congenitally-marked (APC+) and stimulated CD8+ T cells expressing the OT-I were transferred on day 7 and analyzed on day 12. **b** Infiltration of OT-I CD8+ T cells (left) and expression of CD69 on OT-I cells (right) were analyzed by flow cytometry. For CD8+ T cells, $n = 12$, $n = 12$ and $n = 13$ in control-*Icam1*WT, sh*Lkb1*-*Icam1*WT and sh*Lkb1*-*Icam1*OE groups, respectively. ****$p < 0.0001$, ****$p < 0.0001$. For CD69, $n = 12$, $n = 11$ and $n = 12$ in control-*Icam1*WT, sh*Lkb1*-*Icam1*WT and sh*Lkb1*-*Icam1*OE groups, respectively. ***$p = 0.0003$. **c** Flow cytometry analysis of CD69 and PD-1 expression on activated T cells with or without co-culturing with LLC1 tumor cells (sh*Lkb1*-*Icam1*WT, sh*Lkb1*-*Icam1*OE) for 24 h. The *E:T* ratio (effector to target) was 1:1, 5:1, and 10:1 respectively. $n = 3$ biologically independent samples examined over 1 independent experiment. CD69, *$p = 0.038$, **$p = 0.0012$, ***$p = 0.0009$; PD-1, **$p = 0.0036$, ***$p = 0.0007$, ***$p = 0.0009$. **d** Flow cytometry analysis of CD69 expression on activated T cells co-culturing with LLC1 tumor cells (control-*Icam1*WT, $n = 9$; sh*Lkb1*-*Icam1*WT, $n = 9$; control-*Icam1*OE, $n = 6$; sh*Lkb1*-*Icam1*OE, $n = 9$) for 24 h.

The *E:T* ratio (effector to target) was 10:1. ****$p < 0.0001$, ****$p < 0.0001$. **e** Flow cytometry analysis of CD69 and CD44 expression on activated T cells with co-culturing of LLC1 tumor cells (LLC1-sh*Lkb1*-*Icam1*OE), with or without LFA-1 antibody blockade therapy for 24 h. The *E:T* ratio (effector to target) was 10:1. $n = 6$ biologically independent samples examined over 1 independent experiment. ****$p < 0.0001$, ***$p = 0.0006$. **f** Correlation analysis of *ICAM1* expression and cytolytic activity (CYT) patient samples (TCGA) with or without *LKB1* mutations. *LKB1* wild-type, $n = 374$ samples, $p < 0.0001$; *LKB1* mutation, $n = 68$ samples, $p = 0.257$. **g** Correlation analysis of IHC staining with anti-CD8a and anti-ICAM1 antibodies on a tissue microarray (TMA) of lung adenocarcinoma patients ($n = 152$ TMA elements). $p < 0.0001$. **h** IHC of ICAM1 and CD8 expression in *LKB1*-WT and *LKB1*-mutant lung cancer patients. Scale bar, 20 μm (40×), 50 μm (20×). $n = 11$ (*LKB1*-WT) or $n = 7$ (*LKB1*-mutant) biologically independent samples examined over 1 independent experiment. (Results were presented as mean ± SEM. One-way ANOVA followed by Tukey's multiple comparisons test or two-tailed Student's *t*-test was used to analyze the data. ns, not significant; *$p < 0.05$; **$p < 0.01$; ***$p < 0.001$; ****$p < 0.0001$. Source data are provided as a Source Data file.).

the number significantly increased in LLC1-sh*Lkb1*-*Icam1*OE-OVA tumors than LLC1-sh*Lkb1*-*Icam1*WT-OVA groups (Fig. 3b). Besides, ICAM1 overexpression significantly improved the activated state of OT-I CTLs (CD69) in *Lkb1*-deficient tumors (Fig. 3b). Taken together, these data suggest that ICAM1 may be prerequisite for the homing of activated tumor-specific T cells in *LKB1*-deficient tumors. We also verified this phenomenon from co-culture experiments, in which overexpression of ICAM1 on cancer cells enhanced CD69 and PD-1 expression on CD8+ T cells after their direct contact with tumor cells for 24 h (Fig. 3c, d). ICAM1-mediated adhesion on cancer cells is a

prerequisite for their interaction with T cells through LFA-1[24]. To determine the receptor that interacts with ICAM1 on cancer cells, antibody blocking experiments were performed. Pre-treatment of CD8+ T cells with anti-CD11a antibodies (LFA-1 blockade), reduced CD69 and CD44 expressions on T cells after co-culture experiments (Fig. 3e), indicating that ICAM1 on cancer cells mediates the interaction with T cells by binding to LFA-1.

Other co-culturing experiments demonstrating the mechanism by which LKB1 induces an antitumor effect through ICAM1 were performed (Supplementary Fig. 8A–C). KL parental cells (*KRAS*-mutant,

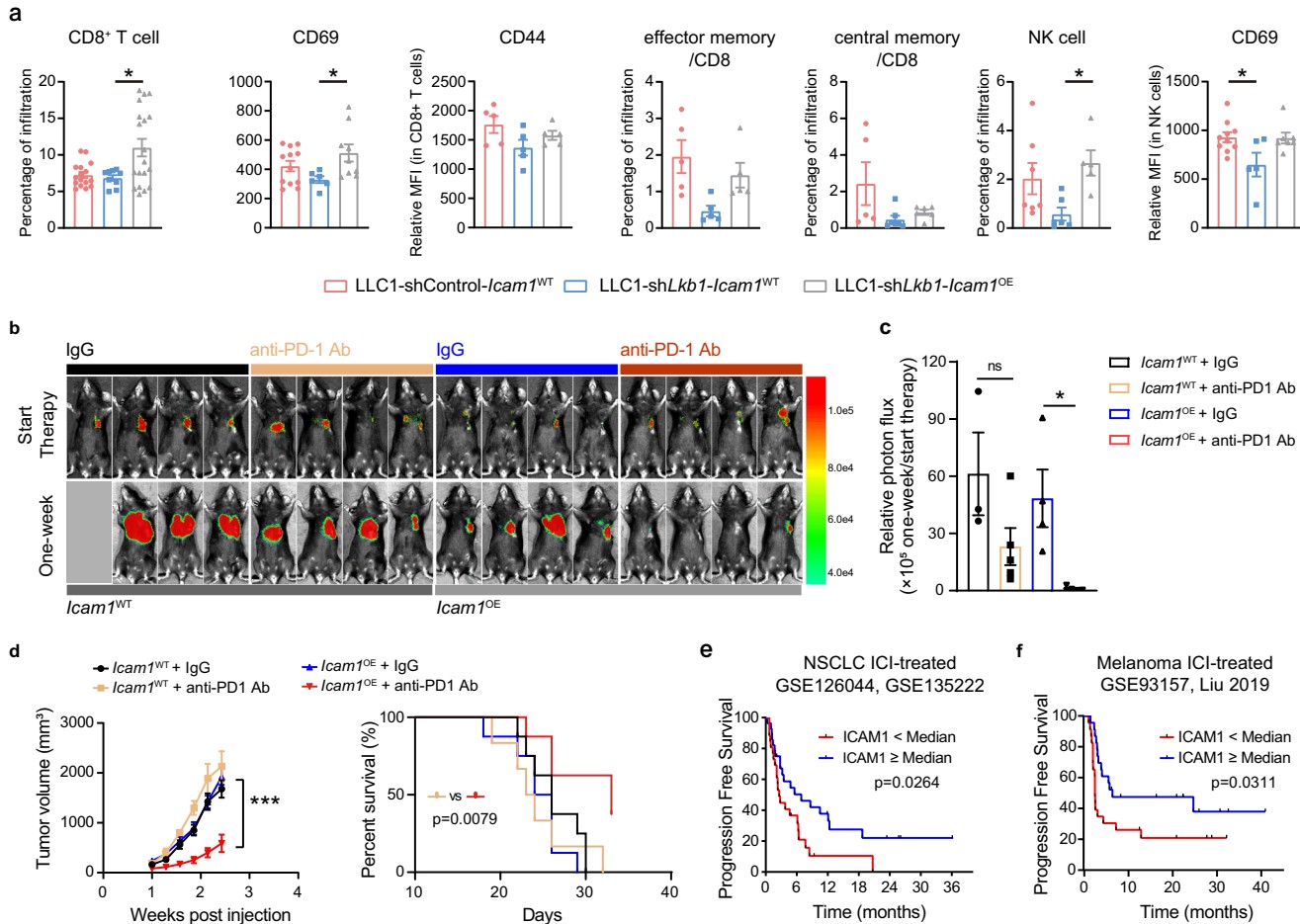

**Fig. 4 | ICAM1 overexpression reactivated the TME and sensitized anti-PD-1 immunotherapy in _Lkb1_ deficient lung tumors. a** Total CD8[+] T cells relative to CD45[+] cells in tumor tissue, relative mean fluorescence intensity (MFI) of CD69 and CD44 on CD8[+] T cells, percentage of CD62L[+]CD44[+] (central memory T cells) and CD62L[−]CD44[+] (effector memory T cells) in CD8[+] T cells, and total NK cells relative to CD45[+] cells in tumor tissue and relative MFI of CD69 on NK cells from mice bearing LLC1 tumors (LLC1-shControl-_Icam1_[WT], LLC1-sh_Lkb1_-_Icam1_[WT], LLC1-sh_Lkb1_-_Icam1_[OE]) were analyzed by flow cytometry. _n_ = 1 independent experiment. CD8[+] T-cell, _n_ = 13 in shControl-_Icam1_[WT], _n_ = 8 in sh_Lkb1_-_Icam1_[WT], _n_ = 16 in sh_Lkb1_-_Icam1_[OE]; *_p_ = 0.0193. CD69, _n_ = 12 in shControl-_Icam1_[WT], _n_ = 6 in sh_Lkb1_-_Icam1_[WT], _n_ = 9 in sh_Lkb1_-_Icam1_[OE]; *_p_ = 0.0436. CD44, _n_ = 5 samples in each group. Effector memory, _n_ = 5 samples in each group. Central memory, _n_ = 5 in shControl-_Icam1_[WT], _n_ = 7 in sh_Lkb1_-_Icam1_[WT], _n_ = 6 in sh_Lkb1_-_Icam1_[OE]. NK cells, _n_ = 7 in shControl-_Icam1_[WT], _n_ = 6 in sh_Lkb1_-_Icam1_[WT], _n_ = 5 in sh_Lkb1_-_Icam1_[OE]; *_p_ = 0.0415. CD69, _n_ = 10 in shControl-_Icam1_[WT], _n_ = 5 in sh_Lkb1_-_Icam1_[WT], _n_ = 7 in sh_Lkb1_-_Icam1_[OE]; *_p_ = 0.0333. **b** LLC1-sh_Lkb1_-_Icam1_[WT] or LLC1-sh_Lkb1_-_Icam1_[OE] luc cells were injected into the left chest of mice and tumor formation was detected using a bioluminescence imager every week. Bioluminescent images in mice bearing lung tumors treated with isotype and anti-PD-1.

**c** Quantification of bioluminescence results. _n_ = 3 in _Icam1_[WT] receiving IgG, _n_ = 5 in _Icam1_[WT] receiving anti-PD-1 Ab, _n_ = 4 in _Icam1_[OE] receiving IgG, _n_ = 5 in _Icam1_[OE] receiving anti-PD-1 Ab. *_p_ = 0.0477. **d** C57BL/6 mice were subcutaneously inoculated with Lewis lung cancer cells with _Lkb1_ knockdown and stably _Icam1_ overexpression (_Icam1_[OE]) or its parental (_Icam1_[WT]) cell line. They were administered with different treatments (control immunoglobulin G (IgG), or anti-PD-1 Ab). Tumor size (left, _n_ = 6 in _Icam1_[WT] receiving IgG, _n_ = 5 in _Icam1_[WT] receiving anti-PD-1 Ab, _n_ = 9 in _Icam1_[OE] receiving IgG, and _n_ = 8 in _Icam1_[OE] receiving anti-PD-1 Ab. ***_p_ = 0.0006.) and survival (right, _n_ = 8 in _Icam1_[WT] receiving IgG, _n_ = 6 in _Icam1_[WT] receiving anti-PD-1 Ab, _n_ = 8 in _Icam1_[OE] receiving IgG, and _n_ = 8 in _Icam1_[OE] receiving anti-PD-1 Ab. _p_ = 0.0079) in different treatment arms were monitored. Kaplan–Meier curves of progression-free survival according to the expression level of _ICAM1_ in lung adenocarcinoma patients (**e** GSE126044 and GSE135222, _n_ = 56 in total.) and melanoma patients (**f** GSE93157 and Liu et al.[26], _n_ = 46 in total.) following immune checkpoint inhibitor therapy. (Results are presented as mean ± SEM. Mixed-effects model followed by Tukey's multiple comparison test or log-rank test was used to analyze the data. ns, not significant; *_p_ < 0.05; ***_p_ < 0.001. Source data are provided as a Source Data file.).

_LKB1_-loss; A549 and H460; LV-Ctrl) or LKB1-overexpressing cells (LV-_LKB1_-WT and LV-_LKB1_-mutant) were co-cultured with human NK-92 cells or activated human PBMCs. LKB1-overexpressing cells exhibited higher levels of apoptosis than _LKB1_-deficient cells after co-culturing with activated T or NK cells. Overexpression of _ICAM1_ significantly enhanced tumor cell apoptosis, while knockdown of _ICAM1_ inhibited NK-mediated apoptosis (Supplementary Fig. 8D–J), which indicated that LKB1-induced differences in response to cytotoxic immune cells were dependent on ICAM1. T-cell migration and cell adhesion assay further proved that LKB1 facilitated CD8[+] T-cell adhesion and migration in vitro through ICAM1 (Supplementary Figs. 9, 10 and 11A, B).

We next analyzed the RNA-sequencing data from TCGA based on the cytolytic activity (CYT), a quantitative index calculated from

transcript levels of two key cytolytic effectors: granzyme A and perforin[25]. A positive correlation between ICAM1 and CYT was only observed in patients with WT _LKB1_ (Fig. 3f). The differences in CYT between _LKB1_ WT and mutated samples thus corroborated our co-culture results.

Subsequently, we sought to address whether ICAM1 correlates with augmented CD8[+] T cells in lung cancer patients' samples. IHC of human NSCLC tissue arrays showed that the number of CD8-positive cells was positively correlated with ICAM1 expression on tumor cells (Pearson correlation: _r_ = 0.642, _p_ < 0.0001). Additionally, CD8[+] T cells were often distributed in the ICAM1-positive regions of the tumors (Fig. 3g, h, Supplementary Fig. 11C, D). IHC of paired samples from NSCLC patients furtherly verified that _LKB1_ mutant patients displayed

lower ICAM1 expression and fewer CD8[+] T cells infiltration (Fig. 3h, Supplementary Fig. 12A, B).

## ICAM1 overexpression shifts the tumor immune profiles and sensitizes *Lkb1* deficient lung tumors to anti-PD-1 immunotherapy

We firstly investigated the functional effects of ICAM1 reinforcement on the TME in the LLC1-sh*Lkb1* orthotopic lung tumor system. ICAM1 overexpression significantly increased CD8[+]-cell and NK cell infiltration (Fig. 4a). Besides, expression level of CD69 (T-cell activation) on CD8[+] T cells was significantly increased, and CD44 expression exerted an enhancing tendency. For the differentiation status of CD8[+] T cells (effector memory CD8[+] T cells, and central memory CD8[+] T cells), no significant changes were observed, though an enhancing tendency was displayed in effector memory CD8[+] T cells when ICAM1 was augmented (Fig. 4a). Taken together, *LKB1* deficiency leads to rarity of immune cytotoxic cells, and ICAM1 can enhance infiltration and activation of CD8[+] T cells and NK cells.

We next questioned whether ICAM1 overexpression might impact the efficacy of ICI therapy. A luciferase plasmid was stably transduced into LLC1-sh*Lkb1-Icam1*[OE] and its parental LLC1-sh*Lkb1-Icam1*[WT] cell line, and these cells were then implanted in situ. Overexpression of *Icam1* sensitized *Lkb1*-deficient tumors to anti-PD-1 immunotherapy as better tumor control (Fig. 4b, c). Besides, *Icam1* overexpression-initiated response to immunotherapy was also observed in the subcutaneous tumor model (Fig. 4d). However, cancer cell-intrinsic knockout of *Icam1* reduces tumors responsiveness to immunotherapy (Supplementary Fig. 13A–D). Taken together, these data illustrated that ICAM1 overexpression sensitized *LKB1* deficient lung tumors to anti-PD-1 immunotherapy.

Finally, we assessed the correlation between the ICAM1 expression and the survival of ICIs-treated cancer patients. A significant progression-free survival (PFS) benefit was observed in patients with higher expression of *ICAM1* both in NSCLC cohorts (n = 43, with RNA sequencing data: GSE126044, n = 16; GSE135222, n = 27, Samsung Medical Center; Fig. 4e) and two melanoma cohorts (GSE93157, n = 25; Liu et al.[26], n = 21; Fig. 4f) (Gide et al.[27], n = 41; Supplementary Fig. 14A), although *ICAM1* itself was not a good prognosis factor (Supplementary Fig. 14B, C). These results indicated that ICAM1 may be exploited as a potential predictor for immunotherapy. We furtherly demonstrated that *ICAM1* is an independent predictive biomarker in the NSCLC cohort using multivariate COX regression analysis (*ICAM1*: HR, 0.412; p = 0.013; *PD-L1*: HR, 0.405; p = 0.021; respectively; Supplementary Fig. 14D), while this was not observed in the melanoma cohort (*ICAM1*: HR, 0.693; p = 0.465; *PD-L1*: HR, 0.205; p = 0.001; Supplementary Fig. 14E).

## CDK4/6 inhibitors reverse the ICAM1-defected, immune-resistant state of *LKB1* mutant lung cancer

Considering the key role of ICAM1 in response to ICIs, we sought to find a clinically accessible approach to restore the expression of ICAM1 in *LKB1* deficient lung cancer. A combined drug screening strategy was adopted. We first selected 94 drugs from the Genomics of Drug Sensitivity in Cancer (GDSC) database based on sensitivity to *LKB1* mutation. Next an FDA-approved immunology compound library was taken into consideration, and 5 candidate drugs overlapped between them (Fig. 5a). These drugs were evaluated whether they can upregulate expression of ICAM1. Results revealed that palbociclib increased expression of ICAM1 remarkably in both A549 and H460 cells at both transcriptional and protein levels (Fig. 5b–d). To rule out potential off-target effects of palbociclib, we used another validated CDK4/6 inhibitor, ribociclib, to repeat the experiments. Results showed that ribociclib could effectively increase the expression of *ICAM1* in both A549 and H460 cells (Supplementary Fig. 15A), corroborating the efficacy of CDK4/6 inhibitor. We furtherly assessed whether palbociclib could

ameliorate the immune-resistant state. An immune-resistant program, which was derived from a large-scale single cell sequencing analysis in human melanoma, was employed to rate the ability of cancer cells to induce immune resistance[28]. The program consisted of two parts of genes whose upregulation or downregulation promoted immune resistance, and a score was calculated to represent the total resistant level. Three cell lines with *LKB1* loss (A549, H460, H2122) were treated with or without palbociclib, and their transcriptomic sequencing data (GSE110397) were then analyzed. The heatmap verified that palbociclib reversed immune resistance program (Fig. 5e), and the immune resistance scores were reduced after palbociclib induction (Fig. 5f).

Palbociclib, a clinically approved CDK4/6 inhibitor that was demonstrated to reactivate retinoblastoma protein (RB) by maintaining an unphosphorylated state[29], was thus selected for further analysis. As expected, palbociclib exerted an inhibitory effect on RB phosphorylation (Fig. 5g). Further, it also restored downregulated p-p65 level in *LKB1* deficient cells (Supplementary Fig. 15B). However, upregulation of p-p65-ICAM1 expression was abolished when *RB* was knocked down (Fig. 5g, Supplementary Fig. 15B–D), indicating that these were "on-target" effects.

To further test how ICAM1 is regulated by LKB1, a control vector or a Flag-*LKB1*-overexpressing vector was introduced into H1299 cells, and immunoprecipitation with an anti-Flag antibody was performed, followed by mass spectrometry (MS) analysis. RB constituted as one of the immunoprecipitated candidates based on the MS analysis (Supplementary Fig. 15E). Furthermore, phosphorylated RB can be specifically targeted by CDK4/6 inhibitors. LKB1 bound with RB, which was validated in H1299 cells (Fig. 5h), and the immunofluorescent staining assay displayed that LKB1 and RB were mainly colocalized in the cytoplasm in lung cancer cells (Supplementary Fig. 15F). Moreover, cyclin dependent kinase 4 (CDK4), a known interactor of RB[30], was also pulled down by LKB1, although to a lesser extent than that by RB, further corroborating the LKB1–RB-CDK4 interaction (Fig. 5h). We next sought to specify the effect of LKB1 on RB expression. *LKB1* deficiency, either via complete loss (+Vector) or a kinase-dead mutation (+*LKB1*-MUT), had little impact on total RB and CDK4 levels. However, the level of phosphorylated RB (pRB) was increased in cells with *LKB1* deficiency (Fig. 5i, Supplementary Fig. 15G). Analysis of the reverse-phase protein arrays (RPPA) data of TCGA-LUAD cohort also demonstrated that *LKB1* mutation was correlated with higher phosphorylated level of RB protein (Supplementary Fig. 15H).

Since CDK4 phosphorylates RB[31], LKB1 suppressed the phosphorylation of RB by affecting CDK4, we therefore speculated that LKB1 competed with CDK4 for RB binding, which led to reduced RB phosphorylation. It was reported that CDK4/Cyclin D specifically docked in the C-terminal helix motif of RB[30,32]. Bioinformatic analysis predicted the possibility of LKB1 docking on this site of RB, which was statistically significant (p < 0.05), demonstrating that LKB1 possibly had the same binding site with CDK4, indicating the potential for docking competition (Supplementary Fig. 15I). Subsequent IP experiments supported this hypothesis; specifically, LKB1 overexpression interfered with the interaction of CDK4 with RB, and vice versa (Supplementary Fig. 15J). However, CDK4-RB interaction recovered in *LKB1*-MUT group (Supplementary Fig. 15K). Inspired by this finding, we further constructed a plasmid to generate an RB protein variant with truncation of the RB C terminus (RB[ΔC-term]) to cover the possible binding site comprehensively. We utilized this RB variant (ΔC-term) to investigate whether LKB1 competitively docks on the C domain of RB with CDK4. Immunoprecipitation assay of LKB1 and CDK4 with RB variant were both decreased (Supplementary Fig. 15L). Moreover, we constructed phosphor-mimetic RB (13E mutant)[33,34]. Co-IP analysis demonstrated that phosphor-mimetic RB display no significant differences in interacting with LKB1 (Supplementary Fig. 15M).

We next investigated the influence of LKB1 on pRB and ICAM1 in vivo. IHC staining of mouse samples demonstrated that knockdown

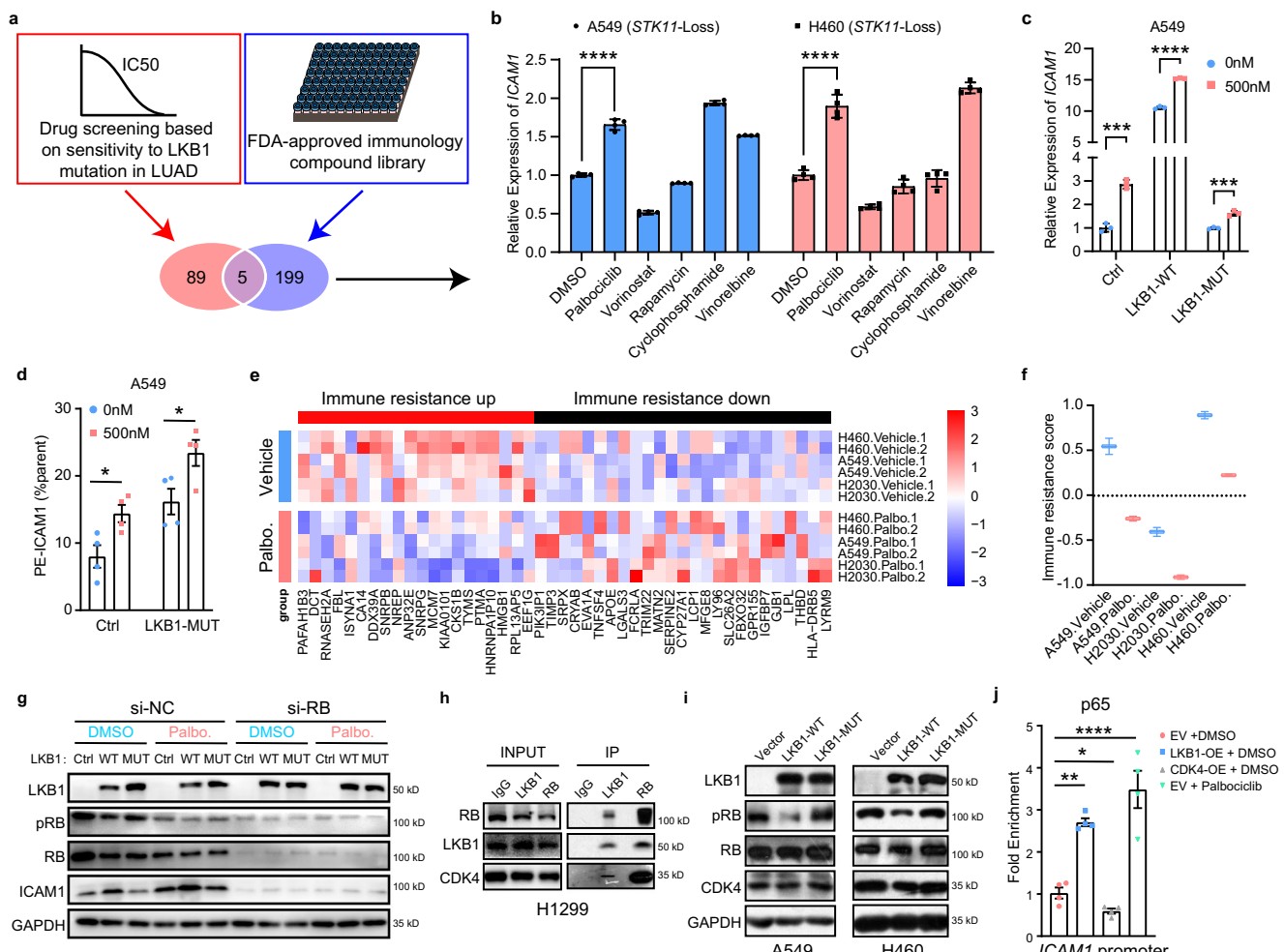

**Fig. 5 | CDK4/6 inhibitors were carefully selected for *LKB1* mutant lung cancer based on the mechanistic link between LKB1-RB-CDK4 interaction. a** Combined drug screening using drugs from both the Genomics of Drug Sensitivity in Cancer (GDSC) database based on sensitivity to *LKB1* mutation and an FDA-approved immunology compound library. **b** Quantitative RT-qPCR of relative *ICAM1* expression in A549 and H460 cells treated with 5 selected drugs (cyclophosphamide monohydrate; vinorelbine tartrate; vorinostat; rapamycin; palbociclib). $n = 4$ biologically independent samples examined over 1 independent experiment. ****$p < 0.0001$, ****$p < 0.0001$. A549 cells transfected with lentivirus expressing the indicated genes (LV-Ctrl, LV-*LKB1*-WT, LV-*LKB1*-Mut), treated ± 500 nM palbociclib were harvested for quantitative RT-qPCR (**c**) and flow cytometry analysis (**d**). $n = 3$ biologically independent samples examined over 1 independent experiment. ***$p = 0.0002$, ****$p < 0.0001$, ***$p = 0.007$ (**c**). $n = 4$ biologically independent samples examined over 1 independent experiment. *$p = 0.023$, *$p = 0.0378$ (**d**). **e**, **f** Impact of CDK4/6i on *LKB1* mutant lung cell line profiles. **e** Expression of immune resistance program genes (columns) that were most differentially expressed in palbociclib-treated (green) versus control (pink) cell lines (rows). Expression is normalized in each cell line. **f** Immune resistance scores in cell lines (A549, H460, and H2030) treated with palbociclib ("Pal") or with DMSO vehicle ("con") (GSE110397). Middle line: median; box edges: 25th and 75th percentiles. $n = 2$ biologically independent samples. **g** A549 cells transfected with lentivirus expressing the indicated genes (LV-Ctrl, LV-*LKB1*-WT, LV-*LKB1*-Mut), treated ± 500 nM palbociclib, transfected ± si*RB* were harvested for immunoblot. $n = 3$ independent experiments. **h** Immunoprecipitation analysis of the interaction among LKB1, RB, and CDK4 performed in H1299 cells expressing intact LKB1. $n = 3$ independent experiments. **i** Immunoblot analysis of the indicated proteins performed in A549 or H460 cells expressing the indicated genes in plasmids. $n = 3$ independent experiments. **j** ChIP assay was performed with cell lysates from A549/vector, A549/LKB1-OE, A549/CDK4-OE and A549/vector-palbociclib cells. A pair of primers flanking the p65 binding site within the *ICAM1* promoter were used in PCR. Real-time PCR was employed to the ChIP assay. $n = 4$ biologically independent samples examined over 1 independent experiment. **$p = 0.0015$, *$p = 0.0249$, ****$p < 0.0001$. (Results are presented as mean ± SEM. One-way ANOVA followed by Tukey's multiple comparisons test or two-tailed Student's *t*-test was used to analyze the data. *$p < 0.05$; **$p < 0.01$; ***$p < 0.001$. Source data are provided as a Source Data file.).

of *Lkb1* resulted in increased pRB and reduced ICAM1 expression levels (Supplementary Fig. 15N–O). Furthermore, IHC of human NSCLC tissue arrays also showed a negative correlation between ICAM1 and pRB (Supplementary Fig. 15P, Q).

p65 is the key transcription factor for *ICAM1*, and we also provided evidence in our analysis (Supplementary Fig. 16A–F). *LKB1* deficiency led to a decreased phosphorylation level of p65 (Supplementary Fig. 15G). Chromatin Immunoprecipitation (ChIP) analysis demonstrated that p65 occupies the promoter of *ICAM1* (Supplementary Fig. 16G). Analysis of the *ICAM1* promoter displayed a predicted p65 binding site (Supplementary Fig. 16D). To ascertain whether the

binding sequences in *ICAM1* gene promoters regulate transcription, we incorporated the binding sites into a luciferase reporter system. The results indicated that the activity of the promoter-driven luciferase was increased when p65 was overexpressed. Moreover, the mutation of the p65 binding site in the *ICAM1* promoter significantly reduced the p65-driven expression of luciferase (Supplementary Fig. 16H). Additionally, p65 occupancy in the promoter of *ICAM1* is increased in *LKB1*-WT group compared to *LKB1* loss group (Fig. 5j), which furtherly demonstrated that LKB1 regulates the transcriptional activity of p65 for the expression of *ICAM1*. In agreement with this, p65 knockdown impaired ICAM1 expression in LKB1 restored group (Supplementary Fig. 16I, J).

Furthermore, p65 occupancy was significantly reduced in CDK4-OE group compared to that of the vector group ($p = 0.0249$) (Fig. 5j). CDK4/6 inhibition decreased RB phosphorylation (the same as that of LKB1 overexpression) (Supplementary Fig. 16K), thus enhanced p65 occupancy versus that of the vector group ($p < 0.0001$) (Fig. 5j).

CDK4-phosphorylated RB specifically binds to p65 and inhibits its transcriptional activity[31]. We treated cells with palbociclib to determine whether blocking CDK4 would interfere the binding between LKB1 and RB. Results showed that palbociclib unambiguously enhanced such binding (Supplementary Fig. 16L). And LKB1 reconstitution interfered RB-p65 interactions (Supplementary Fig. 16M–O). RB-p65 interaction depends on CDK4/6 S249/T252 phosphorylation of RB[31]. We therefore constructed RB S249/T252 phosphorylation resistant mutant (S249A/T252A) plasmids. The following co-IP assay demonstrated that RB mutant (S249A/T252A) plasmids markedly disturbed RB-p65 interaction (Supplementary Fig. 16P). When *LKB1* was overexpressed or CDK4/6 inhibitors were used, RB-p65 interaction was observably decreased even when wild-type RB was overexpressed (Supplementary Fig. 16P). *CDK4* overexpression prominently enhanced RB-p65 interaction in RB wild-type group, but no such effect was observed for RB S249A/T252A group (Supplementary Fig. 16P). Together, LKB1 blocks CDK4-phosphorylated RB (S249/T252 phosphorylation), p65 is thus released and transactivates *ICAM1*.

The dephosphorylation of RB by CDK4/6 inhibitors rescued *ICAM1* transcription and thus might enhance the activation of cytotoxic cells. To segregate the effect of CDK4/6 inhibition on tumor cells versus T cells, we performed ex-vivo co-culturing experiments. Tumor cells were pretreated with CDK4/6 inhibitors or PBS. CDK4/6 inhibition on tumor cells reinforced their interactions with T cells (Supplementary Fig. 17A); CDK4/6 inhibitors alone did not increase tumor cell apoptosis (Supplementary Fig. 17B), while co-culturing of CDK4/6 inhibitors pretreated tumor cells with NK-92 cells significantly increased apoptosis rate of tumor cells (Supplementary Fig. 17C), implying the potential value of CDK4/6 inhibitors in activating the immune microenvironment.

## CDK4/6 inhibition potentiates ICI efficacy of *LKB1* deficient lung tumors in vivo

In light of our in vitro findings, we subsequently investigated whether CDK4/6 inhibitors exhibit a synergistic effect with ICI therapy in vivo. Since *LKB1* deficiency directly led to immunotherapy resistance (Supplementary Fig. 18A)[15], we thus implanted LLC1-sh*Lkb1* cells subcutaneously into C57BL/6 mice, and administered different treatments: vehicle, palbociclib, anti-PD-1 antibody, or their combination (Supplementary Fig. 18B). Importantly, combination of palbociclib and anti-PD-1 antibody significantly alleviated the tumor burden and prolonged survival compared with monotherapy or vehicle (Fig. 6a–c). The degree of selectivity of the palbociclib/anti-PD-1 approach for *LKB1*-deficient NSCLC was also assessed. Compared to *Lkb1* wild-type or overexpressed tumors (Supplementary Fig. 18C, D), *Lkb1* deficient NSCLC is uniquely sensitive to the synergistic effect of the combination therapy.

To further investigate the antitumor efficacy of co-targeting CDK4/6 and PD-1, we constructed a LLC1-sh*Lkb1*-luc cell line and established in situ tumor models, followed by different treatment arms. The results showed that co-inhibition of CDK4/6 and PD-1 displayed enhanced tumor control (Fig. 6d, e). Similar results were observed in our genetically engineered *Kras*-driven mouse model with conditional deletion of *Lkb1* (KL GEMM) (Fig. 6f). A significant tumor regression was observed in the combination arm (Fig. 6g).

## Combining CDK4/6 inhibition with PD-1 blockade synergistically induces a favorable immune microenvironment

To provide a comprehensive assessment of immunotherapeutic responses fueled by CDK4/6 inhibitor and anti-PD-1 treatment, RNA-sequencing of murine tumor specimens was performed and analyzed. Immunomodulators (IMs) are crucial for cancer immunotherapy, and dissecting their expression and modes helps in understanding different states of the TME. Results showed that palbociclib mono- or combination therapy promoted the expression of stimulatory IM and suppressed the inhibitory IM (Fig. 7a). In addition, higher levels of immune cell infiltration were observed in the combination therapy group through murine Microenvironment Cell Population counter (mMCPcounter) analysis (Fig. 7b, Supplementary Fig. 19A) than monotherapy. To depict the physical cell–cell interaction landscape using our bulk-RNA-seq data, we developed intercellular communication signatures. As described above in our scRNA-seq data analysis, we identified 11 cell types and their corresponding marker genes. For each cell type, we matched its marker gene-encoded surface proteins to their cognate ligands/receptors based on the assembled ligand-receptor pairs from the STRING website. Then a signature containing all the ligands/receptors proteins that can bind to its marker gene-encoded proteins was defined as the communication signature for this cell type. The communication signatures for these 11 cell types were summarized in Supplementary Data 2. Results implied that vehicle and anti-PD-1 monotherapy had less intercellular interactions while palbociclib mono- and combination therapy presented an active communication campaign (Fig. 7c).

We then examined the immune profiles of tumor samples in different treatment arms using flow cytometry. Results showed that the number and activity of CD8+ T cells were significantly higher with the combination treatment than with vehicle, as reflected by elevated interferon-γ and decreased PD-1 levels. Activation of the TME was also observed in the palbociclib arm (Fig. 7d). Similar results were obtained regarding NK cell infiltration. The combination of palbociclib and anti-PD-1 antibody increased the infiltration of NK cells and CD107a expression on NK cells (Fig. 7e). These data suggested that the combined inhibition of CDK4/6 and PD-1 exerted a synergistic therapeutic effect by increasing the infiltration of active cytotoxic cells, thereby reversing the immune-resistant TME. We also analyzed treatment-induced changes in myeloid cell subpopulations. CDK4/6 inhibition reduced abundance of MDSCs and TANs in tumor tissue, and the combined CDK4/6 inhibitors and PD-1 blockade therapy may impact dendritic cell infiltration and function in the process of antigen processing and presentation (Supplementary Fig. 19B, C). Moreover, the combination treatment led to a significantly increased expression level of ICAM1 on tumor cells, which was also observed in the palbociclib group (Fig. 7f), but palbociclib did not induce ICAM1 expression on immune cells (Supplementary Fig. 19D, E). Further, the expression of PD-L1 on tumor cells was significantly enhanced in the palbociclib group (Fig. 7f), indicating that CDK4/6 inhibitors re-sensitize tumors to anti-PD-1 treatment.

## Synergistic effect of combined CDK4/6-PD-1 targeting was demonstrated in vivo

To get an insight into the potential synergistic effect of CDK4/6 inhibition and PD-1 blockade in vivo, we performed IHC and Immunofluorescence (IF) staining in mouse tumor samples. Lower pRB level and higher ICAM1 staining intensity were observed in the co-treatment group than the vehicle or anti-PD-1 group. Notably, palbociclib mildly promoted CD8+ cell infiltration, and combined treatment led to an appreciable increase in the number of CD8+ cells (Fig. 8a, b, Supplementary Fig. 20A, B). *Kras*-driven mouse sample with conditional deletion of *Lkb1* (KL GEMM) receiving the combination treatment also had higher membrane expression of ICAM1 and more CD8+ cells through IF staining. It is worth noting that CD8+ T cells tend to aggregate in the ICAM1-positive tumor areas (Fig. 8c), which were also observed in mice harboring orthotopic lung tumors (Fig. 8c).

Subsequently, a proteome profiler mouse cytokine array kit was exploited for the parallel investigation of secretary protein spectrums

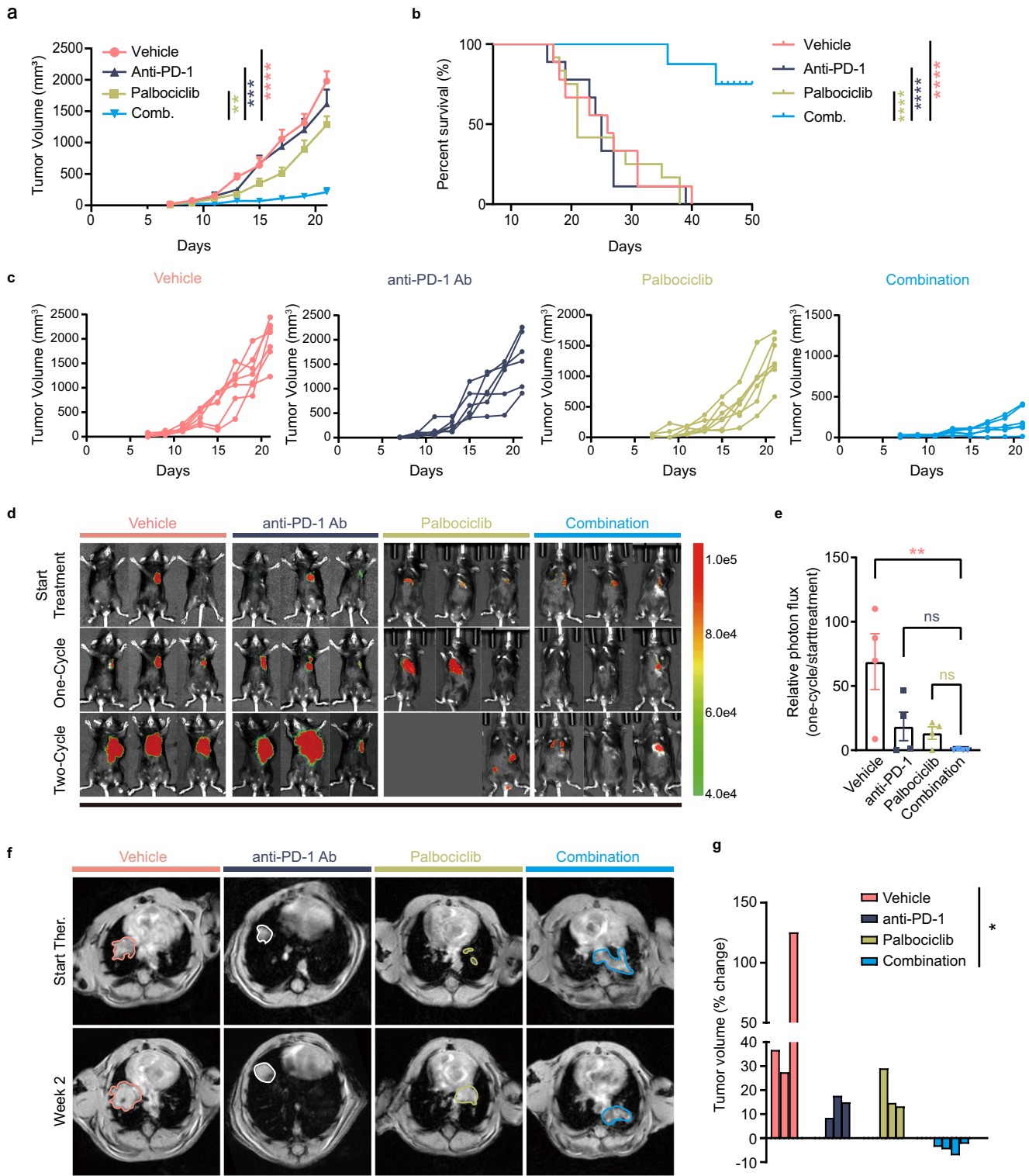

in orthotopic lung tumor lysates having received these four kinds of treatment. Interestingly, we found that secretary ICAM1 expression was also augmented in palbociclib group and combination group (Fig. 8d), which was consistent with findings from IHC and IF.

To further validate that the ICI-senitization effect exerted by CDK4/6 inhibitors is achieved through ICAM1 upregulation, which thereby influences the infiltration and activation of T cells and NK cells, ICAM1-, CD8-, and NK1.1-blocking antibodies were utilized for in vivo experiments. Depletion of ICAM1, CD8+ T cells, or NK cells significantly increased tumor growth (Fig. 8e) and shortened the survival time

(Fig. 8f). Subsequently, isogenic derivatives of the LLC1-sh*Lkb1* cell line with or without knockout of *Icam1* using CRISPR/Cas9 were established. The mice received the same combinational therapy. And we found that antitumor effects of the combined PD-1 blockade and CDK4/6 inhibition were significantly impaired in LLC1-sh*Lkb1*-sg*Icam1* tumors (Fig. 8g).

Accordingly, we systemically evaluated the effects of treatment with palbociclib or the combination therapy on the tumor micro-environment in the setting of ICAM1 depletion. IHC staining of tumor tissue displayed reduced ICAM1 expression and CD8+ T-cell infiltration

**Fig. 6 | CDK4/6i sensitizes LKB1-deficient tumors to anti-PD-1 therapy.**
**a**, **b** C57BL/6 mice were subcutaneously inoculated with Lewis lung cancer cells with *Lkb1* knockdown (LLC1-sh*Lkb1*) and administered different treatments (control immunoglobulin G (Vehicle), palbociclib or anti-PD-1 Ab, or co-treatment with palbociclib and anti-PD-1 Ab). Tumor size (**a**, *n* = 7 for vehicle, *n* = 7 for palbociclib, *n* = 6 for anti-PD-1 Ab, *n* = 6 for combination. Two-way ANOVA followed by Tukey's multiple comparisons test was performed to compare the tumor growth curves in different treatment groups, **\**p* = 0.0022, ***\**p* = 0.0003, ****\**p* < 0.0001) and survival (**b**, *n* = 9 for vehicle, *n* = 12 for palbociclib, *n* = 9 for anti-PD-1 Ab, *n* = 8 for combination. Log-rank test was used to analyze the survival data, ****\**p* < 0.0001, ****\**p* < 0.0001, ***\**p* = 0.0003) in different treatment arms were monitored. **c** Tumor volumes were measured beginning on day 7 and continuing every two days until day 21. **d**, **e** LLC1-sh*Lkb1*-luc cell line was constructed and injected into the left chest of mice. Tumor formation was detected. Mice harboring lung tumors were administered different treatments (control immunoglobulin G (Vehicle), palbociclib or anti-PD-1 Ab, or co-treatment with palbociclib and anti-PD-1 Ab). Representative

bioluminescent images (**d**) and quantification of results (**e**). *n* = 4 biologically independent mice examined over 1 independent experiment. One-way ANOVA followed by Tukey's multiple comparisons test was performed. **\**p* = 0.0097. **f**, **g** Genetically engineered *Kras*-driven mouse model with conditional deletion of *Lkb1* (KL GEMM) Representative MRI images (**f**) of KL GEMM lung tumors prior to treatment and after two weeks of treatment. *n* = 3, *n* = 3, *n* = 3 and *n* = 4 in vehicle, anti-PD-1 Ab, palbociclib and co-treatment groups, respectively. The contours of lung tumors were sketched. Waterfall plot (**g**) shows tumor volume response to the treatment. Each column represents one mouse. One-way ANOVA followed by Tukey's multiple comparisons test was performed. *\**p* = 0.0312. (Results are presented as mean ± SEM. A mixed-effects model followed by Tukey's multiple comparisons was performed to compare the tumor growth curves in different treatment groups. Log-rank test was used to analyze the survival data. ns, not significant; *\**p* < 0.05; **\**p* < 0.01; ***\**p* < 0.001; ****\**p* < 0.0001. Source data are provided as a Source Data file.).

---

in LLC1-sh*Lkb1*-sg*Icam1* tumors compared with LLC1-sh*Lkb1*-sgCon tumors after palbociclib treatment or combination therapy (Supplementary Fig. 21A). Flow cytometry analysis demonstrated that LLC1-sh*Lkb1*-sg*Icam1* tumors exhibited a significant decrease in CD8[+] T-cell infiltrate, and CD69, CD44 expression on CD8[+] T cells versus LLC1-sh*Lkb1*-sgCon tumors after CDK4/6 inhibition or combination therapy. Furthermore, infiltration of NK cells and DCs were significantly lower in LLC1-sh*Lkb1*-sg*Icam1* tumors after combination therapy versus that of LLC1-sh*Lkb1*-sgCon tumors. Taken together, these depletion therapies reversed the advantages conferred by the palbociclib/ICI combination therapy, confirming that CDK4/6 inhibition promotes the effect of anti-PD-1 immunotherapy in an ICAM1-dependent manner (Fig. 8h, Supplementary Fig. 21A, B).

Overall, based on the data, *LKB1* deficiency augments CDK4/6-induced phosphorylated RB, which abrogates ICAM1 expression. Impaired ICAM1-mediated cancer cell-T-cell adhesion and interaction might lead to paucity of T cells in the TME. Using CDK4/6 inhibitors to activate *ICAM1* transcription improved cytotoxic cell infiltration and activity, ameliorating the therapeutic efficacy of anti-PD-1 antibody and reversing immunotherapy resistance (Supplementary Fig. 22).

## Discussion

In real-world clinical practice, patients with NSCLC harboring *LKB1* mutations constitute a large proportion of patients with unmet therapeutic needs[17,35]. The emerging role of *LKB1* mutation as the major driver for resistance to anti-PD-1 therapies has attracted unprecedented research attention[19,20,36,37]. Thus, uncovering the underlying mechanism would help to develop novel therapeutic strategies. Here, through scRNAseq of mutant *Kras/Lkb1* driven lung tumor in genetically engineered mouse models, an unexplored role of mutant *LKB1* in dictating the TME via downregulating ICAM1 was discovered. We brought forward that ICAM1 on cancer cells orchestrates the antitumor immunity, which is prerequisite for the homing and activation of effector T cells for *LKB1* deficient lung cancer. Tailored to it, CDK4/6 inhibitors rescued the dilemma by upregulating ICAM1 and inducing an active immune microenvironment. We designed this study responding to the concerns noted in real-world practice.

From an unbiased profile through scRNA sequencing, we discovered that impairment of ICAM1-mediated cancer cell-T-cell adhesion and interactions plays a crucial role in the formation of T-cell deficient TME. *Icam1* overexpression enhanced the homing and activation of tumor-specific T cells in *Lkb1*-deficient lung cancer. The possible mechanism underlying ICAM1-dependent-T-cell homing on tumor is that the high secreted ICAM1 gradient from the cancer cell source attracts cytotoxic T cells to move toward tumor cells. Inside the tumor region, ICAM1 enhanced T-cell adhesion and interaction with tumor cells. Paradoxically, ICAM1 on tumor cells is previously associated with cancer metastasis and immune evasion[38]. ICAM1 on tumor

cells facilitates their adhesion to leukocytes, which subsequently bind to the endothelium, therefore supporting metastasis[39,40]. Besides, ICAM1-mediated T-cell gathering in tumor tissue, instead of the draining lymph node, was thought to be a trick that tumor escapes from the immune surveillance through overexpression of ICAM1[41]. However, cell adhesion process during cancer progression is complex and divergent. Opposite to their reports, we found that mutant tumor cells proactively reduced ICAM1 and avoided capturing T cells from the outside, which might be another pattern of immune evasion. In this scenario, activated T cells are excluded from the TME, and cancer cells with *LKB1* mutation keep their cellular fitness from immune attack. The evasion process proposed in our work might take place in the late phase of cancer-immunity cycle wherein T cells have been educated and activated in the lymph node; on the contrary, cancer cells arresting uneducated T cells through cell adhesion in the TME facilitating immune evasion might occur in the early stage[42,43]. In addition to that, a recent study reported that ICAM1 expression on the tumor cells dictates the abscopal effect of radiotherapy[44], reflecting the effect of ICAM1 on tumor cells in promoting T-cell infiltration and attack against tumors, which supported our work. We furtherly explained how *LKB1* mutant cancer cells escape immune attack by circumventing ICAM1-driven immune response.

Kitajima et al. reported that *LKB1* loss results in suppression of STING, facilitating escape of type I interferon and other STAT1-driven effector programs mediated immune response, raising important mechanistic insights into their resistance to PD-1/PD-L1 ICB[20]. This excellent study gave us lots of inspirations to discover more mechanisms behind this subtype of lung cancer. Mechanistically, *LKB1* deficiency leads to hyperphosphorylated RB, resulting in impaired p65 activation and *ICAM1* transcription. Furthermore, we believe that our theory to some extent confirmed with the work of Kitajima, et al. They reported that one possible mechanism for hyperactivation of DNMT1 is because of elevated S-adenosylmethionine levels. However, much remains to be understood regarding the mechanism by which DNMT1 is regulated. *LKB1* deficiency leads to hyperphosphorylation of RB and thereby activation of E2F1. *DNMT1*, a recognized E2F target gene, is thus upregulated and plays its role in silencing STING expression, which maybe one possible insightful link. Our work put forward another explanation why *LKB1* loss leads to *DNMT1* upregulation.

Furthermore, additional elements to the current *LKB1* mutant model were provided, *LKB1* deficiency leads to a disturbance in both innate and adaptive immunity. Mice treated with combination of CDK4/6 inhibitors and anti-PD-1 antibody exhibited fully transformed ICAM1-mediated immune activation in terms of NK cell-mediated non-specific immunity and T-cell-mediated targeted immunity. In this regard, we identified that T-cell-mediated specific immunity exerted a more prominent effect than NK cell-mediated non-specific immunity in the CDK4/6 inhibition/PD-1 blockade-induced antitumor immune

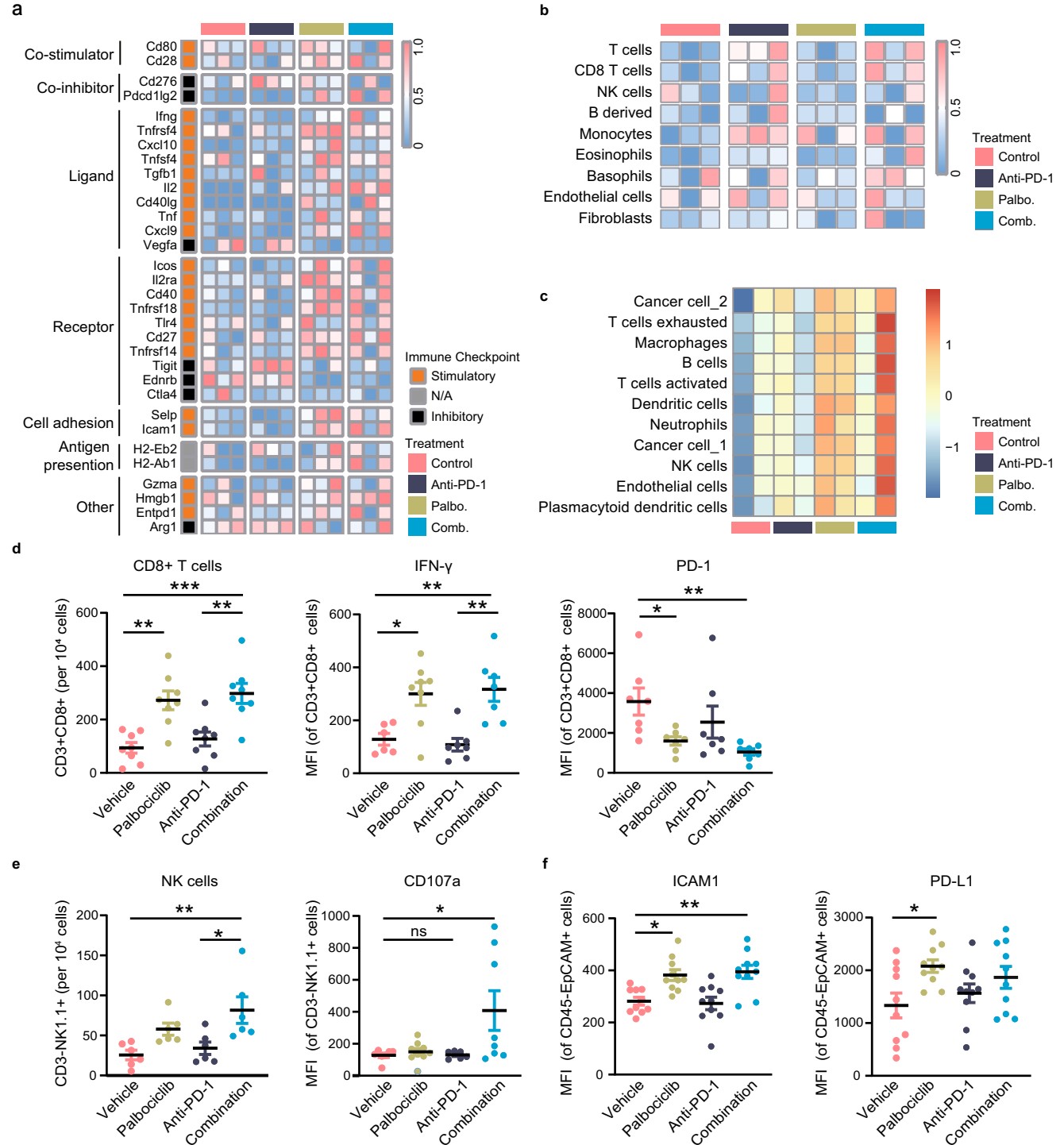

**Fig. 7 | Using CDK4/6 inhibitor and anti-PD-1 antibody triggers an active tumor immune microenvironment. a–c** RNA-sequencing of murine tumor specimens after receiving different treatments was performed and analyzed. **a** Heatmap of stimulatory Immunomodulators (IMs) and inhibitory IM in different treatment arms (*n* = 3 samples in each groups). **b** Murine Microenvironment Cell Population counter (mMCPcounter) analysis of immune cell infiltration level in different treatment arms (*n* = 3 samples in each groups). **c** Cell–cell interactions by calculating the scores of communication signatures in different cell types across different treatment arms (*n* = 2 samples in each groups). **d** Total CD8⁺ T cells relative to cells in tumor tissue (left), MFI of IFN-γ in CD8⁺ T cells (middle), and MFI of PD-1 in CD8⁺ T cells (right) from mice bearing LLC1-sh*Lkb1* tumors receiving the indicated treatments. For CD8⁺ T cells, *n* = 8 samples in each group; \*\**p* = 0.0016, \*\**p* = 0.0025, \*\*\**p* = 0.0003. For IFN-γ, *n* = 6 (vehicle), *n* = 8 (palbociclib), *n* = 7

(anti-PD-1 Ab), and *n* = 7 (combination), respectively; \**p* = 0.0165, \*\**p* = 0.0027, \*\**p* = 0.0099. For PD-1, *n* = 7 samples in each group; \**p* = 0.0422, \*\**p* = 0.0081. **e** Total NK cells relative to cells in tumor tissue (left, *n* = 7 samples in each group) and MFI of CD107a in NK cells (right, *n* = 7 in vehicle, *n* = 9 in palbociclib, *n* = 6 in anti-PD-1 Ab and *n* = 8 in combination group) from mice bearing LLC1-sh*Lkb1* tumors receiving the indicated treatments. NK cells, \**p* = 0.019, \*\**p* = 0.0052. CD107a, \**p* = 0.0351. **f** MFI of ICAM1 in tumor cells (middle, *n* = 10) and MFI of PD-L1 in tumor cells (right, *n* = 10) from mice bearing LLC1/sh*Lkb1* tumors receiving the indicated treatments. ICAM1, \**p* = 0.0119, \*\**p* = 0.0039. PD-L1, \**p* = 0.0399. (Results are presented as mean ± SEM. One-way ANOVA followed by Tukey's multiple comparisons test was used to analyze the data. ns, not significant; \**p* < 0.05; \*\**p* < 0.01; \*\*\**p* < 0.001; \*\*\*\**p* < 0.0001. Source data are provided as a Source Data file.).

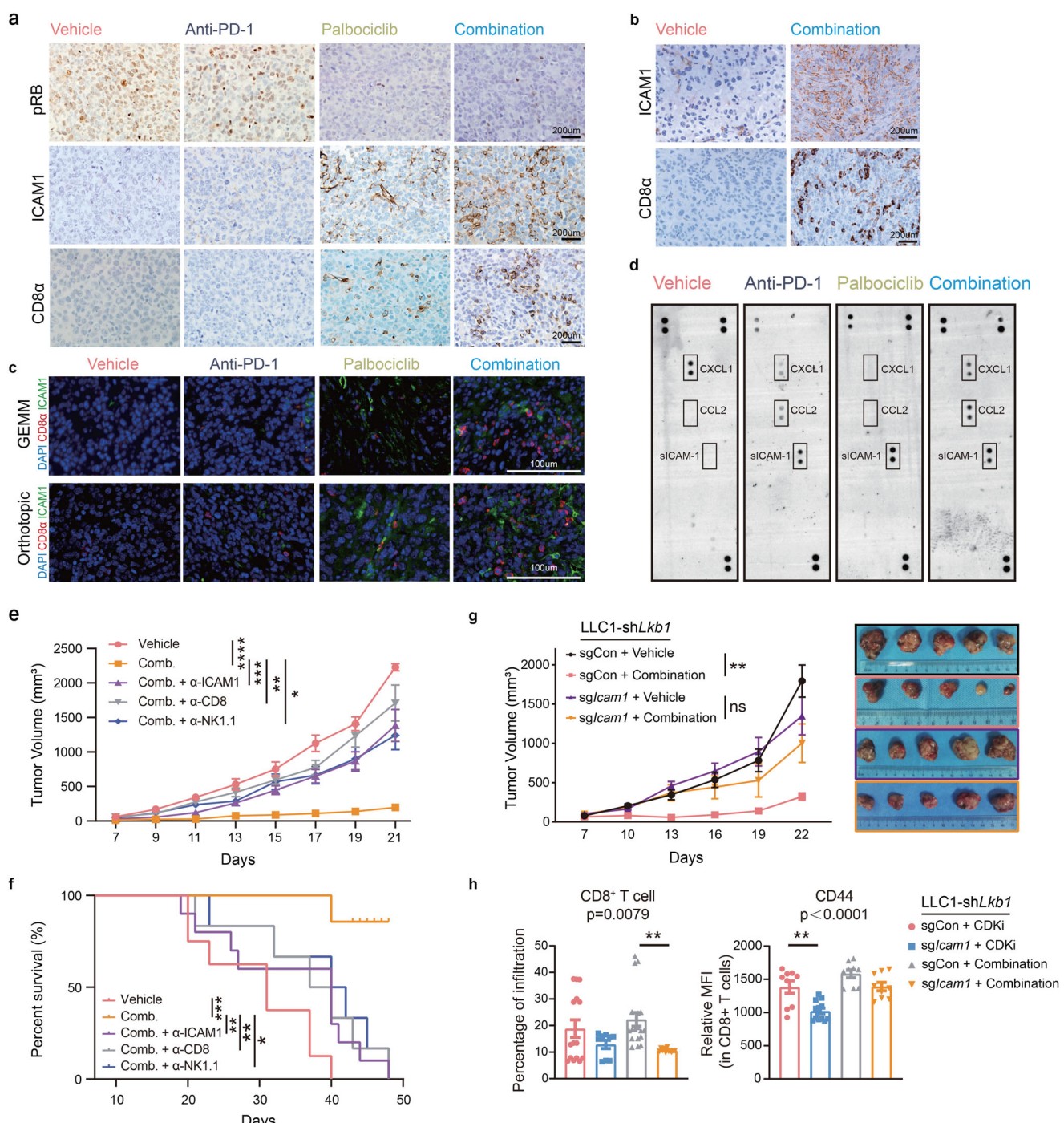

response, as the survival time of mice with CD8[+] T-cell depletion was shorter than that of mice with NK cell depletion. Studies using other models demonstrated that adaptive immune responses are activated by similar mono-targeted therapies[28,45,46]. Another study recently highlighted the activation of the NK cell-mediated innate immune attack in mice treated with both CDK4/6 inhibitors and MEK inhibitors, whereas CDK4/6 inhibitor monotherapy only mildly triggered an NK cell surveillance program that was not as strong as that achieved with the combination[47]. Complementary to these studies, our work illustrated the ICAM1-dependent activation of the T-cell response and supplementary activation of the NK cell response after treatment with CDK4/6 inhibitor and anti-PD-1 antibody. Our findings further highlighted the translational merits of cell cycle checkpoint inhibition in activating the adaptive and innate immune response, which makes cell

cycle checkpoint inhibition a promising partner with therapy targeting the PD-1 axis. This combination treatment is worth examining in other tumor models.

Our results provided a mechanistic understanding of the crosstalk between cell cycle checkpoint inhibition and immune response activation in the context of *LKB1*-mutant lung cancer. A prior GOrilla analysis was performed to identify key biological processes enriched in 3D and xenograft models compared to those in 2D models in *LKB1*-intact and -deficient cells. "Cell cycle" was observed as the top process expressed at lower levels, and "immune response" was observed as one of the top three processes at higher levels than the others[48]. In our study, we demonstrated that LKB1 regulates *ICAM1* transcription by inhibiting the phosphorylation level of RB, a key regulator of the cell cycle, and ICAM1 plays a pivotal role in immune activation. Our work

**Fig. 8 | In situ analyses and blocking treatment unveiled mechanism of synergistic effect of combined CDK4/6-PD-1 targeting therapy. a** IHC staining using anti-pRB, anti-ICAM1, and anti-CD8a antibodies on resected mouse samples following different arms of treatment in subcutaneous tumor models. $n = 5$ biologically independent mice examined over 1 independent experiment. **b** IHC staining using anti-ICAM1, and anti-CD8a antibodies on resected mouse samples following vehicle and combination treatment in GEMM KL models. Scale bar, 200 μm. $n = 5$ biologically independent mice examined over 1 independent experiment. **c** Immunofluorescence (IF) analysis of ICAM1+cells (green), CD8+ T cells (red) in resected KL GEMM samples (upper) and orthotopic mouse models (underneath) following different arms of treatment. Scale bar, 100 μm. $n = 5$ biologically independent mice examined over 1 independent experiment. **d** Lysates of mouse lung tumor tissues having received different treatments (vehicle, palbociclib, anti-PD-1 Ab, or combination of palbociclib with anti-PD-1 Ab) were mixed with a cocktail of biotinylated detection antibodies, and then incubated with the Mouse Cytokine Array. Array images were detected using X-ray films. $n = 1$ experiment with $n = 3$ mice. **e, f** C57BL/6 mice were inoculated with LLC1-sh*Lkb1* cells and received the indicated depletion treatments (CD8-depleted, NK1.1-depleted, or ICAM1-depleted treatment). Tumor size (**e** $n = 9$ for vehicle, $n = 10$ for combination, $n = 8$ for ICAM1-depleted, $n = 6$ for CD8-depleted, and $n = 7$ for NK1.1-depleted. ****$p < 0.0001$,

***$p = 0.0009$, **$p = 0.0069$, *$p = 0.0113$.) and survival (**f** $n = 8$ for vehicle, $n = 7$ for combination, $n = 10$ for ICAM1-depleted, $n = 6$ for CD8-depleted, and $n = 6$ for NK1.1-depleted. ***$p = 0.0003$, **$p = 0.0018$, **$p = 0.0059$, *$p = 0.0108$.) were monitored. **g** C57BL/6 mice were inoculated with LLC1/sh*Lkb1* with or without *Icam1* knockout cells and received the combination treatments versus vehicle. Tumor size were monitored. $n = 5$, $n = 7$, $n = 7$, and $n = 6$, respectively. **$p = 0.0053$. **h** Total CD8+ T cells relative to CD45+ cells in tumor tissue, relative MFI of CD44 on CD8+ T cells from mice bearing LLC1-sh*Lkb1* tumors with or without *Icam1* knockout and receiving the indicated treatments were analyzed by flow cytometry. CD8+ T cells, $n = 15$ for sgCon with CDKi treatment, $n = 9$ for sg*Icam1* with CDKi treatment, $n = 18$ for sgCon with combination treatment, and $n = 12$ for sg*Icam1* with combination treatment; **$p = 0.0095$. CD44, $n = 9$ for sgCon with CDKi treatment, $n = 12$ for sg*Icam1* with CDKi treatment, $n = 9$ for sgCon with combination treatment, $n = 9$ for sg*Icam1* with combination treatment; **$p = 0.0015$. (Results are presented as mean ± SEM. A mixed-effects model followed by Tukey's multiple comparisons was performed to compare the tumor growth curves in different treatment groups. Log-rank test was used to analyze the survival data. One-way ANOVA followed by Tukey's multiple comparisons test was used to analyze the flow cytometry data. n.s., not significant;*$p < 0.05$; **$p < 0.01$; ***$p < 0.001$; ****$p < 0.0001$. Source data are provided as a Source Data file.).

further revealed that *LKB1* is a hub gene mechanistically linking the process of cell cycle checkpoint inhibition to the immune activation process.

Finally, investigations about combination immunotherapy in *LKB1*-mutant lung cancer are limited, highlighting the importance of translational preclinical studies. Unlike STING agonists, which have presented challenges in clinical application[20], CDK4/6 inhibitors are clinically approved and suitable for implementation in clinical practice. However, the results of previous studies related to the effect of CDK4/6 inhibitors on antitumor immunity have been inconsistent. Some have reported that CDK4/6 inhibitors strengthen cancer cell immunogenicity, promote an immunomodulatory senescence-associated secretory phenotype (SASP), and induce T effector cell infiltration and activation[28,45–47,49,50]. However, another study found that CDK4/6 inhibitors suppress anticancer immunity by upregulating PD-L1 expression and facilitating cancer immune evasion[31,51]. Therefore, the benefit of CDK4/6 inhibitors in combination with immunotherapy appears to be context-dependent. Here, we demonstrated that CDK4/6 inhibitors induce a favorable immune environment in combination with anti-PD-1 antibodies, which is in an ICAM1-dependent manner.

In summary, this study focused on the clinical dilemma of primary resistance to ICIs in LUAD harboring *LKB1* mutations. The results led us to propose a cancer immune evasion mode wherein mutant tumor cells proactively downregulate ICAM1 expression and escape the adhesion and interaction with T cells in the TME. Personalized combination of CDK4/6 inhibitors with anti-PD-1 antibodies is effective, and might become an option for dealing with this dilemma. Currently, clinical trials investigating the efficacy of this combination strategy in advanced solid tumors are on the process of recruiting patients (NCT02791334). Based on our results, this treatment may also be administered in particularly *LKB1*-mutant LUAD patients, which is worth to be investigated in future clinical studies.

## Methods

### Ethics statement
The study protocol of mouse care and experiments was approved by the Animal Care and Use Committee of Southern Medical University (protocol #SMUL2022189). All animal studies complied with relevant ethical regulations for animal testing and research.

### In vivo mouse models
C57BL/6 mice (5–8 weeks old) were purchased from and maintained in the specific-pathogen-free facility of the Laboratory Animal Center of Southern Medical University (Guangzhou, China). Both male and female mice were used (no selection for sex of mice).

### In vitro CD8+ T-cell co-culture assay
CD8+ T cells were generated from spleen of OT-I mice using EasySep™ Mouse CD8+ T-Cell Isolation Kit (STEMCELL, #19853). Cells were stimulated with 0.5 μg/ml OVA peptide (Sigma, #S7951) in RPMI 1640 containing 10% FBS, 55 μM β-Mercaptoethanol and 5 ng/ml murine IL-2 (PeproTech, #212-22) for 5 days. Lewis lung cancer cells (LLC1) expressing OVA were seeded in 24-well plates and allowed to adhere overnight in complete medium. For detecting T-cell migration, activated CD8+ T cells were then placed on the upper layer of a cell culture insert with permeable membrane. For detecting T-cell-tumor cell interaction, activated CD8+ T cells were directly added to the well containing adhered tumor cells. The effector to target cell (E:T) ratio was 10:1 and cells were maintained in complete medium composed of RPMI 1640 supplemented with 10% FBS for 24 h before detecting. The co-culture systems were analyzed by flow cytometry.

### Genetically engineering mouse model with *Lkb1* deletion
*Kras*^G12D/+ mice were gifts from Professor Liang Chen at Institute of Life and Health Engineering, Jinan University (Guangzhou, China). For *Lkb1* conditional deletion, $2 × 10^4$ pfus of pSECC-sg*Lkb1* lentiviruses were given to *Kras*^G12D/+ mice by nasal inhalation. Tumor formation was analyzed with an MRI (PharmaScan70/16 US) or micro-CT scanner (PINGSENG Healthcare) after 12 weeks of virus infection. Once we had observed gasping from lung tumor-burdened mice and monitored lung nodules, then these mice were randomized into various study cohorts. These enrolled mice had comparable tumor volume/bulk at the outset. Tumor response was recorded every two-week. 3-D slicer software (version 5.0.2) was used to reconstruct MRI volumetric measurements and quantify tumor volumes as described previously[46].

### Cell lines
H460, H1299, A549, H1573, H1944, Calu1, H3255, and HCC4006 cells were provided by Guangdong Lung Cancer Institute or purchased these cells from the American Type Culture Collection (ATCC), and Authentication of Human Cell Lines Reports of these cell lines were provided. All of them were matched with alleles of corresponding cells from ATCC. LLC1 cell line was purchased from Guang Zhou Jennio Biotech Co.,Ltd. NK-92 cells were provided by Chou Yang, who purchased cells from Guang Zhou Jennio Biotech Co.,Ltd. All cells were maintained in a humidified incubator at 37 °C with 5% CO$_2$, and grown in RPMI 1640 or DMEM supplemented with 10% FBS and 100 IU/ml penicillin/streptomycin. All cell lines used were negative for mycoplasma. An updated detection report of mycoplasma was also provided.

## Antibodies

The antibodies used in this study are listed in Supplementary Data 4.

## Immunoblotting

Cells were lysed using RIPA buffer (Beyotime) supplemented with phosphatase inhibitors (PhosSTOP Phosphatase Inhibitor Cocktail, Roche) and protease inhibitors (complete Protease Inhibitor Cocktail, Roche) and were sonicated on ice for 30 s. Protein concentration was determined using a BCA kit (Thermo Fisher Scientific). For each sample, 30 μg protein lysate was loaded onto 10% SDS-PAGE gel. Proteins were separated and transferred to polyvinylidene difluoride (PVDF) membranes (Millipore). Membranes were immunoblotted with primary antibodies as listed in Supplementary Data 4.

## Quantitative real-time PCR

Total RNA was extracted using TRIzol reagent and was reverse transcribed into complementary DNA using PrimeScript™ RT reagent Kit (TAKARA). Real-time PCR was performed using TB Green™ Premix Ex Taq™ (TAKARA) in a Roche LightCycler 480 System. The data were analyzed using the average $2^{-\Delta\Delta CT}$ value. The sequences of the PCR primers used in this study are listed in Supplementary Data 3.

## Chromatin immunoprecipitation assay (ChIP)

A549 cells with different *LKB1* statuses were fixed with 1% formaldehyde and chromatin immunoprecipitation assay was performed with Pierce™ Agarose ChIP Kit (Thermo Fisher Scientific, #26156) according to the manufacturer' protocol. Purified DNA was amplified with Roche LightCycler480 II System using TB Green® Premix Ex Taq™ (Takara Bio, #RR420A). Primers for the *ICAM1* promoter (−625 to −446): 5′-TCCCACGGTTAGCGGTCGCCG-3′ and 5′-CCTCTTTAATC GAGTGGATGAGCC-3′.

## Single cell RNA sequencing data pre-processing

Libraries were sequenced on an Illumina Novaseq platform, paired-end mode. For the raw fastq data, we first filter low quality data with trimomatic software (Version 0.39), followed by computational alignment using CellRanger (version 6.0, 10x Genomics) to map to the mm10 reference genome (GRCm38.91). Subsequently, datasets were subjected to quality control steps using Seurat (R package, version 3.1.5) that included selecting cells with a library complexity of between 500 and 6000 features and selecting genes expressed by at least three cells, and filtering out cells with high percentages of mitochondrial genes (>20%). Finally, 14260 cells were obtained for the downstream analysis. We used Seurat 3.1.5 to first normalize expression matrices by function NormalizeData and ScaleData. Unique molecular identifiers from each cell were scaled by a library size to 10000 and log transformed. To identify major axes of variation within our data, we first examined only highly variable genes across all cells. Then FindVariable function was applied to select the top 200 variable genes and perform principal component analysis. The first 20 principal components were used for further dimensionality reduction by Uniform Manifold Approximation and Projection (UMAP). Then we employed unsupervised clustering using the FindClusters with default parameters, using resolution 0.06, to generate 10 cell clusters. Marker genes defining each cluster were identified using Seurat's FindAllMarkers function, which employs a Wilcoxon rank sum test to determine significant genes. These marker genes were used to assign cluster identity to individual cell type. In addition, cell clusters were also annotated using the CellMarker database (http://bio-bigdata.hrbmu.edu.cn/CellMarker/index.jsp), or based on published signatures and existing literature.

## Cell interaction analysis

We applied CellPhone DB (v2.0.0) to our data to infer ligand-receptor interactions present to look for differences between the two samples. Interaction pairs with P values <0.05 returned by CellPhone DB, were selected for the evaluation of relationships between cell types. The mice gene were transformed to human gene using Biomart, and non-log-transformed UMI counts were used as the expression values for receptors and ligands.

## GO enrichment analysis

We performed GO enrichment analysis with BiomaRt (v2.54.0) and the clusterProfiler (v4.6.0) package. Enrichment scores for selected GO annotations were calculated by a hypergeometric statistical test with a significance threshold of 0.05. The data were plotted as the $-\log_{10} p$ values after Benjamini−Hochberg correction. The significance threshold was set at $-\log_{10}(0.05)$.

## Combined drug screening

The Genomics of Drug Sensitivity in Cancer (GDSC) database was harnessed to discover drugs that were sensitive to lung cancer with *LKB1* mutation. To explore more candidates, drugs with $p$ value less than 0.1 were included. Additionally, the list of an FDA-approved immunology compound library from Selleck was used. Combining the sensitivity and immune effect of the drugs, 5 candidate drugs were chosen followed by experimental validation.

## Animal experiments

Individual mice were injected s.c. with $2{-}5 \times 10^6$ Lewis lung cancer cells (LLC1), or by injecting a total of $1 \times 10^5$ LLC1-sh*Lkb1*-luc cells in 100 μL PBS-matrigel mixture into the left chest (orthotopic tumor model). Seven days after injection, the tumor-bearing mice were randomly separated into four groups. Anti-PD-1 antibody (200 μg/mouse, clone RMP1-14, BioXCell) was injected i.p. three times a week (days 1, 3, and 5 of a 7-day cycle). For CDK4/6 inhibitor (palbociclib, Selleck, S1116, 100 mg/kg), mice were treated p.o. on days 1, 2, and 3 of a 7-day cycle. The combination group was treated with both anti-PD-1 antibody and CDK4/6 inhibitor, and the control group was injected with IgG control (clone 2A3, BioXcell). The tumors were measured in two dimensions (length and width), and volume (V) was calculated as $V = \text{length} \times \text{width}^2 \times 0.5$. Two weeks after treatment, tumors were collected and processed for infiltrating lymphocyte isolation or immunohistochemistry. The vast majority of tumors did not exceed the 2000 mm³ permitted by our animal protocol. In some cases, this limit was exceeded on the last day of measurement and the mice were immediately euthanized.

## Bioluminescence imaging (BLI)

Mice were treated with D-luciferin (150 mg/kg body weight) by intraperitoneal injection. After 10 min, the mice were anesthetized and then analyzed on an AMI HTX imaging system (Spectral Instruments Imaging) for luciferase activity. Regions of interest (ROI) were identified.

## Depletions

Cellular subsets were depleted by administering depleting antibody i.p. twice weekly beginning 1 day prior to therapy, as follows: CD8 T cells with anti-CD8α (200 μg/mouse, clone 2.43, BioXCell) and NK cells with anti-NK1.1 (250 μg/mouse, clone PK136, BioXCell). For ICAM1 blockade, mice were injected intraperitoneally with α-ICAM1 (200 μg/mouse; clone YN1/1.7.4, BioXcell) twice per week.

## RNA sequencing

Human NSCLC cell lines A549 transduced with wild-type or mutant *STK11* or empty vector, and LLC1-sh*Lkb1* tumors with different treatments were subjected to RNA-sequencing. Total RNA was extracted from the indicated cells and tissues, and reverse transcribed into cDNA to construct an indexed Illumina library, followed by sequencing using an Illumina Xten platform and Illumina Novaseq platform, respectively. Gene set enrichment analysis was performed to analyze the RNA-seq data using the gage function and non-parametric Kolmogorov−Smirnov test from the GAGE (version 2.22.0) R Bioconductor package.

## Establishment of intercellular communication signature

To depict the physical cell–cell interaction landscape using our bulk-RNA-seq data, we developed intercellular communication signatures. As described above in our scRNA-seq data analysis, we identified 11 cell types and their corresponding marker genes. For each cell type, we matched its marker gene-encoded surface proteins to their cognate ligands/receptors based on the assembled ligand-receptor pairs from the STRING website. Then a signature containing all the ligands/receptors proteins that can bind to its marker gene-encoded proteins was defined as the communication signature for this cell type. The communication signatures for these 11 cell types were summarized in Supplementary Data 2.

## Signature score computation

Signature scores were calculated by ssGSEA using GSVA v1.40.1 in R. We computed the communication signature scores using published RNA-sequencing data of KL and KP cell lines (GSE137244) (Supplementary Fig. 4B) and our sequencing data of different treatment groups (Fig. 6c), and presented in the form of heatmap. Furthermore, we also rated the immune resistance program utilizing the published RNA sequencing data of cell lines with or without palbociclib treatment (GSE110397) (Fig. 4e, f). Since the immune-resistant program consisted of the "up" component and the "down" components, the final resistant score was defined as the score of "up" part minus the score of "down" part.

## Immunohistochemistry (IHC)

Tumor tissues from mice were fixed in 4% paraformaldehyde overnight, embedded in paraffin and sectioned transversely. The sections were then analyzed by Hematoxylin and eosin (H&E) staining and immunohistochemical staining using antibodies against CD8α (D4W2Z) XP® Rabbit mAb (Mouse Specific) (CST, #98941), CD54/ICAM1 (E3Q9N) XP® Rabbit mAb (CST, #67836), Phospho-Rb (Ser807/811) (D20B12) XP® Rabbit mAb (CST, #8516). The intensity of positive cells (0–25% recorded as 1, 25–50% as 2, 50–75% as 3 and >75% as 4) were evaluated using H score. The quantification of pRB+ nuclei, CD8+ T cells and NKp46+ NK cells were counted from three high power fields per section and were subsequently averaged. All of them were judged by an experienced pathologist.

## Immunofluorescence (IF)

Preliminary process of tumor tissues from mice was similar to that of IHC. Antigen retrieval was performed by high-pressure heating with Tris-EDTA buffer (pH 9.0). Three percent $H_2O_2$ was used to block endogenous peroxidase activity. After serum blocking, the sections were then incubated overnight at 4 °C with mouse ICAM1 (Proteintech, #60299) and CD8 (Abcam, ab217344) antibodies. After washing with PBS-T, the sections were incubated with Dylight 488-conjugated goat anti-mouse IgG(H + L) (Abbkine, #A23210) and Dylight 594-conjugated goat anti-rabbit IgG(H + L) antibodies (Abbkine, #A23420) for 1 h at room temperature. Nuclei were stained with DAPI (Abcam, ab104139).

## Preparation of single cell suspension

Tumors were dissected from mice, cut into small pieces and digested in RPMI 1640 supplemented with 10% FBS, DNase I (0.1 mg/ml) and Collagenase IV (0.5 mg/ml) at 37°C for 30 min with gentle shaking. The mixtures were filtered through 75-μm cell strainers and separated by centrifugation ($300 \times g \times 5$ min) to harvest the single cells. Thereafter, the cells were resuspended in PBS supplemented with 2% FBS and used for subsequent experiments.

## Flow cytometry

Cells were trypsinized, resuspended in PBS supplemented with 2% BSA, incubated with Fc receptor blocking agent (Biolegend, 101302), and stained with fluorescently labeled antibodies for 30 min on ice: FITC-conjugated CD8α, CD4, and PE-conjugated NK1.1, CD45, and PerCP/Cyanine5.5-conjugated CD3, Gr-1, and BV421-conjugated F4/80 and BV510-conjugated CD11b, and APC-conjugated CD107a, CD54, and PE/CY7-conjugated PD-1, PD-L1. For intracellular detection, cells were stimulated with PMA and ionomycin (BD Biosciences) for 6 h, fixed and permeabilized with Cytofix/Cytoperm Kit (BD Biosciences) and stained with specific antibodies: BV510-conjugated IFNγ. For apoptosis assay, an annexinV/PI apoptosis detection kit was used according to the manufacturer's protocol (Biovision). The data were analyzed with FlowJo software (version 10.5; Tree Star).

## Proteome profiler mouse cytokine array

Tumor tissues were excised into small pieces, homogenized in PBS containing protease inhibitors and Triton X-100 (1%) and frozen at −40 °C. After thawing, the samples were centrifuged at $10,000 \times g$ for 5 min to remove cellular debris. Sample protein concentrations were determined and normalized. The relative expression of cytokines and chemokines was quantified using the Proteome Profiler Mouse Cytokine Array Kit, Panel A (ARY006, R&D Systems, Minneapolis, USA) according to the manufacturer's instruction.

## NanoLC-ESI-MS/MS analysis

The lyophilized peptide was resuspended in 2% acetonitrile containing 0.1% formic acid, and 4-μL aliquots were loaded into a ChromXP C18 (3 μm, 120 Å) trap column. The online chromatography separation was performed on the Ekspert NanoLC 415 system (SCIEX, Concord, ON). The trapping and desalting procedures were carried out at a flow rate of 4 μL/min for 5 min with 100% solvent A (water/acetonitrile/formic acid 98/2/0.1%). Then, an elution gradient of 8-38% solvent B (water/acetonitrile/formic acid 2/98/0.1%) was used on an analytical column (75 μm × 15 cm C18-3 μm 120 Å, ChromXP, Eksigent) over 25 min. IDA (information-dependent acquisition) MS techniques were used to acquire tandem MS data on a Triple TOF 6600 tandem mass spectrometer (Sciex, Concord, Ontario, Canada) fitted with a Nanospray III ion source. Data were acquired using an ion spray voltage of 2.4 kV, curtain gas of 35 PSI, nebulizer gas of 12 PSI, and an interface heater temperature of 150 °C. The MS was operated with TOF-MS scans. For IDA, survey scans were acquired in 250 ms and up to 40 product ion scans (50 ms) were collected if a threshold of 260 cps with a charge state of 2–4 was exceeded. A rolling collision energy setting was applied to all precursor ions for collision-induced dissociation. Dynamic exclusion was set for 16 s.

## Co-immunoprecipitation

Proteins were extracted by Cell Lysis Buffer for Western & IP (Beyotime) supplemented with phosphatase and proteinase inhibitor. Primary antibodies against Anti-MYC-Tag mAb (DIA-AN, #2097), FLAG-Tag mAb (DIA-AN, #2064), LKB1 (27D10) Rabbit mAb (CST, #3050), Rb (D20) Rabbit mAb (CST, #9313) and p65 (CST, #8242) were added. The mixtures were incubated at 4 °C with gentle shaking overnight. Protein A/G agarose beads were added and allowed to incubate for another 2 h. The proteins were precipitated through centrifugation and dissolved in SDS sample buffer. Samples were then fractionated by 10% SDS-PAGE gel. Coomassie Blue Fast Staining (Beyotime, #P0017) was performed according to the manufacturer's protocols.

## NK cell co-culture assay

Human NK-92 cells were used for co-culture assays with human NSCLC cells. After pretreating in the presence or absence of drugs for 48 h, human A549 or H460 cells were trypsinized and stained with CellTracker CFSE dye (Thermo Fisher Scientific). The tumor cells were seeded in 24-well plates and allowed to adhere overnight in complete medium. NK-92 cells stained with CellTracker Red dye (Thermo Fisher Scientific) were then added to the plates containing labeled tumor cells. For NK cell adhesion assay, the effector to target cell (*E:T*) ratio was 10:1 and cells

were maintained in complete medium composed of RPMI 1640 supplemented with 10% FBS for 4 h before imaging. For NK cell cytotoxicity assay, NK cells were co-cultured with tumor cells at an E:T ratio of 1.25:1 for 4 h and the target cells were analyzed by flow cytometry.

## Quantification and statistical analysis

Statistical analyses were performed as described in the figure legend for each experiment. Data are presented as mean ± SEM. Group size was determined on the basis of the results of preliminary experiments, and no statistical method was used to predetermine sample size. The indicated sample size (*n*) represents biological replicates. Group allocation and outcome assessment were not performed in a blinded manner. All samples that met proper experimental conditions were included in the analyses. Survival was measured using the Kaplan–Meier method. Statistical significance was determined by one- or two-way ANOVA, Student's *t* test, log-rank test, or Pearson's correlation using Prism 8 software (GraphPad Software) as indicated. Significance was set at $P < 0.05$.

## Reporting summary

Further information on research design is available in the Nature Portfolio Reporting Summary linked to this article.

## Data availability

Both the raw and processed RNA-seq data for single cell RNA sequencing and murine tumors treated by different therapeutic panels generated in this study have been deposited in the GEO database under the following accession code: GSE180963; GSE182228. The publicly available data used in this study are available in the GEO database or the European Nucleotide Archive (ENA) under accession code: GSE72094; GSE126044;GSE135222;GSE93157;GSE110397;GSE137244;GSE110397;PRJEB23709. The RNA-seq TPM matrix of Liu's melanoma immunotherapy cohort[26] can be download from https://www.nature.com/articles/s41591-019-0654-5#additional-information. The remaining data are available within the article, Supplementary Information or Source Data file. Source data are provided with this paper.

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

## Acknowledgements

We are deeply grateful to professor Liang Chen (Institute of Life and Health Engineering, Jinan University, Guangzhou, China) for providing gifts of *Kras*<sup>G12D/+</sup> mice. And we would like to thank Peng Zhou (pathologist in Department of Pathology, The Second Xiangya Hospital, Central South University, Changsha 410011, Hunan, China) for offering technical support in the evaluation of IHC results. We would also like to thank Lei Lv, Xiang-Yu Wu, Yi-Chong Xu, and Jin-Qiang Chen for donating blood for PBMC isolation. In addition, we want to thank Editage (www.editage.cn) for English language editing. This study was supported by the National Natural Science Foundation for Young Scientists of China (Grant No. 81802863 to Z.Y.D., 81902353 to X.B., 82103081 to Z.Z.F.); the National Natural Science Foundation of China (Grant No. 81770173 to X.H.P., 32071452 to X.H.P., 82272820 to Z.Y.D., 82272731 to X.B.); the Outstanding Youths Development Scheme of Nanfang Hospital, Southern Medical University (Grant No. 2020J011 to X.B.); The Fundamental Research Funds for the Central Universities of Jinan Universities (21620341 to Z.Z.F.), Science and Technology Program of Guangzhou (202102021248 to Z.Z.F.).

## Author contributions

Conception and design, Z.Y.D, and X.B.; Organizing data and article writing, X.B., Z.Q.G. and Y.P.Z.; Acquisition of data, X.B., Z.Q.G., Y.P.Z. and L.L.L.; Material support including animals and facilities, D.H.W. Z.Y.D. and X.H.P.; Technical support, Z.Z.F., X.H.P., L.J.L., Y.F., X.R.T. and D.H.W.; Experimental operation, and data acquisition, X.B., Z.Q.G., Y.P.Z., L.L.L., L.J.L., and Y.J.Z.; Analysis and interpretation of data, X.B., Z.Q.G., Y.P.Z., Z.Z.F, L.J.L. and S.C.M.; Statistical analysis, biostatistics, or bioinformatic analysis, J.W. and L.L.; Review and/or revision of the manuscript, Z.Y.D., X.B., X.H.P., Z.Q.G., Y.P.Z., Z.Z.F, L.L.L. and S.C.M.; Study supervision, D.Z.Y., D.H.W., and X.H.P..

## Competing interests

The authors declare no competing interests.
