## [Peer Review File · Nature Communications]

CDK4/6 inhibition triggers ICAM1-driven immune response and sensitizes LKB1 mutant lung cancer to immunotherapyREVIEWER COMMENTS

Reviewer #1 (Remarks to the Author): with expertise in LKB-1 mutant lung cancer, immunotherapy

The manuscript by Bai et al, examines the immune-suppressed phenotype of NSCLC with KRAS;LKB1 co-mutated NSCLC and seeks to probe its mechanistic basis. This is an important scientific area with direct translational potential in view of the high prevalence of STK11/LKB1 mutations in NSCLC and their established role as a driver of resistance to immune checkpoint inhibitor therapy, particularly in the context of KRAS-mutant NSCLC. The authors propose that LKB1-loss driven downregulation of ICAM-1 in tumor cells impairs tumor-effector cell interactions and compromises effector T-cell trafficking and adhesion, ultimately resulting in impaired response to anti-PD-1 that can be reversed by overexpression of Icam-1. Finally, they posit that treatment with the CDK4/CDK6 inhibitor palbociclib can increase Icam-1 expression in Lkb1-deficient cells and restore sensitivity to PD-1 inhibitor therapy. Certain aspects of the work, particularly the effects on LKB1 loss and ICAM-1 expression in immune-cell mediated tumor cell killing assays are novel and interesting. Other components of the proposed mechanism are less well developed and in particular the authors provide little evidence to support the key claims that trafficking and adhesion of effector immune cells are directly impacted by LKB1 loss driven ICAM-1 downregulation. In addition, the immune modulatory potential of CDK4/6i and specifically palbociclib has been previously demonstrated (for example Deng et al, Cancer Discovery 2018) including in the KL, KP and K models and is therefore not novel. Overall, substantial additional experiments are required to increase the rigor of the work and substantiate the key claims. Major points include the following:

1. While the cytotoxic cell killing assays in the A549 LKB1 isogenic system with OE or KD of ICAM-1 are interesting, they assess cytotoxic immune cell killing rather than trafficking or adhesion. In fact no direct data are presented to support the key hypothesis that reduced expression of ICAM-1 in KL cells affects trafficking or adhesion of effector immune cells, which is the crux of this manuscript. The authors need to provide more direct evidence that homing of effector cells is linked to reduced ICAM-1 expression in LKB1-deficient cells. For example, experiments assessing homing of adoptively transferred SIINFEKL-specific CD8+ T cells (OT-I) to LKB1 proficient/deficient tumors (the latter with or without overexpression of ICAM-1) that are engineered to present ovalbumin SIINFEKL peptide may provide important insights into the dominant defect in LKB1-deficient tumors. In addition, does ICAM-1 promote migration of CD8+ T-cells in transwell migration assays?

2. The NK and PBMC assay in figure S5 should be repeated with at least one additional LKB1 proficient/deficient isogenic system as this is an important part of the story.

3. A broader set of cell lines should be assayed to assess the relationship between LKB1 loss and ICAM-1. It should also be determined whether this effect is observed in both KRAS-mutant and WT models in view of clinical data suggesting that the effect of LKB1 loss on immunotherapy responses is dependent on KRAS status. KD of LKB1 should be performed in LKB1 WT lines. It should also be clarified whether the effects in human tumors are observed. Did ICAM-1 expression differ depending on KRAS status in the TMA from 2G?. What about IHC for ICAM-1 in the GEMMS or co-IF with LKB1 expression to assess whether ICAM1 expression correlates with loss of LKB1 on a cell by cell basis.

4. The impact of Icam-1 expression on IO response should be verified in additional isogenic systems and not just in the context of exogenous overexpression. How does endogenous ICAM-1 expression in LLC-1 compare with the LLC1-shLkb1-ICAM-1WT? The authors should knock out Icam-1 in the parental LLC-1 and assess response to immunotherapy, and ideally in at least a second IO-responsive system even if not of lung origin (such as the CT26 syngeneic model in BALB/c).

5. Are the differences in ICAM expression in the human tumors (for example Figure 2G) observed in tumor cells, immune cells or both? This should be clarified. Furthermore, does treatment with palbociclib induce ICAM-1 expression also in immune cells? This should be easy to assess using co-IF in the syngeneic and GEMM models.

6. It would be important to present more comprehensive and quantified immune profiling data from the GEMM K and KL models as well as changes in TME in response to treatment. Although the effects on CD8+ TILs observed in the GEM KL models are in keeping with previous reports, the authors don't observe any changes in myeloid cells populations and in particular, tumor associated neutrophils, that have consistently been reported to be increased in the context of LKB1 loss in both GEMMs and syngeneic model of KRAS-mutant NSCLC. It is not clear what accounts for this discrepancy. Mosaic LKB1 loss might play a role, I note that LKB1 loss is partial in WBs from tumors. Did the scRNA analysis confirm reduced LKB1 expression in the model? Additional confirmation of LKB1 loss by IHC and its relationship to ICAM1 expression should be presented. Were compared tumors of comparable size? In addition, treatment-induced changes in myeloid cell subpopulations are not adequately examined despite previously published work that palbociclib can impact myeloid cell subsets. Finally, more detailed analysis of the functional effects of ICAM-1 OE on the TME in the LLC1-shLkb1 system is required (Figure 3) is required analogous to the analysis of the impact of palbociclib treatment in later figures. Absolute (as % of CD45+ cells) and relative fractions of immune cells subsets should be presented, including effects on myeloid cells and more detailed characterization of proliferation, activation and differentiation status of TILs.

7. Based on the author's hypothesis, LKB1 competes with CDK4 for RB binding. However, the effects of LKB1 on RB phosphorylation and ICAM-1 expression appear to be kinase activity-dependent. How would LKB1 kinase activity impact LKB1-RB binding in this model? Does exogenous kinase-deficient LKB1 interact with CDK4 and RB? Does overexpression of kinase deficient LKB1 affect the CDK4-RB interaction? These questions should be addressed.

8. Despite inclusion of several experiments the signaling axis linking LKB1 loss with reduced ICAM-1 expression is not clear. The authors initially posit that RB/E2F activity is involved and include a series of experiments but then also propose that NF- κ B signaling may be mediated however these latter experiments are limited. Is there increased NF κ B occupancy in the promoter of ICAM-1 in LKB1 deficient compared to proficient cells? Does KD of p65 affect expression of ICAM-1?

9. It is critical to prove the extent to which the increased sensitivity to palbociclib and a-PD-1 is due to upregulation of ICAM-1 (since palbociclib can have pleotropic effects). The authors should establish isogenic derivatives of the LLC1-shLkb1 cell line with or without knockout of ICAM-1 using CRISPR/Cas9 and repeat the therapeutic experiment. The experiment shown in Figure 7EF does not adequately address this, since anti-ICAM-1 may also target ICAM-1 in non-tumor cells. The effects of treatment with palbociclib on the tumor microenvironment in the setting of ICAM-1 depletion should also be evaluated (FACS, IHC).

10. The degree of selectivity of the palbociclib/anti-PD-1 approach for LKB1-mutant/deficient NSCLC should be assessed by performing the same experiment in the parental cell line. To what extent is this cell line uniquely sensitive to the palbociclib/anti-PD-1 combination. To what extent are the immune-modulatory effects dependent on Lkb1 status?

11. There are some methodological concerns regarding the in vivo preclinical experiments in Fig 5:

- Treatment appears to start very early in the LLC1-shLkb1 model in Figure 5A. The experiment should be performed with randomization to treatment arms occurring when tumors reach adequate volume (for example 300mm³)
- In Figure 5D, treatment bulk at the time of treatment initiation appears to differ between the groups, with mice in the combo arm apparently having the lowest tumor bulk. Although the authors measure the relative change in photon flux after one cycle of treatment compared to baseline clearly efficacy can be affected by the amount of tumor present at baseline
- The change in tumor volume in Figure 5F in the GEMM preclinical experiment should be quantified and formally compared between the different groups. It is not clear from the methods section when treatment started and whether mice had comparable tumor volume/bulk at the outset. It is not clear from the presented data whether the combination is effective in the autochthonous GEMM model.

One additional limitation is that the model will likely exhibit mosaic and non-uniform deletion of Lkb1 (as it is based on infection with a PSECCsgLkb1 vector).

12. The clinical datasets of immunotherapy- treated patients in Figure 3A are small and incompletely characterized. Based on the presented data it is not clear whether ICAM-1 is predictive of poor outcomes from immunotherapy specifically or simply represents an adverse prognostic marker. These effects should be clarified. It should also be established whether the impact of ICAM-1 expression on clinical outcomes with immunotherapy is independent of other variables, such as PD-L1 expression etc.

Minor points:

13. Several legends in both main figures and particularly supplemental figures don't appear to correspond to the part of the figure referenced. For example, S6d, S7G, S4f and Figure 2D. In addition the manuscript would benefit from additional editing in several areas.

14. The effects of LKB1 KD on p-p65 expression are not identical in Figures S6A and S6C

15. It appears that OE of even mutant (kinase dead) LKB1 induces ICAM1 expression. How is this explained?

16. Was the LLC1-shLkb1-IcamWT cell line also infected with Lenti-control or was this not infected at all?

17. In Figure S5E transfection with s-NC seems to result in increased ICAM-1 expression over a period of 48 hours – does transfection itself result in increased expression?'

Reviewer #2 (Remarks to the Author): with expertise in LKB-1 mutant lung cancer, immunotherapy

NCOMMS-21-37118

Title: CDK4/6 inhibitor triggers ICAM1-driven immune response and sensitizes LKB1 mutant lung cancer to immunotherapy

In this manuscript, the authors proposed an interesting idea of ICAM is responsible for the responses of augmented response to anti-PD1 in LKB1 mutant tumors through CDK4/6 inhibition. The biggest issue with the series of experiments shown is that the authors did not definitively demonstrate that the phenotype is driven by ICAM1 expression on cancer cells. Furthermore, whether sensitivity in response to CDK4/6 plus PD1 inhibition is LKB1-loss dependent remains unclear. Specifically, there are questions below need to be addressed.

1. Figure 1, in page 5, the authors stated "n comparison with K mouse, KL mouse had lower number of TCF7+ activated T cells and higher number of CTLA4+ exhausted T cells". It's not really shown in the figures for CTLA4+ levels or quantification of relative populations change in K and KL model. This issue is throughout the whole session of scRNAseq data, which makes interpretation of the data challenging, and need to be addressed.

2. Figure 2C, the CellphoneDB analysis was meant to analyze the interaction between ligand and receptors rather than cell-cell interaction. The authors need to specify which ligand/receptor was used for the analysis. In addition, the authors only lay out three different complexes of ICAM1, including LFA-1, ALB2 and SPN. More details are needed for interpretation of the data.

3. Figure 3A-C, the authors showed ICAM1 levels in patients is correlated with PFS in 3A and 3B. While in mouse model used, ICAM1 levels do not affect tumor growth or survival (3C), instead, it impacts on response to anti-PD1 treatment. Why is this discrepancy between human and animal data?

4. Figure 3F, the authors claims that the increased sensitivity to anti-PD1 treatment is due to increased CD8+ T cell infiltration and less exhausted T cells, based on the rational that ICAM1

mediated cancer cell-T cell adhesion and interaction (Figure 2C). However, the following ex vivo experiment showed interaction between ICAM1 expressing tumor cells with PBMC and NK cells. There is no data proving ICAM1 high tumor cells impact on their interaction with T cells. This is very important point need to be proved since the whole story is built upon this hypothesis.

5. Figure 3H, page 12, the authors trying to use IHC examine the “trafficking and infiltration of CD8+ T cells into the TME” (page 12). This assay cannot really distinguish between trafficking versus proliferation of T cells, it merely showed total CD8+ T cells at tumor site. This part needs to be clarified/corrected.

6. Figure 4 and supporting evidences, the IP and MS analysis for the interaction between LKB1 and Rb (Fig. S6B) is very obscure and need further clarification. For the regulation of ICAM1 by LKB1, the authors showed it is through Rb (Fig S6H). Looks like ICAM1 levels depend on Rb1 levels rather than LKB1, thus whether LKB1 is deficient is not relevant to either ICAM1 levels. In line with this, the authors further showed LKB1 deficiency “led to a decreased phosphorylation level of p65” (page 16), which is responsible to transcriptional regulation of ICAM1. How about Rb1 status when p65 is inhibited? This part is very confusing since the authors trying to demonstrate two different factors Rb1 and p65, further experimental evidence is needed (either role of p65 in Rb1 deficient cells, or role of Rb1 in p65 knockdown cells during ICAM1 regulation). Figure 4D, what’s the status of ICAM1 levels in LKB1 WT cells?

7. Figure 5, the authors only showed LLC-shLKB1 tumor in response to palbo+PD1, how about LKB1 wt LLC parental cells? Do they respond to palbo+PD1? More importantly, if you overexpress or knockdown ICAM1, will the tumor still respond to palbo+PD1? Obviously CDK4/6 inhibitor can directly activate T cells and augment response to PD1 blockade, the authors need to segregate the effect of CDK4/6 inhibition on tumor cells versus T cells (and other immune cells), and demonstrate whether the phenotype observed is due to upregulation of ICAM1.

Reviewer #3 (Remarks to the Author): with expertise in scRNAseq, system immunology

It is an interesting topic to study the underline mechanism between LKB1 mutation and the resistance to immune checkpoint inhibitors (ICIs) treatment in lung adenocarcinoma (LUAD). Novel therapies are urgently needed to improve clinical outcomes of LUAD patients with LKB1 mutation, as they are refractory to most currently available therapies.

The authors applied single-cell RNA-seq technology to profile the tumor microenvironment with genetically engineered mouse models (Lkb1-loss/Kras mut). They found LKB1 mutant tumors reduced interaction with infiltrated immune-competent cells by downregulation of the cell surface protein ICAM1. They further demonstrated CDK4/6 inhibitor would rescue ICAM1 expression in LKB1 mutant cell line models and in vivo survival study also shows CDK4/6 inhibitor plus anti-PD-1 antibody has synergistic effects and induced immunologically “hot” tumor microenvironment. Although both the topic and findings are intriguing, the overall logical flow of the paper and the connections between evidence and conclusion need major revision. Some descriptions in the method section are not clear enough.

Major:

1. Figure1, what are those activated T or exhausted T clusters? Any specific T cell markers to show their identity? And why using PD-1 antibody for combination instead of anti-CTLA4 antibody, exhausted T cells seems expression high levels of CTLA-4.

2. Figure3. A and B show the ICAM1 expression level correlates with patient survival (LUAD and melanoma). How about adding LKB1 mutation status as a cofactor. For the survival study design, to rule out the possibility that ICAM1 overexpression alone is enough to show benefit from PD-1 blockade treatment, I would suggest adding the parental lewis lung cancer cells without Lkb1 knockdown with or without ICAM1 over-expression as controls. For figure3G, in the main text, the

authors mentioned using the GSE72094 dataset, while the figure legend label shows the TCGA dataset, please clarify.

3. Figure 2. E and F show the ICAM1 protein levels in LKB1-MUT and LKB1-loss cells are significantly different. Is there any difference between different LKB1 mutations, both in the cell line model and patients?

4. Figure 6. C is hard to understand and the communication signature score calculation method needs a clear description.

5. The experimental evidence to support LKB1 competing with CDK4/6 and modulate RB phosphorylation is too weak. Figure S6F, IP endogenous RB protein and check the CDK4 and LKB1 protein binding would tell if the two bind to the similar site on RB.

6. The authors provided very limited evidence to support the very detailed mechanisms and somehow, it's confusing to catch the key point (the final model figure 7G): for example, they mentioned deficiency of LKB1 activate DNMT1 and thereby silencing STING expression, meanwhile, they also mentioned ICAM1 might be regulated by STING signaling.

Minor:

1. Figure 2. B please label the gene numbers in the Venn diagram. Since H460 was used as a model as well. Any specific reason why only shows A549 DE genes?

2. Figure 2. C is difficult to understand. Please add more description or reorganized the plot. If the point is to show ICAM1 from cancers will recruit T cells by T cell surface proteins (ALB2, LFA, SPN). I don't think it's necessary to show ICAM1 from T cell and T cell surface proteins in cancer cells.

3. Figure S6H add phospho-Rb blot should be added.

4. Figure 6. B, I would suggest adding a quantification plot for easier reading the result.

5. Figure 2C, 2D labels were swapped.

6. Figure S5. B no legend on the plot (the meaning of the dots?). What is the purpose of using NK-92 or activated PBMCs?

Reviewer #4 (Remarks to the Author): with expertise in CDK4/6

Summary: The manuscript from Bai et al. interrogates the coordinate impact of KRAS and LKB1 on the immunological landscape in the lung tumor microenvironment. Using single cell sequencing they show that tumors with combined KRAS/LKB1 perturbation exhibit more T-cell exhaustion, with less activated T-cells. Analysis of the tumor compartment revealed that ICAM1 deficiency is associated with LKB1 deficient tumors, which was interrogated in mouse and human preclinical models, as well as TCGA data. ICAM1 expression is associated with improved outcome with anti-PDL1 based therapies in pre-clinical models and melanoma and lung cancer clinical cohorts. CDK4/6 inhibitors can increase ICAM1 in an RB-dependent fashion, which is believed to involve a direct interaction of LKB1/RB/CDK4. Subsequently, CDK4/6 inhibition cooperates with immune checkpoint inhibition.

Critique: Overall this is a well performed study investigating the impact of LKB1/KRAS on the tumor microenvironment and immunological responses. However, there is concern relative to the interpretation of results and the intersection with multiple other studies in this area of research. In particular, given that multiple studies have described the basis for which LKB1 deficiency impact on the tumor microenvironment the contribution of this study needs to be carefully evaluated versus these other mechanisms which have been described in comparable models.

Specific points are enumerated below.

1. To indicate that the tumor microenvironment is an immune desert in the KL mice seems hard to reconcile with the substantial infiltration within the tumor microenvironment. Rather T-cell activation status is the main determinant as well as additional B-cells. The quantitation/reproducibility of the findings from Figure 1 would be important in fully establishing the "atlas" relative to these studies in this area.

2. It is difficult to reconcile the work relative to the tumor immune microenvironment in this study vs. the work of others that have ascribed different mechanisms (e.g. STING) in modulating the immunological milieu and response to immunecheckpoint inhibitors.

3. Given the correlation between ICAM1 and LKB1 the results in Figure 3A/B are confirmatory of the known relationship of LKB1-status and response to immune-checkpoint inhibitor. The increase in CD8+ T-cells is relatively marginal with the increase expression of ICAM1 and there is no evaluation of activation/exhaustion status. If in fact, ICAM1 is the full mechanisms by which LKB1 deficiency acts, it should be possible to shift the totality of the immune profile (e.g. by single cell analysis) back to that of an LKB1 wild-type tumor.

4. RB knockdown and effect on ICAM1 is difficult to appreciate in Figure 4. Similarly, while it is suggested that RB/CDK4/LKB1 form a complex the data are by no means definitive and just raises the question of what the complex is doing and/or relevance to ICAM1 expression?

5. There is an extensive literature that CDK4/6 inhibitors will augment the response to immune checkpoint inhibition via multiple mechanisms. These are known to be RB-dependent. The work herein would seem to be at odds with work from other labs that CDK4/6 inhibitors alone are insufficient to have a potent impact on immune-checkpoint inhibitors or the tumor microenvironment in lung cancer models. More in depth analysis would be important to illustrate that these effects are ICAM1-dependent in the tumor compartment.

REVIEWER COMMENTS

Reviewer #1 (Remarks to the Author): with expertise in LKB-1 mutant lung cancer, immunotherapy

The manuscript by Bai et al, examines the immune-suppressed phenotype of NSCLC with KRAS; LKB1 co-mutated NSCLC and seeks to probe its mechanistic basis. This is an important scientific area with direct translational potential in view of the high prevalence of STK11/LKB1 mutations in NSCLC and their established role as a driver of resistance to immune checkpoint inhibitor therapy, particularly in the context of KRAS-mutant NSCLC. The authors propose that LKB1-loss driven downregulation of ICAM-1 in tumor cells impairs tumor-effector cell interactions and compromises effector T-cell trafficking and adhesion, ultimately resulting in impaired response to anti-PD-1 that can be reversed by overexpression of Icam-1. Finally, they posit that treatment with the CDK4/CDK6 inhibitor palbociclib can increase Icam-1 expression in Lkb1-deficient cells and restore sensitivity to PD-1 inhibitor therapy. Certain aspects of the work, particularly the effects on LKB1 loss and ICAM-1 expression in immune-cell mediated tumor cell killing assays are novel and interesting. Other components of the proposed mechanism are less well developed and in particular the authors provide little evidence to support the key claims that trafficking and adhesion of effector immune cells are directly impacted by LKB1 loss driven ICAM-1 downregulation. In addition, the immune modulatory potential of CDK4/6i and specifically palbociclib has been previously demonstrated (for example Deng et al, Cancer Discovery 2018) including in the KL, KP and K models and is therefore not novel. Overall, substantial additional experiments are required to increase the rigor of the work and substantiate the key claims. Major points include the following:

1. While the cytotoxic cell killing assays in the A549 LKB1 isogenic system with OE or KD of ICAM-1 are interesting, they assess cytotoxic immune cell killing rather than trafficking or adhesion. In fact, no direct data are presented to support the key hypothesis that reduced expression of ICAM-1 in KL cells affects trafficking or adhesion of effector immune cells, which is the crux of this manuscript. The authors need to provide more direct evidence that homing of effector cells is linked to reduced ICAM-1 expression in LKB1-deficient cells. For example, experiments assessing homing of adoptively transferred SIINFEKL-specific CD8+ T cells (OT-I) to LKB1 proficient/deficient tumors (the latter with or without overexpression of ICAM1 that are engineered to present ovalbumin SIINFEKL peptide may provide important insights into the dominant defect in LKB1-deficient tumors. In addition, does ICAM-1 promote migration of CD8+ T cells in transwell migration assays?

Response: Thank you so much for your valuable, insightful and professional suggestion. We need more direct data to support the key hypothesis that downregulation of ICAM1 in KL cells affects trafficking or adhesion of effector immune cells. Therefore, as with your suggestions, LKB1 proficient/deficient lung tumor bearing orthotopic mouse models (the latter with or without overexpression of ICAM-1) engineered to present ovalbumin SIINFEKL peptide were established (LLC1-shcontrol-*Icam1*^{WT}-OVA; LLC1-sh*Lkb1*-

Icam1^{WT}-OVA; LLC1-*shLkb1-Icam1*^{OE}-OVA). Subsequently, CD8⁺ T cells isolated from the spleen of OT-I mice were stimulated with 2 nM OVA_{257–264} (SIINFEKL) for 5 days with the presence of 10 ng/mL IL-2. Then the cells in the culture were dyed with cell-trace far-red and viewed as CTLs. 5×10⁶ OT-I CTLs were injected intravenously into LLC1-OVA tumor bearing mice (Supporting Figure 1A). Gratifyingly, on day 5 after T cells injection, we observed a sharp decrease of OT-I cells in LKB1 deficient tumors than in LKB1 proficient tumors, indicating the role of LKB1 in influencing CTL infiltration; furthermore, the number significantly increased in LKB1 deficient tumors with ICAM1 overexpression than LKB1 deficient groups (Supporting Figure 1B). Besides, we also investigated into the activated state of OT-I CTLs. CD69 expression was significantly lower in LKB1 deficient tumors than in proficient groups. And ICAM1 overexpression significantly improved the activated state of OT-I CTLs in LKB1 deficient tumors (Supporting Figure 1B). Taken together, these data suggest that ICAM1 may be prerequisite for the homing of activated tumor-specific T cells in LKB1 deficient tumors.

Supporting Figure 1: ICAM1 may be prerequisite for the homing of activated tumor-specific T cells in LKB1 deficient tumors. (A) LLC1-OVA tumor cells (LLC1-*shcontrol-Icam1*^{WT}-OVA; LLC1-*shLkb1-Icam1*^{WT}-OVA; LLC1-*shLkb1-Icam1*^{OE}-OVA) were implanted. Previously-marked (APC⁺) and stimulated CD8⁺ T cells expressing the OT-I were transferred on day 7 and analyzed on day 12. Infiltration number of OT-I cells and expression of CD69 on OT-I cells were analyzed by flow cytometry. (B) Results are presented as mean ± SEM. One-way ANOVA test was used to analyze the data. ns, not significant; *p<.05; **p<.01; ***p<.001; ****p<.0001.

In addition, we also performed transwell migration assays to verify that LKB1 promotes migration of CD8⁺ T cells dependent on ICAM1. We observed that A549-*LKB1*-WT cells were able to significantly increase CD8⁺ T cell migration (Supporting Figure 2A). Overexpression of ICAM1 significantly increased CD8⁺ T cell migration in LKB1 deficient groups (Supporting Figure 2B), while knockdown of ICAM1 abrogated CD8⁺ T cell migration in LKB1 WT group (Supporting Figure 2C).

Supporting Figure 2: LKB1 promotes the migration of CD8⁺ T cells which is dependent on ICAM1. T cell migration assay of A549 cell lines. (A) Conditioned medium from A549 parental cells (*LKB1*-loss) or *LKB1*-overexpressing A549 cells (*LKB1*-WT and *LKB1*-mutant) were collected and filled in the bottom well. Activated PBMCs were loaded into the top chamber of transwell inserts. PBMCs that migrated to the bottom layer were stained by crystal violet blue. Representative images (left) and quantification (right) were shown. (B-C) Overexpression of ICAM1 was conducted in *LKB1*-loss and *LKB1*-mutant A549 cells (B), and knockdown of ICAM1 was performed in *LKB1*-WT A549 cells (C). These cells were subsequently subjected to T cell migration assay. Representative images (left) and quantification (right) were shown. (Results were presented as mean \pm SEM. Student's t-test was used to compare the variables of two groups, and one-way ANOVA was performed for multi-group comparisons. n.s., not significant; * $p < .05$; ** $p < .01$; *** $p < .001$; **** $p < .0001$)

Besides, we repeated these experiments in LLC1 cells with or without *Lkb1* knockdown. Similarly, we observed that CD8⁺ T cell migration decreased in LLC1-sh*Lkb1* group, and overexpression of ICAM1 increased CD8⁺ T cell migration in *LKB1* deficient groups (LLC1-

shLkb1) (Supporting Figure 3).

Supporting Figure 3: T cell migration assay of LLC1 cell lines. CD8+ T cells were sorted from spleen of OT-I mice and were stimulated for 5 days. LLC1 cells expressing OVA with optional expression of LKB1 and ICAM1 were seeded in 24-well plates and allowed to adhere overnight in complete medium. Activated CD8+ T cells were then placed on the upper layer of a cell culture insert with permeable membrane for 24h. Cells that migrated to the bottom layer were stained by crystal violet blue. The E: T ratio (effector to target) was 5: 1 for for CD8+ T cells. Representative images (left) and quantification (right) were shown. Results are presented as mean \pm SEM. One-way ANOVA test was used to analyze the data. ns, not significant; * p <.05; ** p <.01; *** p <.001; **** p <.0001.

To further validate our hypotheses, cell adhesion assay was performed. As expected, the number of adhesive NK or PBMC in *LKB1*-WT cells increased (Supporting Figure 4A). Exogenous expression of ICAM1 strongly enhanced the binding of NK cells to tumor cells (Supporting Figure 4B). Together, we demonstrated that LKB1 facilitated CD8+ T cell adhesion and migration in vitro through ICAM1.

Supporting Figure 4: LKB1 facilitated CD8+ T cell adhesion in vitro through ICAM1. (A) A549 parental cells (*LKB1*-loss) or *LKB1*-overexpressing A549 cells (*LKB1*-WT and *LKB1*-mutant) stained with CellTrace CFSE dye were seeded in 24-well plates and allowed to adhere overnight in complete medium. Immune cells (NK-92 cells and PBMCs) stained with CellTrace CFSE Red dye were then added to the plates containing labeled tumor cells and cocultured for 4h before complete washing and imaging. Representative images (left) and quantification (right) were shown. (B) Overexpression of ICAM1 were conducted in *LKB1*-loss and *LKB1*-mutant A549 cells and adhesion assay with NK-92 cells were

performed. Representative images (left) and quantification (right) were shown. Results are presented as mean \pm SEM. One-way ANOVA test was used to analyze the data. ns, not significant; * $p < .05$; ** $p < .01$; *** $p < .001$; **** $p < .0001$.

Corresponding changes and explanations we have added in the revised manuscript in the results part were as follows:

Page 11: To fully demonstrate that ICAM1 mediated cancer cell-T cell adhesion and interactions affects trafficking and activation of effector immune cells in LKB1 deficient NSCLC, we adoptively transferred congenitally-marked (APC+) stimulated CD8+ T cells expressing the OT-I, which is specific for OVA, into mice bearing LKB1 proficient/deficient orthotopic lung tumors (LLC1-shcontrol-Icam1WT-OVA; LLC1-sh*Lkb1*-Icam1WT-OVA; LLC1-sh*Lkb1*-Icam1OE-OVA) and assessed OT-I cell infiltration and activation status in lung tumors (Fig. 3a). Gratifyingly, on day 5 after T cells injection, we observed a sharp decrease of OT-I cells in LKB1 deficient tumors than that in LKB1 proficient tumors, indicating the role of LKB1 in influencing CTL infiltration; furthermore, the number significantly increased in LLC1-sh*Lkb1*-Icam1OE-OVA tumors than LLC1-sh*Lkb1*-Icam1WT-OVA groups (Fig. 3b). Besides, ICAM1 overexpression significantly improved the activated state of OT-I CTLs (CD69) in LKB1 deficient tumors (Fig. 3b). Taken together, these data suggest that ICAM1 may be prerequisite for the homing of activated tumor-specific T cells in LKB1 deficient tumors.

Page 12: Another co-culturing experiments demonstrating the mechanism by which LKB1 induces an antitumor effect through ICAM1 were performed (Supplementary Fig. 8A-C). KL parental cells (KRAS-mutant, LKB1-loss; A549 and H460; LV-Ctrl) or LKB1-overexpressing cells (LV-LKB1-WT and LV-LKB1-mutant) were co-cultured with human NK-92 cells or activated human PBMCs. LKB1-overexpressing cells exhibited higher levels of apoptosis than LKB1-deficient cells after coculturing with activated T or NK cells. Overexpression of ICAM1 significantly enhanced tumor cell apoptosis, while knockdown of ICAM1 inhibited NK-mediated apoptosis (Supplementary Fig. 8D-J), which indicated that LKB1-induced differences in the response to cytotoxic immune cells were dependent on ICAM1. T cell migration and cell adhesion assay furtherly proved that LKB1 facilitated CD8+ T cell adhesion and migration in vitro through ICAM1 (Supplementary Fig. 9-10, 11A-B).

2.The NK and PBMC assay in figure S5 should be repeated with at least one additional LKB1 proficient/deficient isogenic system as this is an important part of the story.

Response: Thank you so much for your useful suggestion. We repeated the assay in another LKB1 proficient/deficient human lung adenocarcinoma cells H460, according to your comments. H460 parental cells (*LKB1*-loss) or LKB1-overexpressing H460 cells (*LKB1*-WT and *LKB1*-mutant) were co-cultured with human NK-92 cells or activated human PBMCs. Similarly, LKB1-overexpressing cells exhibited higher levels of apoptosis than LKB1-deficient cells after coculturing with activated PBMCs or NK cells (Supporting Figure 5A). Overexpression of ICAM1 significantly enhanced tumor cell apoptosis, while knockdown of ICAM1 inhibited NK-mediated apoptosis (Supporting Figure 5B), which again demonstrated that LKB1 influences cytotoxicity of immune cells through ICAM1.

Supporting Figure 5: LKB1 influences cytotoxicity of immune cells through ICAM1 in H460 cells. (A) Flow cytometry analysis of apoptosis by Annexin V/PI staining in the H460 cell line with different LKB1 status co-cultured with human NK-92 cells or activated peripheral blood mononuclear cells (PBMCs) from volunteer blood donors for 4h. The E: T ratio (effector to target) was 1.25: 1 for NK-92 cells and 40: 1 for PBMCs. Representative images (left) and quantification results (right) were shown. (B) Assessment of apoptosis in ICAM1-overexpressing H460 (B) cell lines. Cancer cells were co-cultured with human NK-92 cells for 4h. The E: T ratio (effector to target) was 1.25: 1. Representative images (left) and quantification results (right) were shown. Results are presented as mean \pm SEM. One-way ANOVA test was used to analyze the data. ns, not significant; * p <.05; ** p <.01; *** p <.001; **** p <.0001.

Corresponding changes and explanations we have added in the revised manuscript in the results part were as follows:

Page 12: KL parental cells (KRAS-mutant, LKB1-loss; A549 and H460; LV-Ctrl) or LKB1-overexpressing cells (LV-LKB1-WT and LV-LKB1-mutant) were co-cultured with human NK-92 cells or activated human PBMCs. LKB1-overexpressing cells exhibited higher levels of apoptosis than LKB1-deficient cells after coculturing with activated T or NK cells. Overexpression of ICAM1 significantly enhanced tumor cell apoptosis, while knockdown of ICAM1 inhibited NK-mediated apoptosis (Supplementary Fig. 8D-J), which indicated that LKB1-induced differences in the response to cytotoxic immune cells were dependent on ICAM1. T cell migration and cell adhesion assay furtherly proved that LKB1 facilitated CD8+ T cell adhesion and migration in vitro through ICAM1 (Supplementary Fig. 9-10, 11A-B).

3. A broader set of cell lines should be assayed to assess the relationship between LKB1 loss and ICAM-1. It should also be determined whether this effect is observed in both KRAS-mutant and WT models in view of clinical data suggesting that the effect of LKB1 loss on immunotherapy responses is dependent on KRAS status. KD of LKB1 should be performed in LKB1 WT lines. It should also be clarified whether the effects in human tumors are observed. Did ICAM-1 expression differ depending on KRAS status in the TMA from 2G? What about IHC for ICAM-1 in the GEMMS or co-IF with LKB1 expression to assess whether ICAM1 expression correlates with

loss of LKB1 on a cell-by-cell basis.

Response: Thank you so much for your valuable suggestions which we think our work benefited a lot from your suggestions.

1) We firstly investigated the relationship between LKB1 loss and ICAM1 expression in a broader set of cell lines. *KRAS* mutant lung adenocarcinoma cell lines in the CCLE repository were subdivided into 2 classes: K (n=14) harboring a *KRAS* mutation with intact *LKB1*, and KL (n=7) harboring a *KRAS* and *LKB1* mutation, respectively. And we found that ICAM1 was significantly reduced in KL group than K group (Supporting Figure 6A). We also validated our findings in a broader set of cell lines through western blotting analysis. LKB1 loss was robustly associated with downregulation in ICAM1 protein levels (Supporting Figure 6B).

2) Besides, we also probed into whether this effect is observed both *KRAS*-mutant and WT models. We compared ICAM1 expression in *LKB1* mutation versus *LKB1* intact group among *KRAS* WT models in CCLE. And ICAM1 was also reduced in *LKB1* mutant groups (Supporting Figure 6A).

3) We also explored ICAM1 expression level by knocking down LKB1 in LKB1 WT cell lines (H1299, CALU1, and LLC1). And we found that knocking down LKB1 using siRNA or shRNA led to downregulation of ICAM1 expression (Supporting Figure 6C).

4) We subsequently validated our findings from lung cancer patient samples in TCGA. ICAM1 expression levels were significantly downregulated in patients harboring *LKB1* mutation, regardless of *KRAS* co-mutation, and different *LKB1* mutation types (Supporting Figure 6D-E). According to study by Ferdinandos Skoulidis, they concluded that the effect of STK11/LKB1 deficiency on PD-1 blockade immunotherapy extends to the entire LUAC population, regardless of *KRAS* status (Ferdinandos Skoulidis, 2019). We also found that LKB1 deficiency alone leads to downregulation of ICAM1, irrelevant of *KRAS* status.

5) To further validate this finding, according to your valuable suggestions, we used IHC and CO-IF to analyze tumor cell-specific ICAM1 protein levels in tumor sample of GEMMS models (Supporting Figure 6F-G) and across patient-derived NSCLC samples (Supporting Figure 7). LKB1 deficiency was robustly associated with significant reduction in ICAM1 levels in tumor cells, in both mouse samples and patients' tissue. Taken together, these data confirmed that LKB1 deficiency is associated with downregulation of ICAM1 in lung cancer.

6) Besides, according to your valuable suggestion, we also verified if ICAM1 expression differ depending on *KRAS* status in the TMA from 2G. Among 7 *LKB1* mutated patients, only 2 of them with *KRAS* mutation, while in 11 *LKB1* WT patients, all of them harbored *KRAS* mutation. We compared ICAM1 expression level in $KRAS^{mutant}LKB1^{mutant}$ patients with $KRAS^{WT}LKB1^{mutant}$ patients, almost all of them had low expression of ICAM1 on tumor tissue, therefore ICAM-1 expression did not differ depending on *KRAS* status in the TMA from 2G (Supporting Figure 7).

Supporting Figure 6: LKB1 deficiency is associated with downregulation of ICAM1 in lung cancer. (A) Analyses of correlation between ICAM1 expression and different LKB1 mutation types, with or without KRAS co-mutation using CCLE dataset. (B) Western blotting of the indicated proteins in lung cancer cells with LKB1-loss or LKB1-intact. (C) Western blot showing ICAM1 expression of lung cancer cell lines (CALU1, H1299 and LLC1) treated with or without small interference or short hairpin RNA mediated knocking down of LKB1. (D-E) Analyses of correlation between ICAM1 expression and different LKB1 mutation types, with or without KRAS co-mutation using TCGA dataset. (F) Representative immunofluorescence images of K and KL lung tumors using anti-LKB1 and anti-ICAM1 antibodies. Scale bar, 50 μ m (40 \times). (G). Representative images of IHC staining of K and KL lung tumors using anti-LKB1, anti-CD8 α and anti-ICAM1 antibodies. Scale bar, 50 μ m (40 \times). Results are presented as mean \pm SEM. T test or One-way ANOVA test was used to analyze the data. ns, not significant; * p <.05; ** p <.01; *** p <.001; **** p <.0001.

A *LKB1*-WT

B *LKB1*-Mutant

Supporting Figure 7: IHC of ICAM1 and CD8 expression in *LKB1*-WT (A) and *LKB1*-mutant (B) lung cancer patients. Scale bar, 20 μ m (40 \times), 50 μ m (20 \times).

Corresponding changes and explanations we have added in the revised manuscript in the results part were as follows:

Page 9: Gene profiles of patients with LUAD from TCGA were analyzed. The tumors with mutated *LKB1* displayed lower levels of ICAM1 than those with WT *LKB1* (Fig. 2c), regardless of *KRAS* co-mutation and *LKB1* mutation types (Supplementary Fig. 7A-B). We then assessed the association between *LKB1* and ICAM1 in clinical samples by IHC. Decreased ICAM1 was observed in more than 80% of *LKB1*-mutated tumor samples examined, whereas 80% of *LKB1*-WT specimens showed intensified ICAM1 staining (Fig. 2d). Together, these data confirmed that *LKB1* deficiency is associated with suppression of ICAM1 in lung cancer. To demonstrate this finding further, we examined ICAM1 expression across a panel of *LKB1* mutant versus *LKB1* wildtype lung cancer cell lines. ICAM1 mRNA and protein levels were either significantly downregulated or completely undetectable in *LKB1* mutant cell lines, regardless of *KRAS* co-mutation (Fig. 2e-f). We subsequently investigated the causal relationship between *LKB1* and ICAM1. Small interference RNA or short hairpin RNA mediated knocking down of *LKB1* decreased ICAM1 expression in vitro and in vivo (Supplementary Fig. 7C-D). Finally, *LKB1* reconstitution in ICAM1^{lo} KL cell lines rescued ICAM1 expression, which also required *LKB1* kinase activity (Fig. 2g, h and Supplementary Fig. 7E-G). Therefore, *LKB1* positively regulates ICAM1 expression.

4. The impact of Icam-1 expression on IO response should be verified in additional

isogenic systems and not just in the context of exogenous overexpression. How does endogenous ICAM-1 expression in LLC-1 compare with the LLC1-shLkb1-ICAM-1WT? The authors should knock out *Icam1* in the parental LLC-1 and assess response to immunotherapy, and ideally in at least a second IO-responsive system even if not of lung origin (such as the CT26 syngeneic model in BALB/c).

Response: Thank you so much for your professional suggestion. We deeply agree with you that *Icam1* expression on IO response should be demonstrated in additional isogenic systems. Accordingly, we firstly compared endogenous ICAM1 expression in LLC1-shcontrol cells with LLC1-sh*Lkb1* cells (the latter with ICAM1 expression). And the data showed that endogenous ICAM1 expression reduced in LLC1-sh*Lkb1* cells in compared to LLC1-shcontrol cells and parental LLC1 (Supporting Figure 8A).

Besides, we also utilized CRISPR/cas9-mediated *Icam1* knockout in parental LLC-1 cells and MC38 cells (MC38 syngeneic model in C57, an IO-responsive system). Knock out of ICAM1 was validated in both LLC1 and MC38 cell lines (Supporting Figure 8B-C). We took advantage of these two experimental models to test the hypothesis that cancer cell-intrinsic knock-out of *Icam1* reduces tumors responsiveness to ICB. We observed no significant difference in mice with *Icam1*^{KO} LLC1 between vehicle and ICB group. They were poorly responsive to ICB than mice with *Icam1*^{WT} LLC1, and their tumor growth was faster after immunotherapy (Supporting Figure 8D). Mice with *Icam1*^{WT} MC38 tumors were sensitive to PD-1 blockade therapy, whereas mice bearing *Icam1*^{KO} tumors had faster tumor growth when treated with anti-PD-1 antibody (Supporting Figure 8E). These results demonstrate that tumor-intrinsic ICAM1 expression may be a powerful responsive mechanism to ICB.

Supporting Figure 8: Tumor-intrinsic ICAM1 expression may be a powerful responsive mechanism to immune checkpoint blockade (ICB) therapy. (A-C) Western blotting of ICAM1 with LKB1 expression of murine tumor cell lines (LLC1 and MC38) transfected with shLkb1, or transfected with sgCon or sg*lcam1* sgRNA. (D-E) C57BL/6 mice were implanted subcutaneously with LLC1 or MC38 cells with optional knockout of *lcam1* and treated with or without PD-1 mAb. Macroscopic appearance of tumors (left) and plots of tumor volume (right) for LLC1 (D) and MC38 (E) xenografts. (Results are presented as mean \pm SEM. A mixed-effects model followed by Tukey's multiple comparisons was performed to compare the tumour growth curves in different treatment groups. n.s., not significant; * $p < .05$; ** $p < .01$; *** $p < .001$; **** $p < .0001$)

Corresponding changes and explanations we have added in the revised manuscript in the results part were as follows:

Page 15: However, cancer cell-intrinsic knock-out of *lcam1* reduces tumors responsiveness to immunotherapy (Supplementary Fig. 13A-D). Taken together, these data illustrated that ICAM1 overexpression sensitized *LKB1* deficient lung tumors to anti-PD-1 immunotherapy.

5. Are the differences in ICAM expression in the human tumors (for example Figure 2G) observed in tumor cells, immune cells or both? This should be clarified. Furthermore, does treatment with palbociclib induce ICAM-1 expression also in immune cells? This should be easy to assess using co-IF in the syngeneic and GEMM models.

Response: Thank you so much for your professional suggestion. We really need to describe the expression characteristics of ICAM1 clearly. According to our pathologist in our center, the differences of ICAM1 were observed only in tumor cells, which characterized as both positive on the membrane and cytoplasm (Supporting Figure 7). In addition, patients with stronger ICAM1 expression on tumor cells had more CD8+ T cells infiltration (Supporting Figure 7).

Besides, we have assessed if treatment with palbociclib induce ICAM-1 expression also in immune cells using co-IF in our syngeneic and GEMM models. And we found that ICAM1 and CD8 co-expression was scarcely observed in both GEMM models and orthotopic lung cancer mice models (Supporting Figure 9A), which indicated that palbociclib mainly induced ICAM1 expression in cancer cells. To furtherly demonstrate this effect, we evaluated treatment-induced ICAM1 expression on CD8+ T cells, we found that CDK4/6 inhibition and combination therapy increased abundance of CD8+ T cells as expected, but no significant changes of ICAM1 were observed on CD8+ T cells among these treatment groups, which further precluded that palbociclib induce ICAM-1 expression also in immune cells (Supporting Figure 9B).

Supporting Figure 9: Palbociclib did not induce ICAM-1 expression in immune cells. (A) Immunofluorescence (IF) analysis of ICAM1+cells (green), CD8+ T cells (red) in resected KL GEMM samples (right) and orthotopic mouse models (left) following different arms of treatment. Scale bar, 100 µm. (B) Flowcytometry analysis of infiltration of CD8+ T cells and their ICAM1 membrane expression in LLC1-sh*Lkb1* tumors with different treatments. (Results are presented as mean ± SEM. One-way ANOVA was performed for multi-group comparisons. n.s., not significant; **p*<.05; ***p*<.01; ****p*<.001; *****p*<.0001) Corresponding changes and explanations we have added in the revised manuscript in the results part were as follows:

Page 25: Moreover, the combination treatment led to a significantly increased expression level of ICAM1 on tumor cells, which was also observed in the palbociclib group (Fig. 7f), but palbociclib did not induce ICAM-1 expression on immune cells (Supplementary Fig.19D-E).

6. It would be important to present more comprehensive and quantified immune profiling data from the GEMM K and KL models as well as changes in TME in response to treatment. Although the effects on CD8+ TILs observed in the GEM KL models are in keeping with previous reports, the authors don't observe any changes in myeloid cells populations and in particular, tumor associated neutrophils, that have consistently been reported to be increased in the context of LKB1 loss in both GEMMs and syngeneic model of KRAS-mutant NSCLC. It is not clear what accounts for this discrepancy. Mosaic LKB1 loss might play a role, I note that LKB1 loss is partial in WBs from tumors. **Did the scRNA analysis confirm reduced LKB1 expression in the model?** Additional confirmation of LKB1 loss by IHC and its relationship to ICAM1 expression should be presented. Were compared tumors of comparable size? In addition, treatment-induced changes in myeloid cell subpopulations are not adequately examined despite previously published work that palbociclib can impact myeloid cell subsets. Finally, more detailed analysis of the **functional effects** of ICAM-1 OE on the TME in the LLC1-sh*Lkb1* system is required (Figure 3) is required analogous to the analysis of the impact of palbociclib treatment in later figures. Absolute (as % of CD45+ cells) and relative fractions of immune cells subsets should be presented, including effects on myeloid cells and more detailed characterization of proliferation, activation and differentiation status of TILs.

Response: Thank you so much for your professional and intensive suggestion. Indeed, more comprehensive and quantified immune profiling data from the GEMM K and KL models are needed.1) To gain deep insight into the broad impact of *Lkb1* mutation on T cell phenotype and the landscape of the tumor immune microenvironment in vivo, lung tumors confirmed by micro-CT in *Kras*^{mutant}/*Lkb1*^{mutant} (KL) versus *Kras*^{mutant} (K) mice were sent for single-cell RNA sequencing (scRNA-seq) analyses. In terms of canonical feature cell markers, eleven major cell clusters were discovered, and cell annotations were subsequently calibrated by evaluating the most highly expressed marker genes among each cell clusters (Supporting Figure 10A). Cell composition analysis showed reduced activated T cells and increased exhausted T cells and B cells in tumors from KL mice compared with those from K mice (Supporting Figure 10B-C). To provide more detailed quantitative analysis of the immune profiling data and to evaluate the general effect of *Lkb1* deficiency on different immune cell clusters, we also calculated gene signature-based scores at the single-cell level in different cell types using single sample gene set enrichment analysis (ssGSEA) (nature, 2009). And we found that activated T cells isolated from KL tumors showed significantly lower “T cells homing on tumor” and “T cell proliferation in immune response” signature scores (Supporting Figure 11A); While exhausted T cells isolated from KL tumors showed significantly higher “PD-1 pathway”, “T cell anergy” signature scores compared with T cells from K tumors (Supporting Figure 11B), indicating lower T cell infiltration and activation, and higher T cell dysfunction by *Lkb1* deficiency. However, no significant difference in signature scores was observed among the other cell clusters, including B cells (Supporting Figure 11C-F). These data suggest that T-cell activation status is the main determinant downstream signaling of LKB1 in dictating the tumor immune microenvironment.

Supporting Figure 10: Quantified cell composition analysis of single cell sequencing data in *Kras*^{mutant}/*Lkb1*^{mutant} (KL) versus *Kras*^{mutant} (K) mice. (A) Heatmap of canonical cell-type markers of 11 major cell types. (B) Absolute numbers of cells from K and KL samples for clusters identified in Figure 1. (C) Expression level of *Tcf7* in activated T cells and that of *Ctla4* in exhausted T cells from lung tumor samples of K and KL mouse. Statistical significance was tested with a two-tailed Mann–Whitney U test. ns, not significant; **p*<.05; ***p*<.01; ****p*<.001; *****p*<.0001.

Supporting Figure 11: Gene set variation analysis with immune-related signatures for cells in the (A) activated T cells cluster, (B) exhausted T cells cluster, (C) neutrophils cluster, (D) macrophages cluster, (E) B cells cluster and (F) DCs cluster. (Data were presented as violin plot (median, 25%-75%, range). Statistical significance was tested with a two-tailed Mann–Whitney U test. ns, not significant; * $p < .05$; ** $p < .01$; *** $p < .001$; **** $p < .0001$)

Corresponding changes and explanations we have added in the revised manuscript in the results part were as follows:

Page 5-6: After quality control and cell filtering, we catalogued 14,260 cells into eleven distinct cell lineages annotated with canonical feature cell markers, thus identifying cancer cell types 1, cancer cell types 2, endothelial cells, and immune cells types (including activated T cells, exhausted T cells, B cells, neutrophils, macrophages cells, NK cells, dendritic cells and plasmacytoid dendritic cells) (Fig. 1d, e, f). Cell composition analysis showed reduced activated T cells and increased exhausted T cells in tumors from KL mice compared with those from K mice (Fig. 1f, Supplementary Fig. 1D). Activated T cells in KL group were characterized by significantly lower expression of *Tcf7* relative to K group ($p < 0.001$, Fig. 1g, Supplementary Fig. 1E); while exhausted T cells displayed significantly higher expression of *Ctla4* in KL mice versus those of K mice ($p < 0.0001$, Fig. 1g, Supplementary Fig. 1E). To further evaluate the general effect of *LKB1* deficiency on different immune cell clusters, we also calculated gene signature-based scores at the

single-cell level in different cell types using single sample gene set enrichment analysis (ssGSEA) 22. Cancer cells in KL group showed significantly higher LKB1 loss signature scores 23, confirming LKB1 deficiency in KL mice (Supplementary Fig. 1F). And we found that activated T cells isolated from KL tumors showed significantly lower “T cells homing on tumor” signature scores; while exhausted T cells isolated from KL tumors showed significantly higher “PD-1 pathway”, “T cell anergy” signature scores compared with T cells from K tumors (Supplementary Fig. 2A, B). However, no significant difference in signature scores was observed among the other cell clusters, including B cells (Supplementary fig. 2C-F). These data suggest that T-cell activation status is the main determinant downstream signaling of LKB1 in dictating the tumor immune microenvironment.

2) For the reason why, there is discrepancy of tumor associated neutrophils, which have consistently been reported to be increased in the context of LKB1 loss in both GEMMs and syngeneic model, we carefully analyzed this problem. And we found that maybe this is due to the particularity in single cell sequencing data. It was reported that, analysis of neutrophils is challenging in single cell transcriptomic data because of their low RNA content and relatively high levels of RNase. Besides, neutrophils are sensitive to degradation after collection, and the protocol we used for sample collection may be more suitable to preserve both tumor cells and all immune cells as much as possible. Therefore, some neutrophils may be lost. This may in some way lead to the discrepancy.

Besides, we also used syngeneic mice injected with LLC1-sh*Lkb1* or LLC1-sh*control* cells, and conducted flow cytometry of the tumor tissue to assess the impact of *Lkb1* alteration on MDSCs in orthotopic lung cancer model. Consistent with previous observations, we observed higher infiltration of MDSCs in mice harbouring LLC1-sh*Lkb1* tumors than in control mice. (Supporting Figure 12A).

3) Furthermore, we agree with you that one limitation is that the model exhibited mosaic and non-uniform deletion of *Lkb1* (as it is based on infection with a PSECCsg*Lkb1* vector) in different subpopulations of the cancer cells. We have presented confirmation of LKB1 loss by IHC and CO-IF in GEMMs models (Supporting Figure 6F-G). They were compared with tumors of comparable size (micro-CT, figure 1A-B). In addition, we also analyzed the scRNA data to confirm the deficiency level of LKB1. Knock out of gene *Lkb1* should not be evaluated simply through mRNA expression level of cancer cell subpopulations in scRNA data, therefore we are trying to quantify the functional deficiency level of LKB1 through conditional knockout of this gene. LKB1 deficiency was reported to be associated with a consistent pattern of gene expression and a predictive 16-gene signature was thus derived from this pattern, which can discriminate LKB1-deficient tumors from those with functional LKB1 (Jacob M, 2014, JTO). And we therefore exploited this gene signature and calculated the signature score. We found that cancer cells in KL group showed significantly higher LKB1 loss signature scores (Supporting Figure 12B), accurately confirming LKB1 deficiency in KL mice.

Supporting Figure 12: (A) Flow cytometry analysis of infiltration of MDSCs (CD45+ CD11b+ Gr1+) in LLC1-sh*Lkb1* lung tumors versus that LLC1-shcontrol mice. (B) Gene set variation analysis with LKB1 loss gene set for cells in cancer_cells1 and cancer_cells2 cluster in K and KL samples. (Results were presented as mean \pm SEM. Student's t test or two-tailed Mann–Whitney U test were used to analyze the data. ** $p < .01$; **** $p < .0001$).

4) In addition, according to your valuable suggestion, we analyzed treatment-induced changes in myeloid cell subpopulations adequately (dendritic cells, myeloid-derived suppressor cells, tumor associated macrophages and neutrophils) in our orthotopic models. And we found that abundance of MDSCs and tumor associated neutrophils (TANs) were significantly reduced after CDK4/6 inhibition treatment, no significant difference was observed in frequency of tumor associated macrophages within the tumor environment among these three treatment groups (Supporting Figure 13A). One possible explanation may be that CDK4/6 inhibition leads to impaired proliferation of bone marrow hematopoietic progenitors (Deng, 2018, cancer discovery). They also demonstrated that the anti-proliferative effect of CDK4/6 inhibition does not result in a decrease of TILs, but does lead to a reduced abundance of the myeloid subpopulation, which was an excellent work.

In addition to tumor infiltration analysis of myeloid cell subpopulations, we also exploited our mRNA sequencing data of tumor tissue in these four treatment arms, to investigate into the functional status of treatment-induced changes in these myeloid cell subtypes. We estimated GSVA scores for gene sets related to the function of myeloid immune cells from the nCounter PanCancer Immune Profiling Panel and the Molecular Signatures Database. It was found that the biological process of dendritic cell antigen processing and presentation was conceivably enhanced in the palbociclib arm and the combination group (Supporting Figure 13B). We also observed mildly increasing of DCs in the Palbociclib group but significantly enhanced level in the combination treatment group (Supporting Figure 13A).

Collectively, our data suggested that in lung tumor model with LKB1 deficiency, CDK4/6 inhibition reduced abundance of MDSCs and TANs in tumor tissue, and the combined CDK4/6 inhibitors and PD-1 blockade therapy may impact dendritic cell infiltration and functioning in the process of antigen processing and presentation.

Supporting Figure 13: Treatment induced changes of CDK4/6 inhibition and the combination therapy on myeloid subpopulations. Combined CDK4/6 inhibitors and PD-1 blockade therapy may affect myeloid cells and CD8+ T cells as well. (A) Flowcytometry analysis of infiltration of DCs (CD45+ CD11b+ CD11c+ CD103+), MDSCs (CD45+ CD11b+ Gr1+), TAN (CD45+ CD11b+ CXCR2+), and TAMs (CD45+ CD11b+ F4/80+) in LLC1-shLkb1 tumors with different treatments. (B) Gene set variation analysis of mRNA sequencing data of tumor tissue in four treatment arms with gene sets related to the function of myeloid immune cells from the nCounter PanCancer Immune Profiling Panel and the Molecular Signatures Database. (Results were presented as mean \pm SEM. One-way ANOVA was used to analyze the data. **** $p < 0.0001$).

Corresponding changes and explanations we have added in the revised manuscript in the results part were as follows:

We also analyzed treatment-induced changes in myeloid cell subpopulations. CDK4/6 inhibition reduced abundance of MDSCs and TANs in tumor tissue, and the combined CDK4/6 inhibitors and PD-1 blockade therapy may impact dendritic cell infiltration and functioning in the process of antigen processing and presentation (supplementary Fig. 19B-C).

6) Finally, we truly agree with you that, more detailed analysis of the functional effects of ICAM-1 overexpression on the TME in the LLC1-shLkb1 system is required. We therefore analyzed absolute and relative fractions of immune cells subsets, including effects on myeloid cells. We also have provided more detailed characterization of proliferation, activation and differentiation status of TILs. It was observed that ICAM1 overexpression significantly increased infiltration level of CD8+ T cells, NK cells, and dendritic cells. Besides, expression level of CD69 (T cell activation) in CD8+ T cells was significantly increased in LKB1 deficient group with overexpression of ICAM1. For the differentiation status of CD8+ T cells (effector memory CD8+ T cells, and central memory CD8+ T cells),

no significant changes were observed after ICAM1 overexpression, though an enhancing tendency was observed in effector memory CD8+ T cells when ICAM1 was augmented. Besides, ICAM1 overexpression exerted minor effects on myeloid cells, only DCs significantly increased, while no differences were observed in TAMs and MDSCs (Supporting Figure 14).

Supporting Figure 14: Detailed analysis of the functional effects of ICAM-1 overexpression on the TME in the LLC1-sh*Lkb1* system. Total CD8+ T cells relative to CD45+ cells in tumor tissue, relative MFI of CD69 and CD44 on CD8+ T cells, percentage of CD62L+CD44+ (central memory T cells) and CD62L-CD44+(effector memory T cells) in CD8+ T cells, and total NK cells relative to CD45+ cells in tumor tissue and relative MFI of CD69 on NK cells, percent of CD45+CD11b+CD103+CD11c cells (DCs), CD45+CD11b+Gr-1+ cells (MDSCs), CD45+CD11b+F4/80+ cells (TAMs) from mice bearing LLC1 tumors (LLC1-control-*Icam1*^{WT}, LLC1-sh*Lkb1*-*Icam1*^{WT}, LLC1-sh*Lkb1*-*Icam1*^{OE}) were analyzed by flow cytometry. (Results were presented as mean ± SEM. One-way ANOVA was used to analyze the data. ns, not significant; **p*<.05; ***p*<.01; ****p*<.001; *****p*<.0001)

Corresponding changes and explanations we have added in the revised manuscript in the results part were as follows:

We firstly investigated the functional effects of ICAM-1 reinforcement on the TME in the LLC1-sh*Lkb1* orthotopic lung tumor system. ICAM1 overexpression significantly increased CD8+ T cell and NK cell infiltration (Fig. 4a). Besides, expression level of CD69 (T cell activation) on CD8+ T cells was significantly increased, and CD44 expression exerted an enhancing tendency. For the differentiation status of CD8+ T cells (effector memory CD8+ T cells, and central memory CD8+ T cells), no significant changes were observed, though an enhancing tendency was displayed in effector memory CD8+ T cells when ICAM1 was augmented (Fig.4a). Taken together, LKB1 deficiency leads to rarity of immune cytotoxic cells, and ICAM1 can enhance infiltration and activation of CD8+ T cells and NK cells.

7. Based on the author's hypothesis, LKB1 competes with CDK4 for RB binding. However, the effects of LKB1 on RB phosphorylation and ICAM-1 expression appear to be kinase activity-dependent. How would LKB1 kinase activity impact LKB1-RB binding in this model? Does exogenous kinase-deficient LKB1 interact with CDK4 and RB? Does overexpression of kinase deficient LKB1 affect the CDK4-RB interaction? These questions should be addressed.

Response: Thank you so much for your professional suggestion. We performed co-IP experiment using exogenous WT LKB1 and exogenous kinase-dead LKB1 in A549 cells to explore how LKB1 interact with CDK4 and RB, and we found that overexpression of kinase-dead *LKB1* (*LKB1*-MUT) displayed an impairment in RB binding. We further validated the interaction between RB and CDK4 in wildtype *LKB1* versus kinase-dead *LKB1* model, and found that wildtype LKB1 disturbed CDK4-RB interaction (Supporting Figure 15) which indicated LKB1 may compete with CDK4 for RB binding, while CDK4-RB interaction recovered in *LKB1*-MUT group, we infer that this may result from the impairment in LKB1-RB interaction.

Supporting Figure 15: Overexpression of kinase deficient LKB1 did not affect the CDK4-RB interaction. Corresponding changes and explanations we have added in the revised manuscript in the results part were as follows:

Page 19: Subsequent IP experiments supported this hypothesis; specifically, LKB1 overexpression interfered with the interaction of CDK4 with RB, and vice versa (Supplementary Fig. 15H). However, CDK4-RB interaction recovered in LKB1-MUT group (Supplementary Fig. 15I).

8. Despite inclusion of several experiments the signaling axis linking LKB1 loss with reduced ICAM-1 expression is not clear. The authors initially posit that RB/E2F activity is involved and include a series of experiments but then also propose that NF- κ B signaling may be media. However, these latter experiments are limited. Is there increased NF κ B occupancy in the promoter of ICAM-1 in LKB1 deficient compared to proficient cells? Does KD of p65 affect expression of ICAM-1?

Response: Thank you so much for your pertinent and professional comments. We indeed agree with you that the underlying mechanism needs to be improved. According to your valuable suggestion, we firstly evaluated how LKB1 regulate ICAM1 expression through

p65. Both qPCR and flowcytometry analysis demonstrated that LKB1 overexpression significantly restored ICAM1 expression, while knockdown p65 remarkably impaired ICAM1 in LKB1 overexpressed cells (Supporting Figure 16A-B). Further, according to your pertinent suggestion, we performed ChIP analysis and found that there is increased p65 occupancy in the promoter of ICAM1 in LKB1-WT group compared to LKB1 loss group (A549-LKB1-WT vs. A549-LKB1-loss) (Supporting Figure 16C), which furtherly demonstrated that LKB1 regulated the transcriptional activity of p65 for the expression of ICAM1.

Supporting Figure 16: LKB1 regulates the transcriptional activity of p65 for the expression of ICAM1. (A-B) Analysis of ICAM1 expression of LKB1-loss cells expressing vector+siNC, LKB1+siNC, or LKB1+siP65. qPCR (A) and flowcytometry (B) analysis of ICAM1 expression. (C) ChIP assay was performed with cell lysates from A549/LV-control and A549/LV-LKB1 cells. A pair of primers flanking the p65 binding site within the ICAM-1 promoter was used in PCR. Real-time PCR was employed to the ChIP assay. (Results were presented as mean \pm SEM. One-way ANOVA or student's t test was used to analyze the data. ** $p < .01$; **** $p < .0001$)

Corresponding changes and explanations we have added in the revised manuscript in the results part were as follows:

Page 19: LKB1 deficiency led to a decreased phosphorylation level of p65 (Supplementary Fig.15E). ChIP analysis demonstrated that p65 occupancy in the promoter of ICAM1 is increased in LKB1-WT group compared to LKB1 loss group (Fig. 5j), which furtherly demonstrated that LKB1 regulates the transcriptional activity of p65 for the expression of ICAM1. In agreement with this, p65 knockdown impaired ICAM1 expression in LKB1 restored group (Supplementary Figure 16G-H).

9.It is critical to prove the extent to which the increased sensitivity to palbociclib and a-PD-1 is due to upregulation of ICAM-1 (since palbociclib can have pleotropic effects). The authors should establish isogenic derivatives of the LLC1-shLkb1 cell line with or without knockout of ICAM-1 using CRISPR/Cas9 and repeat the therapeutic experiment. The experiment shown in Figure 7EF does not adequately address this, since anti-ICAM-1 may also target ICAM-1 in non-tumor cells. The effects of treatment with palbociclib on the tumor microenvironment in the setting of ICAM-1 depletion should also be evaluated (FACS, IHC).

Response: Thank you so much for your professional suggestion. We totally agree with you that increased sensitivity to Palbociclib and PD-1 blockade depending on upregulation of ICAM1 should be demonstrated, which is one of the critical questions of our manuscript.

Therefore, we have established isogenic derivatives of the LLC1-sh*Lkb1* cell line with or without knockout of *Icam1* using CRISPR/Cas9 (Supporting Figure 8A) and repeated the therapeutic experiment. And we found that antitumor effects of the combined PD-1 blockade and CDK4/6 inhibition were significantly impaired in LLC1-sh*Lkb1*-sg*Icam1* tumors (Supporting Figure 17A).

Accordingly, we systemically evaluated the effects of treatment with palbociclib or the combination therapy on the tumor microenvironment in the setting of ICAM-1 depletion (LLC1-sh*Lkb1*-sg*Icam1* vs. LLC1-sh*Lkb1*-sgCon). IHC staining of tumor tissue displayed reduced ICAM1 expression and CD8+ T cell infiltration in LLC1-sh*Lkb1*-sg*Icam1* tumors compared with LLC1-sh*Lkb1*-sgCon tumors after Palbociclib treatment or combination therapy (Supporting Figure 17B). Flow cytometry analysis demonstrated that LLC1-sh*Lkb1*-sg*Icam1* tumors exhibited a significant decrease in CD8+ T cell infiltrate, and CD69, CD44 expression on CD8+ T cells versus LLC1-sh*Lkb1*-sgCon tumors after CDK4/6 inhibition or combination therapy. Furthermore, infiltration of NK cells and DCs were significantly lower in LLC1-sh*Lkb1*-sg*Icam1* tumors after combination therapy versus that of LLC1-sh*Lkb1*-sgCon tumors. Interestingly, MDSCs and tumor-associated neutrophils (TAN) infiltration were significantly increased in the setting of ICAM-1 depletion after the treatment (Supporting Figure 17C). Taken together, CDK4/6 inhibitors or the combination therapy induced an immune-active tumor microenvironments, which is dependent on ICAM1.

Supporting Figure 17: ICAM-1 depletion abrogates anti-tumor immunity conferred by the Palbociclib/ICI combination therapy. (A) C57BL/6 mice were inoculated with LLC1/sh*Lkb1* with or without *Icam1* knock-

out cells and received the combination treatments versus vehicle. Tumor size were monitored. (B) Representative immunohistochemistry images of ICAM1 and CD8 α in LLC1-sh*Lkb1*/sgCon and LLC1-sh*Lkb1*/sg*lcam1* tumors subjected to indicated treatment. Scale bar, 50 μ m. (C) Flowcytometry analysis of immune cell infiltration (percent of CD45+CD3+CD8+ cells, CD45+CD3-NK1.1+ cells, CD45+CD11b+CD11c+CD103+ cells, CD45+CD11b+Gr-1+ cells, CD45+CD11b+CXCR2+ cells, CD45+CD11b+F4/80+ cells), expression of CD69, CD44 on CD8+ T cells and CD107a on NK cells in LLC1-sh*Lkb1*/sgCon and LLC1-sh*Lkb1*/sg*lcam1* tumors followed by indicated treatment arm. (Results were presented as mean \pm SEM. One-way ANOVA was performed for multi-group comparisons. * p <.05; ** p <.01; *** p <.001; **** p <.0001)

Corresponding changes and explanations we have added in the revised manuscript in the results part were as follows:

Page 28: Subsequently, isogenic derivatives of the LLC1-sh*Lkb1* cell line with or without knockout of *lcam1* using CRISPR/Cas9 were established. The mice received the same combinational therapy. And we found that antitumor effects of the combined PD-1 blockade and CDK4/6 inhibition were significantly impaired in LLC1-sh*Lkb1*-sg*lcam1* tumors (Fig. 8g).

Accordingly, we systemically evaluated the effects of treatment with palbociclib or the combination therapy on the tumor microenvironment in the setting of ICAM-1 depletion. IHC staining of tumor tissue displayed reduced ICAM1 expression and CD8+ T cell infiltration in LLC1-sh*Lkb1*-sg*lcam1* tumors compared with LLC1-sh*Lkb1*-sgCon tumors after palbociclib treatment or combination therapy (Fig. 8h, supplementary Fig. 21A). Flow cytometry analysis demonstrated that LLC1-sh*Lkb1*-sg*lcam1* tumors exhibited a significant decrease in CD8+ T cell infiltrate, and CD69, CD44 expression on CD8+ T cells versus LLC1-sh*Lkb1*-sgCon tumors after CDK4/6 inhibition or combination therapy. Furthermore, infiltration of NK cells and DCs were significantly lower in LLC1-sh*Lkb1*-sg*lcam1* tumors after combination therapy versus that of LLC1-sh*Lkb1*-sgCon tumors. Taken together, these depletion therapies reversed the advantages conferred by the palbociclib/ICI combination therapy, confirming that CDK4/6 inhibition promotes the effect of anti-PD-1 immunotherapy in an ICAM1-dependent manner (Fig. 8h, supplementary Fig.21A-B).

10. The degree of selectivity of the palbociclib/anti-PD-1 approach for LKB1-mutant/deficient NSCLC should be assessed by performing the same experiment in the parental cell line. To what extent is this cell line uniquely sensitive to the palbociclib/anti-PD-1 combination. To what extent are the immune-modulatory effects dependent on *Lkb1* status?

Response: Thank you so much for your valuable and professional suggestion. According to your comments, we have performed the same experiments in the parental cell line (LLC1-*Lkb1*^{WT}) and LLC1 with LKB1 overexpression (LLC1-*Lkb1*^{OE}), to fully demonstrate *Lkb1* status on the immune-modulatory effects. And we found that combined PD-1 blockade and CDK4/6 inhibition significantly slowed tumor growth in both the LLC1-*Lkb1*^{WT} and LLC1-*Lkb1*^{OE} group (Supporting Figure 18A-B). However, the synergistic effects of the combination therapy were diluted in comparison with LLC1-*Lkb1*^{KD} group (LLC1-sh*Lkb1*), especially for mice with LKB1 overexpression (Supporting Figure 18B), anti-PD-1 mAb treatment alone already exhibited significant antitumor effect against LLC1-*Lkb1*^{OE} tumors,

implying that LKB1-mutant/deficient NSCLC is uniquely sensitive to the synergistic effect of Palbociclib and anti-PD-1 combination therapy.

Supporting Figure 18: The synergistic effects of the combination therapy were diluted in both the LLC1-*Lkb1*^{WT} and LLC1-*Lkb1*^{OE} group, compared to LLC1-*Lkb1*^{KD} group (LLC1-sh*Lkb1*). (A-B) C57BL/6 mice were implanted subcutaneously with LLC1 cell lines with wild-type *Lkb1* (A) or overexpressed *Lkb1* (B) and treated with isotype IgG, anti-PD-1 Ab, Palbociclib or combination of both anti-PD-1 Ab and Palbociclib. Macroscopic appearance of tumors (left), plots of tumor volume (middle) and quantification of tumor weight (right) were shown.

(Results are presented as mean \pm SEM. A mixed-effects model followed by Tukey's multiple comparisons was performed to compare the tumour growth curves in different treatment groups, and one-way ANOVA was performed for multi-group comparisons. n.s., not significant; * p <.05; *** p <.001)

We furthermore addressed to what extent are the immune-modulatory effect dependent on LKB1 status. RNA-sequencing of murine LLC1-*Lkb1*^{WT} tumor specimens in four different treatment arms were also performed and analyzed in comparison with that of murine LLC1-*Lkb1*^{KD} groups. Pathway enrichment analysis indicated that, several important immune modulatory process (interferon alpha/gama response, adhesion pocess, etc.) were downregulated in LKB1 deficient tumors (LLC1-sh*Lkb1* tumors vs. LLC1-shcontrol tumors), which was correlated with the "immune desert" feature of LKB1 deficient tumors (Supporting Figure 19A). We subsequently compared upregulated immune modulatory process in LKB1 deficient groups receiving combination treatment, with that of LKB1 proficient groups. Interferon alpha/gama response, adhesion pocess and antigen processing/presentation were upregulated (LLC1-sh*Lkb1* tumors vs. LLC1-shcontrol tumors receiving the combination therapy) (Supporting Figure 19B). Taken together, immune activation and modulatory effects were impaired in LKB1 deficient groups, and they were highly recovered from the combination therapy, which was dependent on LKB1 status.

Supporting Figure 19: Immune activation and modulatory effects were impaired in LKB1 deficient groups, and they were highly recovered from the combination therapy. A. Enrichment analysis of downregulated pathways in LKB1 deficient tumors (LLC1-shLkb1 tumors vs. LLC1-shcontrol tumors). B. Enrichment analysis of upregulated pathways in LKB1 deficient tumors after receiving combination therapy (LLC1-shLkb1 tumors vs. LLC1-shcontrol tumors in the combination therapy group).

Corresponding changes and explanations we have added in the revised manuscript in the results part were as follows:

Page 22: The degree of selectivity of the palbociclib/anti-PD-1 approach for LKB1-deficient NSCLC was also assessed. Compared to LKB1 wildtype or overexpressed tumors (Supplementary Fig. 18C-D), LKB1 deficient NSCLC is uniquely sensitive to the synergistic effect of the combination therapy.

11. There are some methodological concerns regarding the in vivo preclinical experiments in Fig 5:

- Treatment appears to start very early in the LLC1-shLkb1 model in Figure 5A. The experiment should be performed with randomization to treatment arms occurring when tumors reach adequate volume (for example 300mm³).

Response: Thank you so much for your pertinent and professional comments. We totally agree with your suggestions. Our treatment was initiated in the LLC1-shLkb1 model on day 7 (tumor was palpable) after tumor cell implantation subcutaneously, and the mice were randomized to different treatment arms, more details should be provided. Accordingly, we also have updated the tumor volume growth curve starting from treatment initiation on day 7 in Fig.4d and Fig. 6a in the revised manuscript. We chose this time point based on studies

using LLC1 cell line as syngeneic mouse model receiving immunotherapy (MC38, CT26 cell lines as well) (Yan Lan, 2021, cancer cell; Antonio Marzio, 2022, cell; Deng, Cancer Discovery 2018). They adopted the timepoint (7 days after tumor implantation) and start randomization and treatment very early, when tumor was palpable. Therefore, we performed our experiments according to their methods.

Corresponding changes and explanations we have added in the revised manuscript in the results part were as follows:

We have updated the tumor volume growth curve starting from treatment initiation on day 7 in Fig.4d and Fig. 6a in the revised manuscript.

•In Figure 5D, treatment bulk at the time of treatment initiation appears to differ between the groups, with mice in the combo arm apparently having the lowest tumor bulk. Although the authors measure the relative change in photon flux after one cycle of treatment compared to baseline clearly efficacy can be affected by the amount of tumor present at baseline.

Response: Thank you so much for your pertinent suggestion. According to your comments, we repeated this experiment, to ensure the consistency of the treatment bulk at the time of therapeutic initiation among the four different groups, treatment randomization was strictly performed. And the results showed that, significant tumor growth inhibition was induced by co-inhibition of CDK4/6 and PD-1 in orthotopic LLC1 lung tumor models. Representative images and statistical results were presented (Supporting Figure 20A-B).

Supporting Figure 20: Significant tumor growth inhibition was induced by co-inhibition of CDK4/6 and PD-1 in orthotopic LLC1 lung tumor models. (A-B) LLC1-sh*Lkb1*-luc cell line was constructed and injected into the left chest of mice. Tumor formation was detected. Mice harboring lung tumors were administered different treatments (control immunoglobulin G (Vehicle), palbociclib or anti-PD-1 Ab, or co-treatment with palbociclib and anti-PD-1 Ab). Representative bioluminescent images (A) and quantification of results (B). (Results were presented as mean \pm SEM. One-way ANOVA was performed for multi-group comparisons. n.s., not significant; * $p < .05$)

Corresponding changes and explanations we have added in the revised manuscript in the results part were as follows:

We have updated Fig.6d and Fig.6e in the revised manuscript.

•The change in tumor volume in Figure 5F in the GEMM preclinical experiment should be quantified and formally compared between the different groups. It is not clear from the

methods section when treatment started and whether mice had comparable tumor volume/bulk at the outset. It is not clear from the presented data whether the combination is effective in the autochthonous GEMM model. One additional limitation is that the model will likely exhibit mosaic and non-uniform deletion of *Lkb1* (as it is based on infection with a PSECCsgLkb1 vector).

Response: Thank you for your kind suggestion. We really need to add more details in our methods. After nasal inhalation, tumor formation was analyzed with a micro-CT or MRI scanner after 12 weeks of virus infection. Once we had observed gasping from lung tumor-burdened mice and monitored lung nodules, then these mice were randomized into various study cohorts. These enrolled mice had comparable tumor volume/bulk at the outset, and tumor response was evaluated by assessing changes in lung tumor bulk and compared with tumors at baseline using micro-CT or MRI imaging every two weeks (Supporting Figure 21A). According to your pertinent comments, we repeated the GEMM preclinical experiment to compare quantifiably and formally among the different groups. MRI volumetric measurements were based on previous references (Deng et al, Cancer Discovery 2018). We observed that co-inhibition of CDK4/6 and PD-1 displayed significantly enhanced tumor control than the vehicle group in our genetically engineered *Kras*-driven mouse model with conditional deletion of *Lkb1* (KL GEMM) (Supporting Figure 21B).

Supporting Figure 21: CDK4/6 inhibitor and PD-1 blockade therapy elicits significantly tumor control in KL GEMM mouse model. (A) Representative MRI scan images show mice lung tumors before and after the treatment. (B) Quantification of tumor volume changes by MRI or micro-CT scan after treatment with vehicle, anti-PD-1 antibody, Palbociclib, and the combination group. Waterfall plot shows tumor volume response to the treatment. Each column represents one mouse. * $p < .05$.

Corresponding changes and explanations we have added in the revised manuscript in the results part were as follows:

We have provided more details in methods of the “Genetically engineering mouse model with *Lkb1* deletion” on page 35 of the revised manuscript. We also have updated in Fig. 6g.

12. The clinical datasets of immunotherapy-treated patients in Figure 3A are small and incompletely characterized. Based on the presented data it is not clear whether ICAM-1 is predictive of poor outcomes from immunotherapy specifically or simply represents an adverse prognostic marker. These effects should be clarified. It should also be established whether the impact of ICAM-1 expression on clinical outcomes with immunotherapy is

independent of other variables, such as PD-L1 expression etc.

Response: Thank you so much for your pertinent comments. According to your valuable suggestion, we firstly presented more details of the clinical datasets in figure 3A and 3B (Fig. 4e-f in the revised manuscript). A total of 43 patients with NSCLC treated with anti-PD-(L)1 based therapy in previously published clinical trials (GSE126044, n=16; GSE135222, n=27, Samsung Medical Center), with RNA sequencing data, were adopted and analyzed in figure 3A (Figure 4e in the revised manuscript). The clinical datasets in figure 3B (Figure 4f in the revised manuscript) were collected from previously published clinical cohorts of patients with melanoma treated with anti-PD-1 monotherapy (GSE93157, n=25; Liu 2019, n=21).

Secondly, we assessed the prognostic value of ICAM1 across lung adenocarcinoma and squamous carcinoma patients from TCGA. And we found that higher ICAM1 is not correlated with good prognosis of lung adenocarcinoma patients from TCGA (HR=1.01, p=0.87), or even correlated with poor prognosis in lung squamous carcinoma (HR=1.15, p=0.0265), which indicates that it is not a good prognostic marker (Supporting Figure 22A-B). However, patients with higher ICAM1 expression had longer PFS (progression free survival) in both datasets of immunotherapy-treated patients (melanoma, GSE93157 and Liu 2019; NSCLC, GSE126044 and GSE135222; melanoma, Gide 2019) (Figure 4e-f in the revised manuscript, Supporting Figure 22C). These findings suggested that ICAM1 acted as a promising predictive biomarker for immunotherapy, instead of prognostic marker. And lung cancer patients with higher ICAM1 expression may be sensitive to immunotherapy.

To demonstrate whether ICAM1 served as independent predictive biomarker or interacted with PD-L1, we did multivariate COX regression analysis of clinical and genetic features associated with PFS in the ICI-treated cohort, and we found that ICAM1 is an independent predictive biomarker in the NSCLC cohort (GSE126044 and GSE135222; ICAM1: HR, 0.412; p= 0.013; PD-L1: HR, 0.405; p=0.021; respectively), regardless of PD-L1, while this was not observed in melanoma cohort (Gide, 2019; ICAM1: HR, 0.693; p=0.465; PD-L1: HR, 0.205; p=0.001; respectively) (Supporting Figure 22D-E). ICAM1 can be exploited as a potential predictor for the efficacy of ICIs therapy, especially for non-small cell lung cancer.

Supporting Figure 22: ICAM1 may be exploited as a potential predictor for the efficacy of ICIs therapy in lung cancer. Higher expression of ICAM1 is not associated with a good prognosis but with a survival benefit in patients receiving ICI therapy. (A-B) Kaplan-Meier survival curves of TCGA-LUAD (A) and TCGA-LUSC (B) datasets. The P-value was calculated using the log-rank test. (C) Kaplan-Meier curves of progression-free survival according to the expression level of ICAM1 melanoma patients (Gide, 2019). (D-E) Multivariate association of ICAM1 and PD-L1 with progression-free survival in ICI-treated lung cancer cohort (GSE126044, n=16; GSE135222, n=27) (D) and melanoma cohort (Gide, 2019) (E). Hazard ratio (HRs) was calculated using a multivariate Cox regression analysis.

Corresponding changes and explanations we have added in the revised manuscript in the results part were as follows:

Page 15: Finally, we assessed the correlation between the ICAM1 expression and the survival of ICIs-treated cancer patients. A significant progression-free survival (PFS) benefit was observed in patients with higher expression of ICAM1 both in NSCLC cohorts (n=43, with RNA sequencing data: GSE126044, n=16; GSE135222, n=27, Samsung Medical Center; Fig. 4e) and two melanoma cohorts (GSE93157, n=25; Liu 2019, n=21; Fig. 4f) (Gide 2019, Supplementary Fig. 14A), although ICAM1 itself was not a good prognosis factor (Supplementary Fig. 14B-C). These results indicated that ICAM1 may be exploited as a potential predictor for immunotherapy. We furtherly demonstrated that ICAM1 is an independent predictive biomarker in the NSCLC cohort using multivariate COX regression analysis (ICAM1: HR, 0.412; p= 0.013; PD-L1: HR, 0.405; p=0.021; respectively; supplementary Fig. 14D), while this was not observed in the melanoma cohort (ICAM1: HR, 0.693; p=0.465; PD-L1: HR, 0.205; p=0.001; supplementary Fig. 14E).

Minor points:

13. Several legends in both main figures and particularly supplemental figures don't appear to correspond to the part of the figure referenced. For example, S6d, S7G, S4f and Figure 2D. In addition, the manuscript would benefit from additional editing in several areas.

Response: Thank you so much for your pertinent and kind suggestion, and we are so sorry for any inconvenience we might have brought in your reviewing process. In this new version of manuscript, we have carefully checked and made sure that the figure and table

numbers in the text correspond to the figures and tables provided. Thank you again for your question.

Besides, according to your kind suggestions, there was room for improvement in the English language and grammar considerations in our paper. We tried our best to improve the manuscript and made several corrections. Besides, we had also invited an expert from USA to help revise our manuscript. These changes will not influence the content and framework of the paper. And here we did not list the changes but marked in the revised version. We appreciate for Editors/Reviewers' warm work earnestly, and hope that these corrections will meet with approval.

14. The effects of LKB1 KD on p-p65 expression are not identical in Figures S6A and S6C

Response: Thank you so much for your valuable suggestion. We repeated this experiment in this model and found that mutant LKB1 (LKB1-KD) resulted in downregulation of p-p65, just presented as in Figure S15A in our revised manuscript.

15. It appears that OE of even mutant (kinase dead) LKB1 induces ICAM1 expression. How is this explained?

Response: Thank you so much for your pertinent question. We also have observed that overexpression of mutant (K78I kinase dead) LKB1 could induce ICAM1 expression, even though it was reduced compared to that of wild-type LKB1. The explanation would be that mutant LKB1 could also pull-down RB protein, although to a lesser extent than wild-type LKB1 (Supporting Figure 15), thereby CDK4 phosphorylates lesser RB, since LKB1 and CDK4 compete for the binding site of RB. And these unphosphorylated RB can induce ICAM1 expression through p65 transcription.

16. Was the LLC1-shLkb1-IcamWT cell line also infected with Lenti-control or was this not infected at all?

Response: Thank you for your question. We need more details in the description of our cell lines. LLC1-shLkb1-Icam1^{WT} cell line also was infected with lenti-control, so that it can be set as the accurate control group, to make sure that the positive results are not because of infection of lenti-virus.

17. In Figure S5E transfection with s-NC seems to result in increased ICAM-1 expression over a period of 48 hours, does transfection itself result in increased expression?

Response: Thank you for your valuable question. Due to our carelessness, we repeated the experiments and found that transfection with si-NC itself did not result in increased ICAM-1 expression over time (Supporting Figure 23). So, we revised this in our manuscript in supplementary Figure 8I.

Supporting Figure 23: A549 cells with ectopic wild-type LKB1 expression were transfected with siRNA targeting ICAM1 or with scrambled negative control siRNA. Quantitative flow cytometry analysis of ICAM1 level was shown.

Reviewer #2 (Remarks to the Author): with expertise in LKB-1 mutant lung cancer, immunotherapy

NCOMMS-21-37118

Title: CDK4/6 inhibitor triggers ICAM1-driven immune response and sensitizes LKB1 mutant lung cancer to immunotherapy

In this manuscript, the authors proposed an interesting idea of ICAM is responsible for the responses of augmented response to anti-PD1 in LKB1 mutant tumors through CDK4/6 inhibition. The biggest issue with the series of experiments shown is that the authors did not definitively demonstrate that the phenotype is driven by ICAM1 expression on cancer cells. Furthermore, whether sensitivity in response to CDK4/6 plus PD1 inhibition is LKB1-loss dependent remains unclear. Specifically, there are questions below need to be addressed.

1. Figure 1, in page 5, the authors stated “In comparison with K mouse, KL mouse had lower number of TCF7+ activated T cells and higher number of CTLA4+ exhausted T cells”. It’s not really shown in the figures for CTLA4+ levels or quantification of relative populations change in K and KL model. This issue is throughout the whole session of scRNAseq data, which makes interpretation of the data challenging, and need to be addressed.

Response: Thank you for your valuable and professional question. We indeed should re-organize and re-analyze our scRNA data more quantifiably. In terms of canonical feature cell markers, eleven major cell clusters were discovered, and cell annotations were subsequently calibrated by evaluating the most highly expressed marker genes among each cell cluster (Supporting Figure 1A). Cell composition analysis showed reduced activated T cells and increased exhausted T cells and B cells in tumors from KL mice compared with those from K mice (Supporting Figure 1B). Activated T cells in KL group were characterized by significantly lower expression of TCF7 relative to K group ($p < 0.001$, Supporting Figure 1C); while exhausted T cells displayed significantly higher expression of CTLA4 in KL mice versus that of K mice ($p < 0.0001$, Supporting Figure 1C).

Supporting Figure 1: Quantified cell composition analysis of single cell sequencing data in *Kras*^{mutant}/*Lkb1*^{mutant} (KL) versus *Kras*^{mutant} (K) mice. (A) Heatmap of canonical cell-type markers of 11 major cell types. (B) Absolute numbers of cells from K and KL samples for clusters identified in Figure 1. (C) Expression level of *Tcf7* in activated T cells and that of *Ctla4* in exhausted T cells from lung tumor samples of K and KL mouse. Statistical significance was tested with a two-tailed Mann–Whitney U test. *** $p < .001$; **** $p < .0001$.

To provide more detailed quantitative analysis of the immune profiling data and to evaluate the general effect of *Lkb1* deficiency on different immune cell clusters, we also calculated gene signature-based scores at the single-cell level in different cell types using single sample gene set enrichment analysis (ssGSEA) (David A Barbie, 2009, nature). And we found that activated T cells isolated from KL tumors showed significantly lower “T cells homing on tumor” and “T cell proliferation in immune response” signature scores (Supporting Figure 2A); While exhausted T cells isolated from KL tumors showed significantly higher “PD-1 pathway”, “T cell anergy” signature scores compared with T cells from K tumors, indicating lower T cell infiltration and activation, and higher T cell dysfunction by *Lkb1* deficiency (Supporting Figure 2B). However, no significant difference in signature scores was observed among the other cell clusters, including B cells (Supporting Figure 2C-F). These data suggest that T-cell activation status is the main determinant downstream signaling of *LKB1* in dictating the tumor immune

microenvironment.

Supporting Figure 2: Single sample gene set enrichment analysis with immune-related signatures for cells in the (A) activated T cells cluster, (B) exhausted T cells cluster, (C) neutrophils cluster, (D) macrophages cluster, (E) B cells cluster and (F) DCs cluster. (Data are presented as violin plot (median, 25%-75%, range). Statistical significance was tested with a two-tailed Mann–Whitney U test. ns, not significant; * $p < 0.05$; ** $p < 0.01$; *** $p < 0.001$; **** $p < 0.0001$)

Corresponding changes and explanations we have added in the revised manuscript in the results part were as follows:

Page 5-6: After quality control and cell filtering, we catalogued 14,260 cells into eleven distinct cell lineages annotated with canonical feature cell markers, thus identifying cancer cell types 1, cancer cell types 2, endothelial cells, and immune cells types (including activated T cells, exhausted T cells, B cells, neutrophils, macrophages cells, NK cells, dendritic cells and plasmacytoid dendritic cells) (Fig. 1d, e, f). Cell composition analysis

showed reduced activated T cells and increased exhausted T cells in tumors from KL mice compared with those from K mice (Fig. 1f, Supplementary Fig. 1D). Activated T cells in KL group were characterized by significantly lower expression of Tcf7 relative to K group ($p < 0.001$, Fig. 1g, Supplementary Fig. 1E); while exhausted T cells displayed significantly higher expression of Ctla4 in KL mice versus those of K mice ($p < 0.0001$, Fig. 1g, Supplementary Fig. 1E). To further evaluate the general effect of *LKB1* deficiency on different immune cell clusters, we also calculated gene signature-based scores at the single-cell level in different cell types using single sample gene set enrichment analysis (ssGSEA) 22. Cancer cells in KL group showed significantly higher LKB1 loss signature scores 23, confirming LKB1 deficiency in KL mice (Supplementary Fig. 1F). And we found that activated T cells isolated from KL tumors showed significantly lower “T cells homing on tumor” signature scores; while exhausted T cells isolated from KL tumors showed significantly higher “PD-1 pathway”, “T cell anergy” signature scores compared with T cells from K tumors (Supplementary Fig. 2A, B). However, no significant difference in signature scores was observed among the other cell clusters, including B cells (Supplementary fig. 2C-F). These data suggest that T-cell activation status is the main determinant downstream signaling of LKB1 in dictating the tumor immune microenvironment.

2. Figure 2C, the CellphoneDB analysis was meant to analyze the interaction between ligand and receptors rather than cell-cell interaction. The authors need to specify which ligand/receptor was used for the analysis. In addition, the authors only lay out three different complexes of ICAM1, including LFA-1, ALB2 and SPN. More details are needed for interpretation of the data.

Response: Thank you for your valuable and professional questions. In order to specify which ligand/receptor was used for the analysis, co-culture experiments of activated CD8+ T cells and LLC1-sh*Lkb1* cells with or without *Icam1* overexpression were performed. After cancer cell-T cell interaction for 24 hours, CD8+ T cells were sorted and analyzed. It is interesting to note that CD69 was upregulated on CD8+ T cells after co-culturing with LLC1-sh*Lkb1* cells with *Icam1* overexpression (Supporting Figure 3A). ICAM-1 interacts with LFA-1 with a high affinity and slow dissociation rate, creating a physical force for the adhesion between cells (Tominaga et al., 1998). And Wei Zhang reported that ICAM1-mediated adhesion on cancer cells is a prerequisite for their interaction with T cells through LFA-1 (Wei Zhang, 2022). We observed that using LFA-1 blockade antibodies significantly reduced CD44 expression on T cells after co-culture experiments of CD8+ T cells and LLC1-sh*Lkb1* cells with *Icam1* overexpression (Supporting Figure 3B). Therefore, ICAM1 on cancer cells mediates the interaction with T cells by binding to LFA-1, and LKB1 deficient cancer cells downregulate ICAM1 to reduce the interaction with T cells through LFA-1.

Taken together, to investigate into the interplay between cancer and T cells, we analyzed the receptor-ligand interactions using Cellphone DB. All the complexes relating to ICAM1 were analyzed, and we found only these three complexes displayed differences between K and KL groups. And we further specified that the ICAM1-LFA-1 complexes between cancer cells and T cells were the most possible interaction in lung cancer models.

Supporting Figure 3: ICAM1 on cancer cells mediates the interaction with T cells by binding to LFA-1. (A) Flow cytometry analysis of CD69 expression on activated T cells with co-culturing of LLC1 tumor cells (LLC1-shcontrol-LV-control, LLC1-shLkb1-LV-control, LLC1-shLkb1-LV-Icam1) for 24 hours. The E: T ratio (effector to target) was 10:1. (B) Flow cytometry analysis of CD69 and CD44 expression on activated T cells co-culturing with LLC1 tumor cells (LLC1-shLkb1-LV-Icam1), with or without LFA-1 antibody blockade pre-treatment for 24 hours. The E: T ratio (effector to target) was 10:1. (Results are presented as mean \pm SEM. One-way ANOVA was performed for multi-group comparisons. *** $p < .001$; **** $p < .0001$)

Corresponding changes and explanations we have added in the revised manuscript in the results part were as follows:

Page 11: We also verified this phenomenon from co-culture experiments, in which overexpression of ICAM1 on cancer cells enhanced CD69 and PD-1 expression on CD8+ T cells after their direct contact with tumor cells for 24 hours (Fig 3c-d). ICAM1-mediated adhesion on cancer cells is a prerequisite for their interaction with T cells through LFA-1 24. To determine the receptor that interacts with ICAM-1 on cancer cells, antibody blocking experiments were performed. Pre-treatment of CD8+ T cells with anti-CD11a antibodies (LFA-1 blockade), reduced CD69 and 44 expressions on T cells after co-culture experiments (Fig 3e), indicating that ICAM1 on cancer cells mediates the interaction with T cells by binding to LFA-1.

3. Figure 3A-C, the authors showed ICAM1 levels in patients is correlated with PFS in 3A and 3B. While in mouse model used, ICAM1 levels do not affect tumor growth or survival (3C), instead, it impacts on response to anti-PD1 treatment. Why is this discrepancy between human and animal data?

Response: Thank you so much for your pertinent comments. According to your valuable suggestion, more details are needed for the interpretation of the role of ICAM1. In revised manuscript Fig. 4e-f, supplementary Fig. 14A, we demonstrated that higher ICAM1 was correlated with longer PFS (progression free survival) in ICI-treated cohorts. To further dissect the prognostic value of ICAM1, we analyzed lung adenocarcinoma and squamous carcinoma patients from TCGA (without immunotherapy). And we found that higher ICAM1 is not correlated with good prognosis of lung adenocarcinoma patients from TCGA (HR=1.01, $p=0.87$), or even correlated with poor prognosis in lung squamous carcinoma (HR=1.15, $p=0.0265$), which indicated that it is not a good prognostic marker (Supporting Figure 4A-B). However, patients with higher ICAM1 expression had longer PFS in both datasets of immunotherapy-treated patients (melanoma, Gide 2019; NSCLC, GSE126044

and GSE135222).

Supporting Figure 4: Higher expression of ICAM1 is not associated with a good prognosis factor. (A-B) Kaplan-Meier survival curves of TCGA-LUAD (A) and TCGA-LUSC (B) datasets. The P-value was calculated using the log-rank test.

Furthermore, in our mouse model, we discovered that ICAM1 overexpression alone did not affect tumor growth or survival (revised manuscript 4b-d), instead, it impacts response to anti-PD1 treatment (revised manuscript 4b-d), which is consistent with findings from patients' cohorts. Collectively, these data suggested that ICAM1 acted as a promising predictive biomarker for immunotherapy, instead of prognostic marker. Patients with higher ICAM1 expression may be sensitive to ICI therapy, especially for NSCLC.

Corresponding changes and explanations we have added in the revised manuscript in the results part were as follows:

Page 15: Finally, we assessed the correlation between the ICAM1 expression and the survival of ICIs-treated cancer patients. A significant progression-free survival (PFS) benefit was observed in patients with higher expression of ICAM1 both in NSCLC cohorts (n=43, with RNA sequencing data: GSE126044, n=16; GSE135222, n=27, Samsung Medical Center; Fig. 4e) and two melanoma cohorts (GSE93157, n=25; Liu 2019, n=21; Fig. 4f) (Gide 2019, Supplementary Fig. 14A), although ICAM1 itself was not a good prognosis factor (Supplementary Fig. 14B-C). These results indicated that ICAM1 may be exploited as a potential predictor for immunotherapy. We furtherly demonstrated that ICAM1 is an independent predictive biomarker in the NSCLC cohort using multivariate COX regression analysis (ICAM1: HR, 0.412; p= 0.013; PD-L1: HR, 0.405; p=0.021; respectively; supplementary Fig. 14D), while this was not observed in the melanoma cohort (ICAM1: HR, 0.693; p=0.465; PD-L1: HR, 0.205; p=0.001; supplementary Fig. 14E).

4. Figure 3F, the authors claims that the increased sensitivity to anti-PD1 treatment is due to increased CD8+ T cell infiltration and less exhausted T cells, based on the rational that ICAM1 mediated cancer cell-T cell adhesion and interaction (Figure 2C). However, the following ex vivo experiment showed interaction between ICAM1 expressing tumor cells with PBMC and NK cells. There is no data proving ICAM1

high tumor cells impact on their interaction with T cells. This is very important point need to be proved since the whole story is built upon this hypothesis.

Response: Thank you for your valuable and professional question. We actually need more direct data to support the key hypothesis that downregulation of ICAM1 in KL cells affects trafficking or adhesion of effector immune cells, so that the core conceptual novelty of our manuscript can be justified. Therefore, as with your suggestion, LKB1 proficient/deficient lung tumor bearing orthotopic mouse models (the latter with or without overexpression of ICAM-1) engineered to present ovalbumin SIINFEKL peptide were established (LLC1-sh*control*-ICAM1^{WT}-OVA; LLC1-sh*Lkb1*-ICAM1^{WT}-OVA; LLC1-sh*Lkb1*-ICAM1^{OE}-OVA). Subsequently, splenocytes isolated from OT-I mice were stimulated with 2 nM OVA₂₅₇₋₂₆₄ (SIINFEKL) for 5 days with the presence of 10 ng/mL IL-2. Then the cells in the culture were dyed with celltrace far-red and viewed as CTLs. 5×10⁶ OT-I CTLs were injected intravenously into LLC1-OVA tumor bearing mice (Supporting Figure 5A). Gratifyingly, on day 5 after T cells injection, we observed a sharp decrease of OT-I cells in LKB1 deficient tumors than in LKB1 proficient tumors, indicating the role of LKB1 in influencing CTL infiltration; furthermore, the number significantly increased in LKB1 deficient tumors with ICAM1 overexpression than LKB1 deficient groups (Supporting Figure 5B). Besides, we also investigated into the activated state of OT-I CTLs. CD69 expression was significantly lower in LKB1 deficient tumors than proficient groups. And ICAM1 overexpression significantly improved the activated state of OT-I CTLs in LKB1 deficient tumors (Supporting Figure 5B). Taken together, these data suggest that ICAM1 may be prerequisite for the homing of activated tumor-specific T cells in LKB1 deficient tumors.

Supporting Figure 5: ICAM1 may be prerequisite for the homing of activated tumor-specific T cells in LKB1 deficient tumors. (A) LLC1-OVA tumor cells (LLC1-sh*control*-*Icam1*^{WT}-OVA; LLC1-sh*Lkb1*-*Icam1*^{WT}-OVA; LLC1-sh*Lkb1*-*Icam1*^{OE}-OVA) were implanted. Previously-marked (APC+) and stimulated CD8+ T cells expressing the OT-I were transferred on day 7 and analyzed on day 12. (B) Infiltration number of OT-I cells and expression of CD69 on OT-I cells were analyzed by flow cytometry. (Results are presented as mean ± SEM. One-way ANOVA test was used to analyze the data. ns, not significant; *p<.05; **p<.01; ***p<.001; ****p<.0001.)

Furthermore, as with your suggestions, we also performed ex-vivo coculture experiments of activated T cells and LKB1 deficient LLC1 with or without ICAM1 overexpression to provide more evidences proving ICAM1 high tumor cells impact on their interaction with T cells. We found that overexpression of ICAM1 did enhance CD69 and PD-1 expression on CD8+ T cells after cancer cell-T cell coculturing, indicating higher ICAM1 on cancer cells augmented their interaction with T cells (Supporting Figure 6). Therefore, ICAM1 played a

decisive role in cancer cell-T cell adhesion and interactions in our model.

Supporting Figure 6: Flow cytometry analysis of CD69 and PD-1 expression on activated T cells with or without co-culturing with LLC1 tumor cells (LLC1-sh*Lkb1*-LV-control, LLC1-sh*Lkb1*-LV-*Icam1*) for 24 hours. The E: T ratio (effector to target) was 1:1, 5:1, and 10:1 respectively. (Results are presented as mean \pm SEM. One-way ANOVA test was used to analyze the data. * $p < .05$; ** $p < .01$; *** $p < .001$.)

Corresponding changes and explanations we have added in the revised manuscript in the results part were as follows:

Page 11: To fully demonstrate that ICAM1 mediated cancer cell-T cell adhesion and interactions affects trafficking and activation of effector immune cells in *LKB1* deficient NSCLC, we adoptively transferred congenitally-marked (APC+) stimulated CD8+ T cells expressing the OT-I, which is specific for OVA, into mice bearing *LKB1* proficient/deficient orthotopic lung tumors (LLC1-shcontrol-*Icam1*^{WT}-OVA; LLC1-sh*Lkb1*-*Icam1*^{WT}-OVA; LLC1-sh*Lkb1*-*Icam1*^{OE}-OVA) and assessed OT-I cell infiltration and activation status in lung tumors (Fig. 3a). Gratifyingly, on day 5 after T cells injection, we observed a sharp decrease of OT-I cells in *LKB1* deficient tumors than that in *LKB1* proficient tumors, indicating the role of *LKB1* in influencing CTL infiltration; furthermore, the number significantly increased in LLC1-sh*Lkb1*-*Icam1*^{OE}-OVA tumors than LLC1-sh*Lkb1*-*Icam1*^{WT}-OVA groups (Fig. 3b). Besides, ICAM1 overexpression significantly improved the activated state of OT-I CTLs (CD69) in *LKB1* deficient tumors (Fig. 3b). Taken together, these data suggest that ICAM1 may be prerequisite for the homing of activated tumor-specific T cells in *LKB1* deficient tumors. We also verified this phenomenon from co-culture experiments, in which overexpression of ICAM1 on cancer cells enhanced CD69 and PD-1 expression on CD8+ T cells after their direct contact with tumor cells for 24 hours (Fig 3c-d).

5. Figure 3H, page 12, the authors trying to use IHC examine the “trafficking and infiltration of CD8+ T cells into the TME” (page 12). This assay cannot really distinguish between trafficking versus proliferation of T cells, it merely showed total CD8+ T cells at tumor site. This part needs to be clarified/corrected.

Response: Thank you so much for your pertinent suggestion. According to your valuable comments, we clarified and corrected this part as follows:

Page 13: Subsequently, we sought to address whether ICAM1 correlates with augmented CD8+ T cells in lung cancer patients' samples. IHC of human NSCLC tissue arrays showed that the number of CD8-positive cells was positively correlated with ICAM1 expression on tumor cells (Pearson correlation: $r = 0.642$, $p < 0.0001$). Additionally, CD8+ T cells were often distributed in the ICAM1-positive regions of the tumors (Fig. 3g-h, Supplementary Fig.

11C-D).

6. Figure 4 and supporting evidences, the IP and MS analysis for the interaction between LKB1 and Rb (Fig. S6B) is very obscure and need further clarification. For the regulation of ICAM1 by LKB1, the authors showed it is through Rb (Fig S6H). Looks like ICAM1 levels depend on RB levels rather than LKB1, thus whether LKB1 is deficient is not relevant to either ICAM1 levels. In line with this, the authors further showed LKB1 deficiency “led to a decreased phosphorylation level of p65” (page 16), which is responsible to transcriptional regulation of ICAM1. How about RB status when p65 is inhibited? This part is very confusing since the authors trying to demonstrate two different factors Rb1 and p65, further experimental evidence is needed (either role of p65 in Rb1 deficient cells, or role of Rb1 in p65 knockdown cells during ICAM1 regulation). Figure 4D, what’s the status of ICAM1 levels in LKB1 WT cells?

Response: Thank you so much for your pertinent comments. More details should be clarified for the interaction between LKB1 and RB. To further test how ICAM1 is regulated by LKB1, a control vector or a Flag-LKB1-overexpressing vector was introduced into H1299 cells (an LKB1-WT human LUAD cell line), and IP with an anti-Flag antibody was performed, followed by mass spectrometry (MS) analysis. An intense band between 100 and 130 kDa was observed in cells expressing ectopic LKB1 after IP (Supporting Figure 7A). RB constituted one of the immunoprecipitated candidates based on subsequent MS analysis (Supporting Figure 7B). Furthermore, phosphorylated RB can be specifically targeted by CDK4/6 inhibitors.

Supporting Figure 7: (A) Immunoprecipitation (IP) analysis in H1299 cells with ectopic expression of flag-LKB1 or transduced with parental flag-NC plasmids. The result was visualized by Coomassie blue staining and further analyzed by NanoLC-ESI-MS/MS Analysis. (B) RB was detected as the selected MS/MS spectrum showed.

Corresponding changes and explanations we have added in the revised manuscript in the results part were as follows:

Page 18: To further test how ICAM1 is regulated by LKB1, a control vector or a Flag-LKB1-overexpressing vector was introduced into H1299 cells, and immunoprecipitation with an anti-Flag antibody was performed, followed by mass spectrometry (MS) analysis. RB constituted as one of the immunoprecipitated candidates based on the MS analysis (Supplementary Fig. 14B-C).

Then we discovered either role of p65 in RB deficient cells, or role of RB in p65 knockdown cells during ICAM1 regulation. We firstly evaluated how LKB1 regulates ICAM1 expression through p65. Both qPCR and flowcytometry analysis demonstrated that LKB1 overexpression significantly restored ICAM1 expression, while knockdown p65 remarkably impaired ICAM1 in LKB1 overexpressed cells (Supporting Figure 8A-B). Furtherly, according to your pertinent suggestion, we performed ChIP analysis and found that there is increased p65 occupancy in the promoter of ICAM1 in LKB1-WT group compared to LKB1 loss group (A549-LKB1-WT vs. A549-LKB1-loss) (Supporting Figure 8C), which furtherly demonstrated that LKB1 regulated the transcriptional activity of p65 for the expression of ICAM1. It was observed that by using CDK4/6 inhibitors, pRB was significantly inhibited, and p-p65 was in the meantime upregulated and ICAM1 was therefore increased (supplementary figure 15A). Therefore, p65 was the downstream regulator of RB during ICAM1 regulation. Xin Jin reported that CDK4-phosphorylated RB could suppress transcriptional activity of p65 by specifically binding to p65 (Xin Jin, molecular cell, 2019). We found that the binding of RB with p65 was reduced in LKB1 overexpressed A549 cells through co-IP and IF analysis (A549-LKB1-WT versus. A549-LKB1-loss) (Supporting Figure 8D-F). Together, LKB1 inhibits RB phosphorylation, and therefore impairs the interaction of RB with p65, and the occupancy of p65 on the promoter of ICAM1 is therefore augmented.

Finally, we found that ICAM1 was also upregulated in LKB1-WT cells after Palbociclib treatment (Supporting Figure 8G).

Supporting Figure 8: (A-B) Analysis of ICAM1 expression of LKB1-loss cells expressing vector+siNC, LKB1+siNC, or LKB1+siip65. RT-PCR (A) and flowcytometry (B) analysis of ICAM1 expression. (C) ChIP assay was performed with cell lysates from A549/LV-control and A549/LV-LKB1 cells. A pair of primers flanking the p65 binding site within the ICAM-1 promoter was used in PCR. Real-time PCR was employed to the ChIP assay. (D) Immunoprecipitation analysis of the interaction among RB, and p65 performed in A549 cells (LKB1-loss cells). (E) Multiple immunofluorescence labeling using anti-p65 and anti-RB antibodies in A549-LKB1-loss cells versus A549-LKB1-WT cells. (F) Immunoprecipitation analysis of the interaction among RB, and p65 performed in A549-LKB1-loss cells versus A549-LKB1-WT cells. (G) Flowcytometry analysis of ICAM1 expression in A549-LKB1-WT cells with or without Palbociclib treatment.

(Results are presented as mean \pm SEM. One-way ANOVA was performed for multi-group comparisons. t test was performed for comparisons of two groups. ** $p < .01$; **** $p < .0001$)

Corresponding changes and explanations we have added in the revised manuscript in the results part were as follows:

Page 19: ChIP analysis demonstrated that p65 occupancy in the promoter of ICAM1 is increased in LKB1-WT group compared to LKB1 loss group (Fig. 5j), which furtherly demonstrated that LKB1 regulates the transcriptional activity of p65 for the expression of ICAM1. In agreement with this, p65 knockdown impaired ICAM1 expression in LKB1 restored group (Supplementary Figure 16G-H). CDK4-phosphorylated RB specifically binds to p65 and inhibits its transcriptional activity²⁹. We observed that LKB1 reconstitution interfered RB-p65 interactions (Supplementary Figure 16I-J).

7. Figure 5, the authors only showed LLC-shLKB1 tumor in response to palbo+PD1, how about LKB1 wt LLC parental cells? Do they respond to palbo+PD1? More importantly, if you overexpress or knockdown ICAM1, will the tumor still respond to palbo+PD1? Obviously CDK4/6 inhibitor can directly activate T cells and augment response to PD1 blockade, the authors need to segregate the effect of CDK4/6 inhibition on tumor cells versus T cells (and other immune cells), and demonstrate whether the phenotype observed is due to upregulation of ICAM1.

Response: 1) Thank you for your valuable and professional question. According to your pertinent suggestions, we firstly performed the same experiments in the parental cell line (LLC1-*Lkb1*^{WT}) and LLC1 with LKB1 overexpression (LLC1-*Lkb1*^{OE}), to fully demonstrate *Lkb1* status on the immune-modulatory effects. And we found that combined PD-1 blockade and CDK4/6 inhibition significantly slowed tumor growth in both the LLC1-*Lkb1*^{WT} and LLC1-*Lkb1*^{OE} group (Supporting Figure 9A-B). However, the synergistic effects of the combination therapy were diluted in comparison with LLC1-*Lkb1*^{KD} group (LLC1-sh*Lkb1*), especially for mice with LKB1 overexpression, anti-PD-1 mAb treatment alone already exhibited significant antitumor effect against LLC1-*Lkb1*^{OE} tumors, implying that LKB1-mutant/deficient NSCLC is uniquely sensitive to the synergistic effect of palbociclib/anti-PD-1 combination. (Supporting Figure 9A-B).

Supporting Figure 9: CDK4/6 inhibitors sensitize LKB1-deficient tumors to ICI therapy. (A-B) C57BL/6 mice were implanted subcutaneously with LLC1 cell lines with wild-type *Lkb1* (A) or overexpressed *Lkb1* (B) and treated with isotype IgG, anti-PD-1 Ab, Palbociclib or combination of both anti-PD-1 Ab and Palbociclib. Macroscopic appearance of tumors (left), plots of tumor volume (middle) and quantification of tumor weight (right) were shown. (Results are presented as mean \pm SEM. A mixed-effects model followed by Tukey's multiple comparisons was performed to compare the tumour growth curves in different treatment groups, and one-way ANOVA was performed for multi-group comparisons. n.s., not significant; * $p < .05$; *** $p < .001$)

Corresponding changes and explanations we have added in the revised manuscript in the results part were as follows:

Page 22: The degree of selectivity of the palbociclib/anti-PD-1 approach for *LKB1*-deficient NSCLC was also assessed. Compared to *LKB1* wildtype or overexpressed tumors (Supplementary Fig. 18C-D), *LKB1* deficient NSCLC is uniquely sensitive to the synergistic effect of the combination therapy.

2) In order to segregate the effect of CDK4/6 inhibition on tumor cells versus T cells, we performed ex-vivo co-culturing experiments. Tumor cells were pre-treated with CDK4/6 inhibitors or DMSO for 48 hours, then they were co-cultured with activated CD8+ T cells for another 24 hours. T cells were subsequently sorted and analyzed with flow cytometry. It was observed that CDK4/6i pre-treatment of tumor cells significantly enhanced the interaction of T cells with cancer cells, since CD69, CD107a and PD-1 expression on T cells were significantly increased, which indicated that CDK4/6 inhibition on tumor cells reinforced the interaction between cancer cells and T cells (Supporting Figure 10).

Supporting Figure 10: CDK4/6 inhibition on tumor cells reinforced their interactions with T cells. LLC1-shLkb1 Tumor cells were pre-treated with CDK4/6 inhibitors or DMSO for 48 hours, and subsequently co-cultured with activated CD8+ T cells for another 24 hours. T cells were subsequently sorted and analyzed with flow cytometry. (Results are presented as mean \pm SEM. One-way ANOVA was performed for multi-group comparisons. t test was performed for comparisons of two groups. *** $p < .001$; **** $p < .0001$)

Corresponding changes and explanations we have added in the revised manuscript in the results part were as follows:

To segregate the effect of CDK4/6 inhibition on tumor cells versus T cells, we performed ex-vivo co-culturing experiments, and tumor cells were pre-treated with CDK4/6 inhibitors or PBS. CDK4/6 inhibition on tumor cells reinforced their interactions with T cells (Supplementary Fig.17A); CDK4/6 inhibitors alone did not increase tumor cell apoptosis (Supplementary Fig.17B), while co-culturing of CDK4/6 inhibitors pretreated tumor cells

with NK-92 cells significantly increased apoptosis rate of tumor cells (Supplementary Fig.17C), implying the potential value of CDK4/6 inhibitors in activating the immune microenvironment.

4) Besides, we totally agree with you that increased sensitivity to Palbociclib and PD-1 blockade depending on upregulation of ICAM1 should be demonstrated, which is one of the critical questions of our manuscript. Therefore, according to your precious suggestions, we have established isogenic derivatives of the LLC1-sh*Lkb1* cell line with or without knockout of *Icam1* using CRISPR/Cas9 and repeated the therapeutic experiment. And we found that antitumor effects of the combined PD-1 blockade and CDK4/6 inhibition were significantly impaired in LLC1-sh*Lkb1*-sg*Icam1* tumors (Supporting Figure 11A). To demonstrate whether the phenotype observed is due to upregulation of ICAM1, we systemically evaluated the effects of treatment with palbociclib or the combination therapy on the tumor microenvironment in the setting of ICAM-1 depletion (LLC1-sh*Lkb1*-sg*Icam1* vs. LLC1-sh*Lkb1*-sgCon). We found that LLC1-sh*Lkb1*-sg*Icam1* tumors exhibited a significant reduction in CD8+ T cell infiltrate, and CD69, CD44 expression on CD8+ T cells compared with LLC1-sh*Lkb1*-sgCon tumors after palbociclib treatment or combination therapy. Furthermore, infiltration of NK cells and DCs were significantly reduced in LLC1-sh*Lkb1*-sg*Icam1* tumors after combination therapy compared with that of LLC1-sh*Lkb1*-sgCon tumors. Interestingly, MDSCs and tumor-associated neutrophils (TAN) infiltration were significantly increased in the setting of ICAM-1 depletion after treatment with palbociclib or the combination therapy (Supporting Figure 11B-C). Taken together, CDK4/6 inhibitors or the combination therapy induced an immune-active tumor microenvironment, which is dependent on ICAM1.

Supporting Figure 11: ICAM-1 depletion abrogates anti-tumor immunity conferred by the Palbociclib/ICI

combination therapy. (A) C57BL/6 mice were inoculated with LLC1/sh*Lkb1* with or without *Icam1* knock-out cells and received the combination treatments versus vehicle. Tumor sizes were monitored. (B) Representative immunohistochemistry images of ICAM1 and CD8 α in LLC1-sh*Lkb1*/sgCon and LLC1-sh*Lkb1*/sg*Icam1* tumors subjected to indicated treatment. Scale bar, 50 μ m. (C) Flowcytometry analysis of immune cell infiltration (percent of CD45+CD3+CD8+ cells, CD45+CD3-NK1.1+ cells, CD45+CD11b+CD11c+CD103+ cells, CD45+CD11b+Gr-1+ cells, CD45+CD11b+CXCR2+ cells, CD45+CD11b+F4/80+ cells), expression of CD69, CD44 on CD8+ T cells and CD107a on NK cells in LLC1-sh*Lkb1*/sgCon and LLC1-sh*Lkb1*/sg*Icam1* tumors followed by indicated treatment arm. (Results are presented as mean \pm SEM. One-way ANOVA was performed for multi-group comparisons. n.s., not significant; * p <.05; ** p <.01; *** p <.001; **** p <.0001)

Corresponding changes and explanations we have added in the revised manuscript in the results part were as follows:

Page 28: Subsequently, isogenic derivatives of the LLC1-sh*Lkb1* cell line with or without knockout of *Icam1* using CRISPR/Cas9 were established. The mice received the same combinational therapy. And we found that antitumor effects of the combined PD-1 blockade and CDK4/6 inhibition were significantly impaired in LLC1-sh*Lkb1*-sg*Icam1* tumors (Fig. 8g).

Accordingly, we systemically evaluated the effects of treatment with palbociclib or the combination therapy on the tumor microenvironment in the setting of ICAM-1 depletion. IHC staining of tumor tissue displayed reduced ICAM1 expression and CD8+ T cell infiltration in LLC1-sh*Lkb1*-sg*Icam1* tumors compared with LLC1-sh*Lkb1*-sgCon tumors after palbociclib treatment or combination therapy (Fig. 8h, supplementary Fig. 21A). Flow cytometry analysis demonstrated that LLC1-sh*Lkb1*-sg*Icam1* tumors exhibited a significant decrease in CD8+ T cell infiltrate, and CD69, CD44 expression on CD8+ T cells versus LLC1-sh*Lkb1*-sgCon tumors after CDK4/6 inhibition or combination therapy. Furthermore, infiltration of NK cells and DCs were significantly lower in LLC1-sh*Lkb1*-sg*Icam1* tumors after combination therapy versus that of LLC1-sh*Lkb1*-sgCon tumors. Taken together, these depletion therapies reversed the advantages conferred by the palbociclib/ICI combination therapy, confirming that CDK4/6 inhibition promotes the effect of anti-PD-1 immunotherapy in an ICAM1-dependent manner (Fig. 8h, supplementary Fig.21A-B).

Reviewer #3 (Remarks to the Author): with expertise in scRNAseq, system immunology

It is an interesting topic to study the underline mechanism between LKB1 mutation and the resistance to immune checkpoint inhibitors (ICIs) treatment in lung adenocarcinoma (LUAD). Novel therapies are urgently needed to improve clinical outcomes of LUAD patients with LKB1 mutation, as they are refractory to most currently available therapies. The authors applied single-cell RNA-seq technology to profile the tumor microenvironment with genetically engineered mouse models (Lkb1-loss/Kras mut). They found LKB1 mutant tumors reduced interaction with infiltrated immune-competent cells by downregulation of the cell surface protein ICAM1. They further demonstrated CDK4/6 inhibitor would rescue ICAM1 expression in LKB1 mutant cell line models and in vivo survival study also shows

CDK4/6 inhibitor plus anti-PD-1 antibody has synergistic effects and induced immunologically “hot” tumor microenvironment. Although both the topic and findings are intriguing, the overall logical flow of the paper and the connections between evidence and conclusion need major revision. Some descriptions in the method section are not clear enough.

Major:

1. Figure 1, what are those activated T or exhausted T clusters? Any specific T cell markers to show their identity? And why using PD-1 antibody for combination instead of anti-CTLA4 antibody, exhausted T cells seems expression high levels of CTLA-4.

Response: Thank you for your valuable and professional question. More details should be provided according to your valuable suggestions. In terms of canonical feature cell markers, eleven major cell clusters were discovered, and cell annotations were subsequently calibrated by evaluating the most highly expressed marker genes among each cell cluster (Supporting Figure 1A). For T cell clusters we used Cd3e, Cd3d and Cd5 to show their identity. And for activated T cells, they were identified with higher level of Tcf7 and gzmb, while for exhausted T cells, they were marked with higher expression of ctla4 (Supporting Figure 1B). The other detailed cell markers to show cell cluster identity were provided in supplementary table 1.

Supporting Figure 1: (A) Heatmap of canonical cell-type markers of 11 major cell types. (B) Expression level of Tcf7 in activated T cells and that of Ctla4 in exhausted T cells from lung tumor samples of K and KL mouse. (Data are presented as violin plot (median, 25%-75%, range). Statistical significance was tested with a two-tailed Mann–Whitney U test. ns, not significant; *p<.05; **p<.01; ***p<.001; ****p<.0001.)

To provide more detailed quantitative analysis of the immune profiling data and to evaluate the general effect of *Lkb1* deficiency on different immune cell clusters, we also calculated gene signature-based scores at the single-cell level in different cell types using single sample gene set enrichment analysis (ssGSEA) (David A Barbie, 2009, nature). And we found that activated T cells isolated from KL tumors showed significantly lower “T cells homing on tumor” and “T cell proliferation in immune response” signature scores (Supporting Figure 2A); While exhausted T cells isolated from KL tumors showed significantly higher “PD-1 pathway”, “T cell anergy” signature scores compared with T cells from K tumors (Supporting Figure 2B). Furthermore, based on clinical guidelines, anti-PD-1/PD-L1 therapy were recommended as first-line therapy for advanced lung cancer. Therefore, we used PD-1 antibody for combination instead of anti-CTLA4 antibody.

Supporting Figure 2: Single sample gene set enrichment analysis with immune-related signatures for cells in the (A) activated T cells cluster, (B) exhausted T cells cluster. (Data are presented as violin plot (median, 25%-75%, range). Student's t-test was used to compare the variables of two groups. n.s., not significant; * $p < .05$; ** $p < .01$; *** $p < .001$; **** $p < .0001$)

Corresponding changes and explanations we have added in the revised manuscript in the results part were as follows:

Page 5-6: After quality control and cell filtering, we catalogued 14,260 cells into eleven distinct cell lineages annotated with canonical feature cell markers, thus identifying cancer cell types 1, cancer cell types 2, endothelial cells, and immune cells types (including activated T cells, exhausted T cells, B cells, neutrophils, macrophages cells, NK cells, dendritic cells and plasmacytoid dendritic cells) (Fig. 1d, e, f). Cell composition analysis showed reduced activated T cells and increased exhausted T cells in tumors from KL mice compared with those from K mice (Fig. 1f, Supplementary Fig. 1D). Activated T cells in KL group were characterized by significantly lower expression of *Tcf7* relative to K group ($p < 0.001$, Fig. 1g, Supplementary Fig. 1E); while exhausted T cells displayed significantly higher expression of *Ctla4* in KL mice versus those of K mice ($p < 0.0001$, Fig. 1g, Supplementary Fig. 1E). To further evaluate the general effect of *LKB1* deficiency on different immune cell clusters, we also calculated gene signature-based scores at the single-cell level in different cell types using single sample gene set enrichment analysis (ssGSEA) 22. Cancer cells in KL group showed significantly higher *LKB1* loss signature

scores 23, confirming LKB1 deficiency in KL mice (Supplementary Fig. 1F). And we found that activated T cells isolated from KL tumors showed significantly lower “T cells homing on tumor” signature scores; while exhausted T cells isolated from KL tumors showed significantly higher “PD-1 pathway”, “T cell anergy” signature scores compared with T cells from K tumors (Supplementary Fig. 2A, B). However, no significant difference in signature scores was observed among the other cell clusters, including B cells (Supplementary fig. 2C-F). These data suggest that T-cell activation status is the main determinant downstream signaling of LKB1 in dictating the tumor immune microenvironment.

2.Figure3. A and B show the ICAM1 expression level correlates with patient survival (LUAD and melanoma). How about adding LKB1 mutation status as a cofactor. For the survival study design, to rule out the possibility that ICAM1 overexpression alone is enough to show benefit from PD-1 blockade treatment, I would suggest adding the parental lewis lung cancer cells without Lkb1 knockdown with or without ICAM1 over-expression as controls. For figure3G, in the main text, the authors mentioned using the GSE72094 dataset, while the figure legend label shows the TCGA dataset, please clarify.

Response: Thank you so much for your pertinent comments.

1) More details should be provided for figure 3A-B (Fig. 4e-f in the revised manuscript). It was observed that ICAM1 expression level correlates with patients' progression free survival of ICI-treated cohort. Clinical information and RNA sequencing data were obtained from previously published article (GSE93157 and Liu 2019; GSE126044 and GSE135222). Since these two ICI-treated cohorts are in lacking of the DNA mutation data, there were limitations for us to analyze *LKB1* mutation status as a cofactor. Interestingly, PD-L1 expression level can be exploited for analysis. To demonstrate whether ICAM1 served as independent predictive biomarker or interacted with PD-L1, we did multivariate COX regression analysis of clinical and genetic features associated with PFS in these two ICI-treated cohort (GSE126044 and GSE135222; Gide2019), and we found that ICAM1 is an independent predictive biomarker in the NSCLC cohort (ICAM1: HR, 0.412; p= 0.013; PD-L1: HR, 0.405; p=0.021; respectively), regardless of PD-L1, while this was not observed in melanoma cohort (ICAM1: HR, 0.693; p=0.465; PD-L1: HR, 0.205; p=0.001; respectively). ICAM1 can be exploited as a potential predictor for the efficacy of ICIs therapy, especially for non-small cell lung cancer.

Supporting Figure 3: Multivariate association of ICAM1 and PD-L1 with progression-free survival in ICI-treated lung cancer cohort (GSE126044, n=16; GSE135222, n=27) (A) and melanoma cohort (Gide 2019, n= 41) (B). Hazard ratio (HRs) was calculated using a multivariate Cox regression analysis.

2) To rule out whether ICAM1 overexpression alone is enough to show benefit from PD-1 blockade treatment, we established LLC1 cell lines with *Icam1* overexpression and intact LKB1 (LLC1-*Icam1*-WT and LLC1-*Icam1*-OE), and implanted these cells subcutaneously, followed by ICI therapy or IgG isotype. Analysis of tumor growth curves and survival showed that LLC1-*Icam1*-WT tumors and LLC1-*Icam1*-OE tumors rendered equivalent responsiveness to ICI therapy, with similar effect of tumor control and survival benefit under ICI therapy (Supporting Figure 4A-B). These results suggest that ICAM1 overexpression is not an essential prerequisite to benefit from ICI therapy when LKB1 is intact, whereas ICAM1 overexpression could sensitize LKB1-deficient tumors to ICI therapy.

Supporting Figure 4: C57BL/6 mice were subcutaneously inoculated with Lewis lung cancer cells with *Icam1* overexpression (LLC1-*Icam1*-OE) and administered different treatments (control immunoglobulin G (Vehicle), palbociclib or anti-PD-1 Ab, or co-treatment with palbociclib and anti-PD-1 Ab). (A-B) Tumor size (A) and survival (B) in different treatment arms were monitored.

3) For figure 3G, we are sorry for our carelessness. Actually, we had used the data from TCGA for analysis. And we have revised it in our new version of manuscript, thank you so much for your kind remaindering.

Corresponding changes and explanations we have added in the revised manuscript in the results part were as follows:

Page 15: Finally, we assessed the correlation between the ICAM1 expression and the survival of ICIs-treated cancer patients. A significant progression-free survival (PFS)

benefit was observed in patients with higher expression of ICAM1 both in NSCLC cohorts (n=43, with RNA sequencing data: GSE126044, n=16; GSE135222, n=27, Samsung Medical Center; Fig. 4e) and two melanoma cohorts (GSE93157, n=25; Liu 2019, n=21; Fig. 4f) (Gide 2019, Supplementary Fig. 14A), although ICAM1 itself was not a good prognosis factor (Supplementary Fig. 14B-C). These results indicated that ICAM1 may be exploited as a potential predictor for immunotherapy. We furtherly demonstrated that ICAM1 is an independent predictive biomarker in the NSCLC cohort using multivariate COX regression analysis (ICAM1: HR, 0.412; p= 0.013; PD-L1: HR, 0.405; p=0.021; respectively; supplementary Fig. 14D), while this was not observed in the melanoma cohort (ICAM1: HR, 0.693; p=0.465; PD-L1: HR, 0.205; p=0.001; supplementary Fig. 14E).

3. Figure 2. E and F show the ICAM1 protein levels in LKB1-MUT and LKB1-loss cells are significantly different. Is there any difference between different LKB1 mutations, both in the cell line model and patients (TCGA) ?

Response: Thank you so much for your valuable questions. LKB1 deficiency (both LKB1 loss and LKB1 mutant) leads to significantly downregulation of ICAM1. Interestingly however, we also have observed that overexpression of mutant (K78I kinase dead) LKB1 could induce ICAM1 expression, even though it was reduced compared to that of wild-type LKB1. Therefore, according to your suggestions, we compared ICAM1 expression among different mutation types of LKB1, both in the cell line model from CCLE (Supporting Figure 5A) and patients' data from TCGA, and no significant difference was observed. But ICAM1 was invariably decreased in LKB1 WT group compared to LKB1 deficient groups (Supporting Figure 5A-C). Together, it was confirmed that LKB1 inactivation is associated with downregulation of ICAM1 in lung cancer model.

Supporting Figure 5: LKB1 regulates ICAM1 expression, regardless of the existence of KRAS co-mutation. (A) Relative RPKM values of ICAM1 in LKB1 mutant versus LKB1 wildtype lung cancer cells from CCLE,

with or without KRAS co-mutations. (B-C) Analyses of correlation between ICAM1 expression and different LKB1 mutation types, with (B) or without (C) KRAS co-mutation using TCGA-LUAD dataset. (Data are presented as box plot (median, 25%-75%, range) or mean \pm SEM. One-way ANOVA was used to analyze the data unless stated. n.s., not significant; * $p < .05$; ** $p < .01$; *** $p < .001$; **** $p < .0001$)

Corresponding changes and explanations we have added in the revised manuscript in the results part were as follows:

Page 9: Gene profiles of patients with LUAD from TCGA were analyzed. The tumors with mutated *LKB1* displayed lower levels of ICAM1 than those with WT *LKB1* (Fig. 2c), regardless of KRAS co-mutation and *LKB1* mutation types (Supplementary Fig. 7A-B). We then assessed the association between *LKB1* and ICAM1 in clinical samples by IHC. Decreased ICAM1 was observed in more than 80% of *LKB1*-mutated tumor samples examined, whereas 80% of *LKB1*-WT specimens showed intensified ICAM1 staining (Fig. 2d). Together, these data confirmed that *LKB1* deficiency is associated with suppression of ICAM1 in lung cancer. To demonstrate this finding further, we examined ICAM1 expression across a panel of *LKB1* mutant versus *LKB1* wildtype lung cancer cell lines. ICAM1 mRNA and protein levels were either significantly downregulated or completely undetectable in *LKB1* mutant cell lines, regardless of *KRAS* co-mutation (Fig. 2e-f).

4.Figure6. C is hard to understand and the communication signature score calculation method needs a clear description.

Response: Thank you so much for your pertinent comments. More clarifications should be provided to make our manuscript clear and understandable.

We revised our manuscript and described the method of the communication signature score calculation as follows:

To depict the physical cell-cell interaction landscape using our bulk-RNA seq data, we developed intercellular communication signatures. As described above in our scRNA-seq data analysis, we identified 11 cell types and their corresponding marker genes. For each cell type, we matched its marker gene-encoded surface proteins to their cognate ligands/receptors based on the assembled ligand-receptor pairs from the STRING website. Then a signature containing all the ligands/receptors proteins that can bind to its marker gene-encoded proteins was defined as the communication signature for this cell type. The communication signatures for these 11 cell types were summarized in supplementary table 2.

5.The experimental evidence to support LKB1 competing with CDK4/6 and modulate RB phosphorylation is too weak. FigureS6F, IP endogenous RB protein and check the CDK4 and LKB1 protein binding would tell if the two bind to the similar site on RB.

Response: Thank you for your valuable and professional question. More details should be provided for the interaction of LKB1 competing with CDK4/6. We have performed IP endogenous RB protein and found that CDK4 and LKB1 protein can both be pulled down by endogenous RB in H1299 cells (revised manuscript Figure 5h). Moreover, CDK4, a known interactor of RB, was also pulled down by LKB1, although to a lesser extent than

that by RB, further corroborating that LKB1 and CDK4 compete for RB binding (revised manuscript Figure 5h). Besides, it was reported that CDK4/Cyclin D specifically docked in the C-terminal helix motif of RB. Bioinformatic analysis predicted the possibility of LKB1 docking on this site of RB, which was statistically significant ($p < 0.05$), demonstrating that LKB1 had similar binding site with CDK4, indicating the potential for docking competition (Supplementary Fig. 14G). Furthermore, we performed the CO-IP experiment to analyze the interactions among LKB1, RB and CDK4. And it was observed that LKB1 overexpression interfered with the association between CDK4 and RB. Taken together, LKB1 competes with CDK4/6 to modulate RB phosphorylation.

Supporting Figure 6: (A) Immunoprecipitation analysis of the interaction among LKB1, RB, and CDK4 performed in H1299 cells expressing intact LKB1. (B) Bioinformatics tool (Pepsite) was utilized to predict the possibility of LKB1 docking on the C-terminal helix motif of RB (SKFQQKLAEMTSTRTRMQKQK). (C) Immunoprecipitation analysis of the relationship among LKB1, RB, and CDK4, was determined using anti-flag or anti-myc magnetic beads with or without ectopic flag-LKB1 and myc-CDK4 overexpression.

We revised our manuscript and described the method of the communication signature score calculation as follows:

Page 19: Since CDK4 phosphorylates RB²⁹, LKB1 suppressed the phosphorylation of RB by affecting CDK4, we therefore speculated that LKB1 competed with CDK4 for RB binding, which led to reduced RB phosphorylation. It was reported that CDK4/Cyclin D specifically docked in the C-terminal helix motif of RB^{28,30}. Bioinformatic analysis predicted the possibility of LKB1 docking on this site of RB, which was statistically significant ($p < 0.05$), demonstrating that LKB1 possibly had the same binding site with CDK4, indicating the potential for docking competition (Supplementary Fig. 15G). Subsequent IP experiments supported this hypothesis; specifically, LKB1 overexpression interfered with the interaction of CDK4 with RB, and vice versa (Supplementary Fig. 15H). However, CDK4-RB interaction recovered in LKB1-MUT group (Supplementary Fig. 15I).

6. The authors provided very limited evidence to support the very detailed mechanisms and somehow, it's confusing to catch the key point (the final model figure 7G): for example, they mentioned deficiency of LKB1 activate DNMT1 and thereby silencing STING expression, meanwhile, they also mentioned ICAM1 might

be regulated by STING signaling.

Response: Thank you for your valuable and professional suggestions. More clarifications and evidences should be provided for our mechanism model. We have demonstrated that LKB1 and CDK4 compete for RB phosphorylation, LKB1 deficiency therefore resulted in RB hyperphosphorylation, and led to p53 inactivation and ICAM1 transcriptional downregulation; CDK4/6 inhibitors can specifically inhibit RB phosphorylation and impaired *ICAM1* transcription was therefore restored, which was our key points explaining how LKB1 regulating ICAM1.

But we also tried to provide more insights in comparisons with the others' work that have investigated into *LKB1* mutant NSCLC. Kitajima et al reported that LKB1 loss results in suppression of STING, facilitating escape of type I interferon and other STAT1-driven effector programs mediated immune response, raising important mechanistic insights into their resistance to PD-1/PD-L1 ICB (Kitajima, 2018). This excellent study gave us lots of inspirations to discover more mechanisms behind this subtype of lung cancer. We provided additional elements to the current model, in which LKB1 deficiency leads to ICAM1 mediated disturbance in both adaptive (CD8+ T cells) and innate (NK cells) immunity. Mechanistically, we have tried to find possible insightful link. LKB1 deficiency leads to the hyperphosphorylation of RB and thereby activation of E2F1 (E2F1 was based on references). DNMT1, a recognized E2F target gene based on previous work, is thus upregulated and plays its role in silencing STING expression, we believe that our work to some extent confirmed with the work of Kitajima, et al.. Besides, we mentioned that ICAM1 might be regulated by STING, which was on the basis of previous work.

Taken together, we have added substantial evidences to support our theory and updated our final model in supplementary figure 21, to make it simple and understandable.

Supporting Figure 7: Schematic of the mechanism through which mutant *LKB1* downregulates *ICAM1* transcription in a phosphorylated RB-dependent manner, and leads to impaired cancer cell-T cell adhesion and interaction. Using CDK4/6 inhibitors activated *ICAM1* transcription and improved cytotoxic cell infiltration and activity, re-sensitize LKB1 mutant cancer cells to anti-PD-1 immunotherapy.

Minor:

1. Figure 2B please label the gene numbers in the Venn diagram. Since H460 was used as a model as well. Any specific reason why only shows A549 DE genes?

Response: Thank you for your valuable and professional question. We actually have sent both H460 and A549 cells for mRNA sequencing analysis, and we would like to provide differentially expressed genes from the sequencing data of H460 cells in our next work, therefore we only adopted A549 cells for further analysis. Besides, as with your suggestions, we have labeled the gene numbers in the Venn diagram in figure 2B, and we have also revised this in our new version of manuscript in Figure 2b.

2. Figure 2C is difficult to understand. Please add more description or reorganized the plot. If the point is to show ICAM1 from cancers will recruit T cells by T cell surface proteins (ALB2, LFA, SPN). I don't think it's necessary to show ICAM1 from T cell and T cell surface proteins in cancer cells.

Response: Thank you for your valuable question. First of all, according to your suggestion, we have selected ICAM1 from cancers with the interaction with T cell surface proteins and highlighted these parts (we have updated in Supplementary Figure 6D of the revised manuscript). Besides, more descriptions should be added. To investigate into the interplay between cancer and T cells, we analyzed the receptor-ligand interactions using Cellphone DB. All the complexes relating to ICAM1 were analyzed, and we found only these three complexes displayed differences between K and KL groups (ALB2, LFA-1, SPN). ICAM-1 interacts with LFA-1 with a high affinity and slow dissociation rate, creating a physical force for the adhesion between cells (Tominaga et al., 1998). And Wei Zhang reported that ICAM1-mediated adhesion on cancer cells is a prerequisite for their interaction with T cells through LFA-1 (Wei Zhang, 2022). Therefore, ICAM1 on cancer cells mediates the interaction with T cells by binding to LFA-1, which is also the most possible complexes in our model, LKB1 deficient cancer cells downregulate ICAM1 to reduce the interaction with T cells through LFA-1.

3. Figure S6H add phospho-Rb blot should be added.

Response: Thank you for your valuable suggestion. We have added the phospho-RB blot, and revised it in our new version of manuscript in Supplementary Figure 14J.

4. Figure 6B, I would suggest adding a quantification plot for easier reading the result.

Response: Thank you for your valuable suggestion. We have added a quantification plot for figure 6B, as Supplementary Figure 19A and we have revised this in our new version of manuscript.

5. Figure 2C, 2D labels were swapped.

Response: Thank you for your valuable remaining. We are so sorry for our carelessness, and we have corrected and updated this in figure 2 of our revised manuscripts.

6.FigureS5. B no legend on the plot (the meaning of the dots?). What is the purpose of using NK-92 or activated PBMCs?

Response: Thank you for your valuable question. The dots in figure S5B indicated the significantly upregulated pathways in ICAM1-high group compared to that of ICAM1-low group. We have also supplemented more details and updated our results in Supplementary Figure 8C of the revised manuscript.

For the reason why we use NK-92 cells or activated PBMCs, we have demonstrated that ICAM1 on tumor cells plays an important role in the interactions of tumor cells with immune cytotoxic cells. ICAM1 functions not only in cell-cell adhesions, but also in co-stimulating the immune cytotoxic cells, including specific immunity, the CD8+ T cells, which we use activated PBMCs as representatives (sorted PBMCs were pretreated with IL-2 and anti-CD28 antibodies for three days); and innate immunity, the NK-92 cells as the study model, in investigating ICAM1 dependent NK cell mediated cytotoxicity. Therefore, we selected NK-92 and activated PBMCs for our analysis.

Reviewer #4 (Remarks to the Author): with expertise in CDK4/6

Summary: The manuscript from Bai et al. interrogates the coordinate impact of KRAS and LKB1 on the immunological landscape in the lung tumor microenvironment. Using single cell sequencing they show that tumors with combined KRAS/LKB1 perturbation exhibit more T-cell exhaustion, with less activated T-cells. Analysis of the tumor compartment revealed that ICAM1 deficiency is associated with LKB1 deficient tumors, which was interrogated in mouse and human preclinical models, as well as TCGA data. ICAM1 expression is associated with improved outcome with anti-PDL1 based therapies in pre-clinical models and melanoma and lung cancer clinical cohorts. CDK4/6 inhibitors can increase ICAM1 in an RB-dependent fashion, which is believed to involve a direct interaction of LKB1/RB/CDK4. Subsequently, CDK4/6 inhibition cooperates with immune checkpoint inhibition.

Critique: Overall this is a well performed study investigating the impact of LKB1/KRAS on the tumor microenvironment and immunological responses. However, there is concern relative to the interpretation of results and the intersection with multiple other studies in this area of research. In particular, given that multiple studies have described the basis for which LKB1 deficiency impact on the tumor microenvironment the contribution of this study needs to be carefully evaluated versus these other mechanisms which have been described in comparable models.

Specific points are enumerated below.

1. To indicate that the tumor microenvironment is an immune desert in the KL mice

seems hard to reconcile with the substantial infiltration within the tumor microenvironment. Rather T-cell activation status is the main determinant as well as additional B-cells. The quantitation/reproducibility of the findings from Figure 1 would be important in fully establishing the "atlas" relative to these studies in this area

Response: Thank you for your valuable and professional question. We indeed should re-organize and re-analyze our scRNA data more quantifiably. In terms of canonical feature cell markers, eleven major cell clusters were discovered, and cell annotations were subsequently calibrated by evaluating the most highly expressed marker genes among each cell cluster (Supporting Figure 1A). Cell composition analysis showed reduced activated T cells and increased exhausted T cells and B cells in tumors from KL mice compared with those from K mice (Supporting Figure 1B). Activated T cells in KL group were characterized by significantly lower expression of TCF7 relative to K group ($p < 0.001$); while exhausted T cells displayed significantly higher expression of CTLA4 in KL mice versus that of K mice ($p < 0.0001$) (Supporting Figure 1C).

Supporting Figure 1: Quantified cell composition analysis of single cell sequencing data in *Kras*^{mutant}/*Lkb1*^{mutant} (KL) versus *Kras*^{mutant} (K) mice. (A) Heatmap of canonical cell-type markers of 11 major cell types. (B) Absolute numbers of cells from K and KL samples for clusters identified in Figure 1. (C) Expression level of Tcf7 in activated T cells and that of ctla4 in exhausted T cells from lung tumour samples of K and KL mouse. (Data are presented as violin plot (median, 25%-75%, range). Statistical

significance was tested with a two-tailed Mann–Whitney U test. ns, not significant; * $p < .05$; ** $p < .01$; *** $p < .001$; **** $p < .0001$.)

To provide more detailed quantitative analysis of the immune profiling data and to evaluate the general effect of *Lkb1* deficiency on different immune cell clusters, we also calculated gene signature-based scores at the single-cell level in different cell types using single sample gene set enrichment analysis (ssGSEA) (David A Barbie, 2009, nature). And we found that activated T cells isolated from KL tumors showed significantly lower “T cells homing on tumor” and “T cell proliferation in immune response” signature scores (Supporting Figure 2A); While exhausted T cells isolated from KL tumors showed significantly higher “PD-1 pathway”, “T cell anergy” signature scores compared with T cells from K tumors, indicating lower T cell infiltration and activation, and higher T cell dysfunction by *Lkb1* deficiency (Supporting Figure 2B). However, no significant difference in signature scores was observed among the other cell clusters, including B cells (Supporting Figure 2C-F). These data suggest that T-cell activation status is the main determinant downstream signaling of LKB1 in dictating the tumor immune microenvironment.

Supporting Figure 2: Single sample gene set enrichment analysis with immune-related signatures for cells in the (A) activated T cells cluster, (B) exhausted T cells cluster, (C) neutrophils cluster, (D) macrophages cluster, (E) B cells cluster and (F) DCs cluster. (Data are presented as violin plot (median, 25%-75%, range). Statistical significance was tested with a two-tailed Mann–Whitney U test. ns, not significant; * $p < .05$; ** $p < .01$; *** $p < .001$; **** $p < .0001$)

Corresponding changes and explanations we have added in the revised manuscript in the results part were as follows:

Page 5-6: After quality control and cell filtering, we catalogued 14,260 cells into eleven distinct cell lineages annotated with canonical feature cell markers, thus identifying cancer cell types 1, cancer cell types 2, endothelial cells, and immune cells types (including activated T cells, exhausted T cells, B cells, neutrophils, macrophages cells, NK cells, dendritic cells and plasmacytoid dendritic cells) (Fig. 1d, e, f). Cell composition analysis showed reduced activated T cells and increased exhausted T cells in tumors from KL mice compared with those from K mice (Fig. 1f, Supplementary Fig. 1D). Activated T cells in KL group were characterized by significantly lower expression of Tcf7 relative to K group ($p < 0.001$, Fig. 1g, Supplementary Fig. 1E); while exhausted T cells displayed significantly higher expression of Ctla4 in KL mice versus those of K mice ($p < 0.0001$, Fig. 1g, Supplementary Fig. 1E). To further evaluate the general effect of *LKB1* deficiency on different immune cell clusters, we also calculated gene signature-based scores at the single-cell level in different cell types using single sample gene set enrichment analysis (ssGSEA) 22. Cancer cells in KL group showed significantly higher *LKB1* loss signature scores 23, confirming *LKB1* deficiency in KL mice (Supplementary Fig. 1F). And we found that activated T cells isolated from KL tumors showed significantly lower “T cells homing on tumor” signature scores; while exhausted T cells isolated from KL tumors showed significantly higher “PD-1 pathway”, “T cell anergy” signature scores compared with T cells from K tumors (Supplementary Fig. 2A, B). However, no significant difference in signature scores was observed among the other cell clusters, including B cells (Supplementary fig. 2C-F). These data suggest that T-cell activation status is the main determinant downstream signaling of *LKB1* in dictating the tumor immune microenvironment.

2. It is difficult to reconcile the work relative to the tumor immune microenvironment in this study vs. the work of others that have ascribed different mechanisms (e.g. STING) in modulating the immunological milieu and response to immune checkpoint inhibitors.

Response: Thank you so much for your pertinent comments. More insights should be provided in comparisons of the others' work that have investigated into *LKB1* mutant NSCLC. Kitajima et al reported that *LKB1* loss results in suppression of STING, facilitating escape of type I interferon and other STAT1-driven effector programs mediated immune response, raising important mechanistic insights into their resistance to PD-1/PD-L1 ICB (Kitajima, 2018). This excellent study gave us lots of inspirations to discover more mechanisms behind this subtype of lung cancer. We provided additional elements to the current model, in which *LKB1* deficiency leads to ICAM1 mediated disturbance in both adaptive (CD8+ T cells) and innate (NK cells) immunity. And we also put forward a possibly

effective combination strategy for these kinds of lung cancer patients, instead of the STING agonists, which needs more tumor-targeting approaches. Mechanistically, LKB1 deficiency leads to hyperphosphorylated RB, resulting in impaired p65 activation and ICAM1 transcription. Furthermore, we believe that our theory to some extent confirmed with the work of Kitajima, et al. They reported that one possible mechanism for hyperactivation of DNMT1 is because of elevated S-adenylmethionine levels. However, much remains to be understood regarding the mechanism by which DNMT1 is regulated. LKB1 deficiency leads to hyperphosphorylation of RB and thereby activation of E2F1. DNMT1, a recognized E2F target gene, is thus upregulated and plays its role in silencing STING expression, which maybe one possible insightful link. Our work put forward another explanation why LKB1 loss led to DNMT1 upregulation. And we had updated it in the discussion part of our revised manuscript.

Corresponding changes and explanations we have added in the revised manuscript in the results part were as follows:

Page 32: Kitajima et al reported that *LKB1* loss results in suppression of STING, facilitating escape of type I interferon and other STAT1-driven effector programs mediated immune response, raising important mechanistic insights into their resistance to PD-1/PD-L1 ICB²⁰. This excellent study gave us lots of inspirations to discover more mechanisms behind this subtype of lung cancer. Mechanistically, LKB1 deficiency leads to hyperphosphorylated RB, resulting in impaired p65 activation and *ICAM1* transcription. Furthermore, we believe that our theory to some extent confirmed with the work of Kitajima, et al. They reported that one possible mechanism for hyperactivation of DNMT1 is because of elevated S-adenylmethionine levels. However, much remains to be understood regarding the mechanism by which DNMT1 is regulated. *LKB1* deficiency leads to hyperphosphorylation of RB and thereby activation of E2F1. DNMT1, a recognized E2F target gene, is thus upregulated and plays its role in silencing STING expression, which maybe one possible insightful link. Our work put forward another explanation why *LKB1* loss leads to DNMT1 upregulation.

Furthermore, additional elements to the current *LKB1* mutant model were provided, *LKB1* deficiency leads to a disturbance in both innate and adaptive immunity.

3. Given the correlation between ICAM1 and LKB1 the results in Figure 3A/B are confirmatory of the known relationship of LKB1-status and response to immune-checkpoint inhibitor. The increase in CD8+ T-cells is relatively marginal with the increase expression of ICAM1 and there is no evaluation of activation/exhaustion status. If in fact, ICAM1 is the full mechanisms by which LKB1 deficiency acts, it should be possible to shift the totality of the immune profile (e.g., by single cell analysis) back to that of an LKB1 wild-type tumor.

Response: Thank you for your valuable and professional question. We truly agree with you that, more detailed analysis of the functional effects of ICAM-1 overexpression on the TME in the LLC1-shLkb1 system is required. We therefore analyzed absolute and relative fractions of immune cells subsets, including effects on immune cytotoxic cells and myeloid cells. We also have provided more detailed characterization of proliferation, activation and differentiation status of TILs. It was observed that ICAM1 overexpression significantly

increased infiltration level of CD8+ T cells, NK cells, and dendritic cells. Besides, expression level of CD69 (T cell activation) on CD8+ T cells was significantly increased, CD44 expression exerted an enhancing tendency in LKB1 deficient group with ICAM1 overexpression. For the differentiation status of CD8+ T cells (effector memory CD8+ T cells, and central memory CD8+ T cells), no significant changes were observed with ICAM1 overexpression, though an enhancing tendency was displayed in effector memory CD8+ T cells when ICAM1 was augmented (Supporting Figure 3). Taken together, LKB1 deficiency leads to rarity of immune cytotoxic cells, and ICAM1 can enhance infiltration and activation of CD8+ T cells and NK cells, therefore shift the immune profiles back to that of an LKB1 wild-type tumor.

Supporting Figure 3: Detailed analysis of the functional effects of ICAM-1 overexpression on the TME in the LLC1-shLkb1 system. Total CD8+ T cells relative to CD45+ cells in tumor tissue, relative MFI of CD69 and CD44 on CD8+ T cells, percentage of CD62L+CD44+ (central memory T cells) and CD62L-CD44+(effector memory T cells) in CD8+ T cells, and total NK cells relative to CD45+ cells in tumor tissue and relative MFI of CD69 on NK cells, percent of CD45+CD11b+CD103+CD11c cells (DCs), CD45+CD11b+Gr-1+ cells (MDSCs), CD45+CD11b+F4/80+ cells (TAMs) from mice bearing LLC1 tumors (LLC1-control-*Icam1*^{WT}, LLC1-shLkb1-*Icam1*^{WT}, LLC1-shLkb1-*Icam1*^{OE}) were analyzed by flow cytometry. (Results are presented as mean ± SEM. One-way ANOVA and log-rank test was used to analyze the data. ns, not significant; *p<.05; **p<.01; ***p<.001; ****p<.0001)

Corresponding changes and explanations we have added in the revised manuscript in the results part were as follows:

We firstly investigated the functional effects of ICAM-1 reinforcement on the TME in the LLC1-shLkb1 orthotopic lung tumor system. ICAM1 overexpression significantly increased

CD8+ T cell and NK cell infiltration (Fig. 4a). Besides, expression level of CD69 (T cell activation) on CD8+ T cells was significantly increased, and CD44 expression exerted an enhancing tendency. For the differentiation status of CD8+ T cells (effector memory CD8+ T cells, and central memory CD8+ T cells), no significant changes were observed, though an enhancing tendency was displayed in effector memory CD8+ T cells when ICAM1 was augmented (Fig.4a). Taken together, LKB1 deficiency leads to rarity of immune cytotoxic cells, and ICAM1 can enhance infiltration and activation of CD8+ T cells and NK cells.

4. RB knockdown and effect on ICAM1 is difficult to appreciate in Figure 4. Similarly, while it is suggested that RB/CDK4/LKB1 form a complex the data are by no means definitive and just raises the question of what the complex is doing and/or relevance to ICAM1 expression?

Response: Thank you for your valuable and professional suggestion. According to your valuable suggestion, we performed more experiments to clarify this question. We firstly evaluated how LKB1 regulate ICAM1 expression through p65. Both qPCR and flowcytometry analysis demonstrated that LKB1 overexpression significantly restored ICAM1 expression, while knockdown p65 remarkably impaired ICAM1 in LKB1 overexpressed cells (Supporting Figure 4A-B). Furtherly, we performed ChIP analysis and found that there is increased p65 occupancy in the promoter of ICAM1 in *LKB1*-WT group compared to *LKB1*-loss group (*A549-LKB1*-WT vs. *A549-LKB1*-loss) (Supporting Figure 4C), which furtherly demonstrated that LKB1 regulated the transcriptional activity of p65 for the expression of ICAM1. It was observed that by using CDK4/6 inhibitors, pRB was significantly inhibited, and p-p65 was in the meantime upregulated and ICAM1 was therefore increased (supplementary figure 15A). Therefore, p65 was the downstream regulator of RB during ICAM1 regulation. Xin Jin reported that CDK4-phosphorylated RB could suppress transcriptional activity of p65 by specifically binding to p65 (Xin Jin, molecular cell, 2019). Therefore, phosphorylated RB (inactive form of RB) influenced p65 transcription. We found that the binding of RB with p65 was reduced in LKB1 overexpressed A549 cells through CO-IP and IF analysis (*A549-LKB1*-WT versus. *A549-LKB1*-loss) (Supporting Figure 4D-F). Together, LKB1 inhibits RB phosphorylation by competing with CDK4, and therefore impairs the interaction of RB with p65, and the occupancy of p65 on the promoter of ICAM1 is therefore augmented. When RB was knockdown, the interaction between LKB1 and RB was also impaired, LKB1 lost its target to regulate *ICAM1* transcription; therefore, in our model, only when RB is intact, LKB1 competes with CDK4 for the binding of RB, and phosphorylated RB is inhibited, transcriptional activity of p65 is therefore enhanced and *ICAM1* transcription is upregulated. Our final model was revised in Supplementary Figure 21.

Supporting Figure 4: (A-B) Analysis of ICAM1 expression of LKB1-loss cells expressing vector+ siNC, LKB1+siNC, or LKB1+sp65. RT-PCR (A) and flowcytometry (B) analysis of ICAM1 expression. (C) ChIP assay was performed with cell lysates from A549/LV-control and A549/LV-LKB1 cells. A pair of primers flanking the p65 binding site within the ICAM-1 promoter was used in PCR. Real-time PCR was employed to the ChIP assay. (D) Immunoprecipitation analysis of the interaction among RB, and p65 performed in A549 cells (LKB1-loss cells). (E) Multiple immunofluorescence labeling using anti-p65 and anti-RB antibodies in A549-LKB1-loss cells versus A549-LKB1-WT cells. (F) Immunoprecipitation analysis of the interaction among RB, and p65 performed in A549-LKB1-loss cells versus A549-LKB1-WT cells. (Results are presented as mean \pm SEM. One-way ANOVA was performed for multi-group comparisons. t test was performed for comparisons of two groups. ** $p < .01$; **** $p < .0001$)

Corresponding changes and explanations we have added in the revised manuscript in the results part were as follows:

Page 19: p65 is the key transcription factor for *ICAM1*, and we also provided evidence in our analysis (Supplementary Fig. 16A-F). *LKB1* deficiency led to a decreased phosphorylation level of p65 (Supplementary Fig.15E). ChIP analysis demonstrated that p65 occupancy in the promoter of *ICAM1* is increased in *LKB1*-WT group compared to *LKB1* loss group (Fig. 5j), which furtherly demonstrated that *LKB1* regulates the transcriptional activity of p65 for the expression of *ICAM1*. In agreement with this, p65 knockdown impaired *ICAM1* expression in *LKB1* restored group (Supplementary Figure 16G-H). CDK4-phosphorylated RB specifically binds to p65 and inhibits its transcriptional activity²⁹. We observed that *LKB1* reconstitution interfered RB-p65 interactions (Supplementary Figure 16I-J).

Together, these results suggested that *LKB1* deficiency led to hyper-phosphorylated RB, which hindered p65 transcriptional activity and *ICAM1* expression. The dephosphorylation of RB by CDK4/6 inhibitors rescued *ICAM1* transcription and thus might enhance the activation of cytotoxic cells.

5. There is an extensive literature that CDK4/6 inhibitors will augment the response to immune checkpoint inhibition via multiple mechanisms. These are known to be RB-dependent. The work herein would seem to be at odds with work from other labs that CDK4/6 inhibitors alone are insufficient to have a potent impact on immune-checkpoint

inhibitors or the tumor microenvironment in lung cancer models. More in-depth analysis would be important to illustrate that these effects are ICAM1-dependent in the tumor compartment.

Response: Thank you for your valuable and professional suggestion.

1) More interpretations of the effects of CDK4/6 inhibitors (CDK4/6i) alone on the tumor microenvironment (TME) in lung cancer models and on immune-checkpoint inhibitors should be provided. Based on our data, CDK4/6 inhibitors can have a positive effect on revitalizing the TME, with increase in CD8+ T cells and NK cell infiltration. Besides, activity of CD8+ T cells were significantly higher with the CDK4/6i treatment than with vehicle, as reflected by elevated interferon- γ and decreased PD-1 levels (figure 6d in the revised manuscript). However, for LKB1 mutant lung cancer, as the “cold” tumor, CDK4/6i and PD-1 blockade combination can have a more synergistic effect by potently triggering an active TME (figure 6d-e in the revised manuscript), which we believe is in some way corroborating the work of others that CDK4/6 inhibitors are powerful and important, especially for “immune cold” tumors.

2). Thank you so much for your valuable suggestions. Indeed, more in depth analysis would be important to illustrate the effects of CDK4/6 inhibition and the combination therapy are ICAM1-dependent in our tumor compartment. Therefore, according to your precious suggestions, we have established isogenic derivatives of the LLC1-sh*Lkb1* cell line with or without knockout of *Icam1* using CRISPR/Cas9 and repeated the therapeutic experiment. And we found that antitumor effects of the combined PD-1 blockade and CDK4/6 inhibition were significantly impaired in LLC1-sh*Lkb1*-sg/*cam1* tumors (Supporting Figure 5A).

Accordingly, we systemically evaluated the effects of treatment with palbociclib or the combination therapy on the tumor microenvironment in the setting of ICAM-1 depletion (LLC1-sh*Lkb1*-sg/*cam1* vs. LLC1-sh*Lkb1*-sgCon). We found that LLC1-sh*Lkb1*-sg/*cam1* tumors exhibited a significant reduction in CD8+ T cell infiltrate, and CD69, CD44 expression on CD8+ T cells compared with LLC1-sh*Lkb1*-sgCon tumors after palbociclib treatment or combination therapy. Furthermore, infiltration of NK cells and DCs were significantly reduced in LLC1-sh*Lkb1*-sg/*cam1* tumors after combination therapy compared with that of LLC1-sh*Lkb1*-sgCon tumors. Interestingly, MDSCs and tumor-associated neutrophils (TAN) infiltration were significantly increased in the setting of ICAM-1 depletion after treatment with palbociclib or the combination therapy (Supporting Figure 5B-C). Taken together, CDK4/6 inhibitors or the combination therapy induced an immune-active tumor microenvironments, which is dependent on ICAM1.

Supporting Figure 5: ICAM-1 depletion abrogates anti-tumor immunity conferred by the Palbociclib/ICI combination therapy. (A) C57BL/6 mice were inoculated with LLC1/shLkb1 with or without *Icam1* knock-out cells and received the combination treatments versus vehicle. Tumor size were monitored. (B) Representative immunohistochemistry images of ICAM1 and CD8 α in LLC1-shLkb1/sgCon and LLC1-shLkb1/sgIcam1 tumors subjected to indicated treatment. Scale bar, 50 μ m. (C) Flowcytometry analysis of immune cell infiltration (percent of CD45+CD3+CD8+ cells, CD45+CD3-NK1.1+ cells, CD45+CD11b+CD11c+CD103+ cells, CD45+CD11b+Gr-1+ cells, CD45+CD11b+CXCR2+ cells, CD45+CD11b+F4/80+ cells), expression of CD69, CD44 on CD8+ T cells and CD107a on NK cells in LLC1-shLkb1/sgCon and LLC1-shLkb1/sgIcam1 tumors followed by indicated treatment arm. (Results are presented as mean \pm SEM. One-way ANOVA was performed for multi-group comparisons. n.s., not significant; *p<.05; **p<.01; ***p<.001; ****p<.0001)

Corresponding changes and explanations we have added in the revised manuscript in the results part were as follows:

Page 28: Subsequently, isogenic derivatives of the LLC1-shLkb1 cell line with or without knockout of *Icam1* using CRISPR/Cas9 were established. The mice received the same combinational therapy. And we found that antitumor effects of the combined PD-1 blockade and CDK4/6 inhibition were significantly impaired in LLC1-shLkb1-sgIcam1 tumors (Fig. 8g).

Accordingly, we systemically evaluated the effects of treatment with palbociclib or the combination therapy on the tumor microenvironment in the setting of ICAM-1 depletion. IHC staining of tumor tissue displayed reduced ICAM1 expression and CD8+ T cell infiltration in LLC1-shLkb1-sgIcam1 tumors compared with LLC1-shLkb1-sgCon tumors

after palbociclib treatment or combination therapy (Fig. 8h, supplementary Fig. 21A). Flow cytometry analysis demonstrated that LLC1-sh*Lkb1*-sg/*cam1* tumors exhibited a significant decrease in CD8+ T cell infiltrate, and CD69, CD44 expression on CD8+ T cells versus LLC1-sh*Lkb1*-sgCon tumors after CDK4/6 inhibition or combination therapy. Furthermore, infiltration of NK cells and DCs were significantly lower in LLC1-sh*Lkb1*-sg/*cam1* tumors after combination therapy versus that of LLC1-sh*Lkb1*-sgCon tumors. Taken together, these depletion therapies reversed the advantages conferred by the palbociclib/ICI combination therapy, confirming that CDK4/6 inhibition promotes the effect of anti-PD-1 immunotherapy in an ICAM1-dependent manner (Fig. 8h, supplementary Fig.21A-B).

REVIEWER COMMENTS

Reviewer #1 (Remarks to the Author):

The authors have performed a substantial amount of new experiments in response to comments from the reviewers and the study has been strengthened substantially. I recommend acceptance for publication.

Reviewer #2 (Remarks to the Author):

the authors have adequately addressed all my comments

Reviewer #3 (Remarks to the Author):

All my concerns have been well addressed.

Reviewer #4 (Remarks to the Author):

Critique: I appreciate that the authors have performed a number of additional experiments for the revised manuscript. These largely address questions related to cause/effect relationships between LKB1 and ICAM1, as well as immune cell functionality. However, a number of relatively substantive issues were not addressed.

Relative to the immune context in the KL vs. K model Figure 1. This represents (as previously stated) a single snapshot that could just be a reflection of tumor heterogeneity. Additionally, the effects RE TCF7/CTLA4 (Fig 1G) while significant are relatively marginal. I worry about reproducibility, as the data in Figure S1 show a complete lack of CD8+ cells which is not reflective of the single cell data? How variant is the data across multiple tumors?

Relative to the mechanism of action with Palbociclib and RB. The data in Figure 5 (as previously stated) are not particularly compelling. Particularly because RB is a nuclear protein and LKB1 is cytoplasmic/membrane associated. The data in Figure S15 similarly has some challenges, most notably RB is a nuclear protein, but is shown as cytoplasmic. RB is associated with a huge number of proteins and in the absence of some clear functionality the binding data is difficult to assess particularly as RB is interacting with both LKB1 and P65. Data in Figure S16 again has RB expressed in the cytoplasm.

The data in Figure 6 and S18 which contrasts LKB1 wild-type and over-expressing models with LKB1 does not make sense if ICAM1 is the determinant of sensitivity to immune checkpoint inhibition?

A key feature for the final model of the paper (Figure S22) is the mechanism through which CDK4/6, RB, LKB1, and p65 control ICAM1. I do not think the presented data strongly support the model as was stated in the prior review.

Reviewer #4 (Remarks to the Author)

Critique: I appreciate that the authors have performed a number of additional experiments for the revised manuscript. These largely address questions related to cause/effect relationships between LKB1 and ICAM1, as well as immune cell functionality. However, a number of relatively substantive issues were not addressed.

1. Relative to the immune context in the KL vs. K model Figure 1. This represents (as previously stated) a single snapshot that could just be a reflection of tumor heterogeneity. Additionally, the effects RE TCF7/CTLA4 (Fig 1G) while significant are relatively marginal. I worry about reproducibility, as the data in Figure S1 show a complete lack of CD8+ cells which is not reflective of the single cell data? How variant is the data across multiple tumors?

Response: Thank you so much for your kind and pertinent suggestions. We indeed agree with you that our single cell sequencing data was a snapshot of the tumor immune microenvironment of KL mouse model versus that of K mouse model. Therefore, we admit there were limitations that quantitative comparisons were significant but marginal between *Tcf7* and *Ctla4*. For the data in figure S1, we had selected the representative view of CD8+ cells. Here, the images of CD8 IHC staining in *Kras*^{mutant}*Lkb1*^{mutant} (KL) group and that of *Kras*^{mutant} (K) group were presented (**Supporting Figure 1A**). There indeed were less CD8+ cells in KL group versus that of K group, and what's more, the KL mice were not completely lack of CD8 T cells in its tumor immune microenvironment. Recently, Huiyu et al. also reported that lacking of TCF1+ CD8 T cells results in poor response of *STK11* mutant NSCLC to anti-PD-1 immunotherapy¹ (cell reports medicine, 2022), which also offered evidences for our data. And in their work, there also existed some TCF1+ CD8 T cells in KPL group. We re-analyzed their data and similarly, we discovered CD8 T cell subpopulations in KPL group (**Supporting Figure 1B**). To further demonstrate the reproducibility of our data, we exploited single cell sequencing data of two KL samples from GEO database (GSE165641), and equally, there still existed T cells in KL mouse samples (**Supporting Figure 1C**). In addition to that, IHC staining of CD8 in patient-derived NSCLC samples showed that, CD8 T cells were less but not completely loss in *LKB1* mutant lung cancer patients (supplementary figure 12B).

Therefore, we believe the consistency among multiple KL tumors, that *LKB1* mutant lung cancer is lacking of CD8 T cells, but not completely devoid of CD8 T cells.

Supporting Figure 1: (A) Representative images of IHC staining of *Kras*^{mutant}*Lkb1*^{mutant} and *Kras*^{mutant} lung tumors using anti-CD8 α antibodies. Scale bar, 100 μ m (20 \times). (B) UMAP plot for the expression of *Ptprc*, *Cd3g*, *Cd4* and *Cd8a* in the scRNA-seq datasets of KP versus KPL mice (Huiyu Li, 2022). (C) tSNE plot for the cells from two KL lung tumor samples, colored by their 6 major cell types (GSE165641).

2. Relative to the mechanism of action with Palbociclib and RB. The data in Figure 5 (as previously stated) are not particularly compelling. Particularly because RB is a nuclear protein and LKB1 is cytoplasmic/membrane associated. The data in Figure S15 similarly has some challenges, most notably RB is a nuclear protein, but is shown as cytoplasmic. RB is associated with a huge number of proteins and in the absence of some clear functionality the binding data is difficult to assess particularly as RB is interacting with both LKB1 and P65. Data in Figure S16 again has RB expressed in the cytoplasm.

Response: Thank you so much for your professional and pertinent questions which inspires us a lot. We truly agree with you that in the classical model, RB localized mostly in the cell nuclear. Whereas, possibly, RB can also present in the cytoplasm, as evidenced by [GENECARDS database \(https://www.genecards.org/cgi-bin/carddisp.pl?gene=RB1&keywords=RB#localization\)](https://www.genecards.org/cgi-bin/carddisp.pl?gene=RB1&keywords=RB#localization) and [COMPARTMENTS database \(https://compartments.jensenlab.org/Entity?figures=subcell_cell_%&knowledge=10&textmining=10&predictions=10&type1=9606&type2=-22&id1=ENSP00000267163\)](https://compartments.jensenlab.org/Entity?figures=subcell_cell_%&knowledge=10&textmining=10&predictions=10&type1=9606&type2=-22&id1=ENSP00000267163). M Macaluso et al compared the cytoplasmic fractions of RB protein with that of nuclear fractions (Cell Death and Differentiation, 2006)². We also observed that RB presented both in the cell cytoplasm and the nucleus in our model (**Supporting Figure 2**).

Supporting Figure 2: Western blotting of cytoplasmic and nuclear fractions of certain proteins (RB, and LKB1) in vector, LKB1-OE, CDK4-OE, and the palbociclib groups.

Furthermore, the binding of RB with LKB1 and the binding of RB with p65 results in different functionalities.

Firstly, for LKB1-RB binding, we had demonstrated that LKB1 competes with CDK4 for the binding of RB (figure 5h, supplementary figure 15I-J), and RB is thus dephosphorylated. Qingyu Guo also reported that ARID1A competes with CDKs, and blocks the interaction between CDKs and RB, reducing the phosphorylation of RB (Cell Death and Differentiation, 2019)³. Unphosphorylated RB is in its active form. In conclusion, the binding of LKB1 with RB can affect the phosphorylation of RB by CDK4. Since we deduce that LKB1 and CDK4 have similar binding site on RB protein, we constructed the RB protein variant with truncation of this mutual site (RB-del) (**Supporting Figure 3A**). Immunoprecipitation assay of LKB1 and CDK4 with RB variant were both decreased. (**Supporting Figure 3B**).

Supporting Figure 3: (A) Schematic diagram of RB truncation mutation; RB protein variant with truncation of the RB C terminus (Rb^{ΔC-term}) was constructed (RB-Del). (B) RB-WT and RB-Del immunoprecipitates with LKB1 and CDK4, respectively.

While when LKB1 is defected, RB is hyperphosphorylated by CDK4. It was reported that CDK4-mediated phosphorylation of RB enables an RB-NF- κ B interaction to inhibit p65 from playing its transcriptional function, indicating that phosphorylation of RB by CDK4 can direct protein interactions (molecular cell, 2019; molecular cell previews, 2019)^{4, 5}. RB-p65

interaction depends on CDK4 mediated S249/T252 phosphorylation of RB⁴. We therefore constructed RB S249/T252 phosphorylation resistant mutant (S249A/T252A) plasmids. Subsequent co-IP assay displayed that RB mutant (S249A/T252A) markedly disturbed RB-p65 interaction (**Supporting Figure 4**). When LKB1 was overexpressed or CDK4/6 inhibitors were used, RB-p65 interaction was observably decreased even when wild type RB was overexpressed (**Supporting Figure 4**). CDK4 overexpression prominently enhanced RB-p65 interaction in wild-type RB group, but no such effect was observed for RB S249A/T252A group (**Supporting Figure 4**).

Supporting Figure 4: RB-WT and RB-mutant (S249A/T252A) co-immunoprecipitates with p65 in the vector, LKB1-overexpression, CDK4-overexpression and the palbociclib group.

For *LKB1* mutant cancer cells, CDK4 over-phosphorylates RB, and thus enables an RB-p65 interaction (supplementary figure 16J-L), and the transcriptional activity of p65 is inhibited (Figure 5j). We had demonstrated that *ICAM1* is the direct transcriptional product of p65, so *ICAM1* is downregulated when *LKB1* is defected (Figure 5j).

3. The data in Figure 6 and S18 which contrasts LKB1 wild-type and over-expressing models with LKB1 does not make sense if *ICAM1* is the determinant of sensitivity to immune checkpoint inhibition?

Response: Thank you so much for your valuable and professional suggestion. The reason why we performed the experiments is to fully demonstrate LKB1 status on the immunomodulatory effects and sensitivity to ICI and the combination therapy. Whether LKB1 is defected or not determines lung cancer patients' sensitivity to immune checkpoint inhibition, therefore we focus on LKB1-defected models, which is of much clinical value (LKB1 is the third most frequently mutated gene in LUAD).

From this, we discovered that combined PD-1 blockade and CDK4/6 inhibition significantly slowed tumor growth in both the LLC1-*Lkb1*^{WT} and LLC1-*Lkb1*^{OE} group (Supplementary figure 18C-D). However, the synergistic effects of the combination therapy were diluted in comparison with LLC1-*Lkb1*^{KD} group (LLC1-sh*Lkb1*) (Fig. 6a-g), especially for mice with LKB1 overexpression (Supplementary figure 18D). Immune checkpoint

inhibition alone already exhibited significant anti-tumor effect against LLC1-*Lkb1*^{OE} tumors, implying that LKB1-mutant/deficient NSCLC is uniquely sensitive to the synergistic effect of palbociclib and anti-PD-1 combination therapy.

Theoretically, ICAM1 level elevated in LLC1-*Lkb1*^{OE} tumors versus that of LLC1-*Lkb1*^{WT} tumors (which can be evidenced by figure 2g). Furthermore, ICI sensitivity was increased in LLC1-*Lkb1*^{OE} group (ICAM1 level elevated) since anti-PD-1 mAb treatment alone already exhibited significant anti-tumor effect against LLC1-*Lkb1*^{OE} tumors ($p=0.0038$), which was not existed in LLC1-*Lkb1*^{WT} tumors ($p=0.277$) (supplementary figure 18), or LLC1-*Lkb1*^{KD} tumors ($p=0.5737$) (Fig. 6a), implying that ICAM1 plays a role in the determinant to ICI sensitivity.

In addition to that, overexpression of *Icam1* sensitized *Lkb1*-deficient tumors to anti-PD-1 immunotherapy as better tumor control (Fig. 4b-c). Besides, ICAM1 overexpression-initiated response to immunotherapy was also observed in the subcutaneous tumor model (Fig. 4d). Furthermore, cancer cell-intrinsic knock-out of *Icam1* reduces tumors responsiveness to immunotherapy (Supplementary Fig. 13A-D). Taken together, these data illustrated that ICAM1 overexpression sensitized LKB1 deficient lung tumors to anti-PD-1 immunotherapy. We also have established isogenic derivatives of the LLC1-sh*Lkb1* cell line with or without knockout of *Icam1* using CRISPR/Cas9 (Fig. 8g) and repeated the therapeutic experiment. And we found that antitumor effects of the combined PD-1 blockade and CDK4/6 inhibition were significantly impaired in LLC1-sh*Lkb1*-sg*Icam1* tumors (Fig. 8g), which demonstrate that ICAM1 is the determinant of sensitivity to the combination therapy and also the ICI therapy.

We also demonstrated ICAM1 level affects patients' sensitivity to ICI therapy. We firstly assessed the prognostic value of ICAM1 across lung adenocarcinoma and squamous carcinoma patients from TCGA, and found that higher ICAM1 is not correlated with good prognosis of lung adenocarcinoma patients from TCGA (HR=1.01, $p=0.87$), or even correlated with poor prognosis in lung squamous carcinoma (HR=1.15, $p=0.0265$), which indicates that it is not a good prognostic marker (supplementary figure 14B-C). However, patients with higher ICAM1 expression had longer PFS (progression free survival) in datasets of immunotherapy-treated patients (melanoma, GSE93157 and Liu 2019; NSCLC, GSE126044 and GSE135222; melanoma, Gide 2019) (Figure 4e-f, supplementary figure 14A).

Therefore, the data in figure 6 and S18 may not reflect directly but to some extent indicates that ICAM1 plays a role in the determinant to ICI sensitivity. And we had also demonstrated the functional role of ICAM1 to ICI sensitivity in several other experiments.

4. A key feature for the final model of the paper (Figure S22) is the mechanism through which CDK4/6, RB, LKB1, and p65 control ICAM1. I do not think the presented data strongly support the model as was stated in the prior review.

Response: Thank you so much for your professional and pertinent suggestions. We recently had added additional evidences relating to the interaction between LKB1 with RB, and RB with p65. We also provided more details of the CHIP experiments which we think may increase substantive mechanism to our final model.

Firstly, we had verified that LKB1 regulates *ICAM1* transcription through p65 (Fig. 2g-h, supplementary figure 7C-G, Supplementary figure 15G, supplementary figure 16A-H). We performed ChIP analysis and found that there is increased p65 occupancy in the promoter of *ICAM1* in LKB1-WT group compared to LKB1-loss group (Fig. 5j), which furtherly demonstrated that LKB1 regulates the transcriptional activity of p65 and leads directly to *ICAM1* transcriptional change. (**LKB1-p65-ICAM1**)

Secondly, we had evidenced that LKB1 competes with CDK4 for the binding of RB (Fig.5h, supplementary figure 15I-K), and thus regulates RB phosphorylation (**LKB1-CDK4-RB**). CDK4 is widely proved to bind RB within the C domain of RB⁶, in which C-terminal alpha-helix docking was regarded as a major driver of RB phosphorylation by CDK4⁴. To validate our hypothesis of competitive binding between CDK4 and LKB1 protein on RB, we harnessed the PepSite (<http://pepsite2.russelllab.org/>), a bioinformatics tool to predict the possibility of LKB1 docking on the C-terminal helix motif of RB (supplementary figure 15I). The result showed that LKB1 has the potential to bind RB on the C-terminal helix motif with statistical significance. Inspired by this finding, we further constructed a plasmid to generate an RB protein variant with truncation of the RB C terminus (RB Δ C-term) to cover the possible binding site comprehensively (**Supporting Figure 3A**). We utilized this RB variant (Δ C-term) to investigate whether LKB1 competitively docks on the C domain of RB with CDK4. Immunoprecipitation assay of LKB1 and CDK4 with RB variant were both decreased. (**Supporting Figure 5A, same with supporting figure 3**)

We adopted a combined drug screening strategy and results revealed that palbociclib increased the expression of *ICAM1* remarkably in *LKB1* mutant cells at both transcriptional and protein levels (Fig. 5a-d). It was observed that by using CDK4/6 inhibitors, pRB was inhibited, and p-p65 was upregulated and *ICAM1* was therefore enhanced (supplementary figure 15A). Therefore, p65 was the downstream regulator of RB during *ICAM1* regulation. CDK4-mediated phosphorylation of RB enables an RB-NF- κ B interaction to inhibit p65 from playing its transcriptional function⁴. We found that the binding of RB with p65 was reduced in LKB1 overexpressed A549 cells through Co-IP and IF analysis (A549-LKB1-WT versus. A549-LKB1-loss) (supplementary figure 16J-L). RB-p65 interaction depends on CDK4 mediated S249/T252 phosphorylation of RB⁴. We therefore constructed RB S249/T252 phosphorylation resistant mutant (S249A/T252A) plasmids. The following co-IP assay demonstrated that RB mutant (S249A/T252A) plasmids markedly disturbed RB-p65 interaction (**Supporting Figure 5B, same with supporting figure 4**). When LKB1 was overexpressed or CDK4/6 inhibitors were used, RB-p65 interaction was observably decreased even when RB wild type plasmids were overexpressed (**Supporting Figure 5B**). CDK4 overexpression prominently enhanced RB-LKB1 interaction in RB wild type group, but no such effect was observed for RB S249A/T252A group (**Supporting Figure 5B**).

Thus, LKB1 blocks CDK4-phosphorylated RB (S249/T252 phosphorylation), p65 is thus released and transactivates *ICAM1*. (**LKB1-RB-p65-ICAM1**).

Our previous ChIP analysis demonstrated that p65 occupancy in the promoter of *ICAM1* is increased in LKB1-WT group versus LKB1 loss group (Fig. 5j). Besides, we furtherly demonstrated that when CDK4 was overexpressed in LKB1 loss cells, RB was over-phosphorylated, therefore p65 occupancy was significantly reduced in compared to that of

the vector group ($p=0.0382$) (**Supporting Figure 5C-D**). Furthermore, CDK4/6 inhibitors treatment significantly decreased RB phosphorylation (the same as that of LKB1 overexpression), thus enhanced p65 occupancy versus that of the vector group ($p=0.0170$) (**Supporting Figure 5C-D**). These four groups can reflect our final model.

Therefore, LKB1 hinders CDK4-phosphorylated RB, while when RB is phosphorylated by CDK4, it can bind with p65. RB-p65 interaction relies on CDK4 mediated S249/T252 phosphorylation of RB⁴. Therefore, p65 cannot play its transcriptional function. From our additional ChIP analysis, we can see that p65 occupancy is thus significantly decreased in CDK4 overexpressed group.

We believe that from others' work and our previous and additional data, and with the help of previous suggestions from reviewers, the main mechanisms may be supported.

Supporting Figure 5: (A) RB-WT and RB-Del immunoprecipitates with LKB1 and CDK4, respectively.

(B) RB-WT and RB-mutant (S249A/T252A) co-immunoprecipitates with p65 in the vector, LKB1-overexpression, CDK4-overexpression and the palbociclib group. (C) Western blotting of certain proteins (LKB1, CDK4, RB, pRB, p65, p-p65, and ICAM1) in the vector, LKB1-OE, CDK4-OE and Palbociclib groups. (D) ChIP assay was performed with cell lysates from A549/vector, A549/LKB1-OE, A549/CDK4-OE cells and A549/vector with palbociclib. A pair of primers flanking the p65 binding site within the *ICAM1* promoter was used in PCR. Real-time PCR was employed to the ChIP assay. (Results were presented as mean \pm SEM. One-way ANOVA was used to analyze the data. * $p<.05$, *** $p<.001$)

Corresponding changes and explanations we have added in the revised manuscript in the results part were as follows:

Page 19: Inspired by this finding, we further constructed a plasmid to generate an RB protein variant with truncation of the RB C terminus (RB Δ C-term) to cover the possible binding site comprehensively. We utilized this RB variant (Δ C-term) to investigate whether LKB1 competitively docks on the C domain of RB with CDK4. Immunoprecipitation assay of LKB1 and CDK4 with RB variant were both decreased (Supplementary Fig. 15L).

Page 20: Chromatin Immunoprecipitation (ChIP) analysis demonstrated that p65 occupancy in the promoter of *ICAM1* is increased in LKB1-WT group compared to LKB1 loss group (Fig. 5j), which furtherly demonstrated that LKB1 regulates the transcriptional activity of p65 for the expression of ICAM1. In agreement with this, p65 knockdown impaired ICAM1 expression in LKB1 restored group (Supplementary Figure 16G-H). While when CDK4 was overexpressed in LKB1 loss cells, RB was over-phosphorylated (Supplementary Figure 16I); therefore, p65 occupancy was significantly reduced in compared to that of the vector group (p=0.0382) (Fig. 5j). Furthermore, CDK4/6 inhibitors treatment decreased RB phosphorylation (the same as that of LKB1 overexpression) (Supplementary Figure 16I), thus enhanced p65 occupancy versus that of the vector group (p=0.0170) (Fig. 5j).

CDK4-phosphorylated RB specifically binds to p65 and inhibits its transcriptional activity²⁹. We observed that LKB1 reconstitution interfered RB-p65 interactions (Supplementary Figure 16J-L). RB-p65 interaction depends on CDK4/6 S249/T252 phosphorylation of RB29. We therefore constructed RB S249/T252 phosphorylation resistant mutant (S249A/T252A) plasmids. The following co-IP assay demonstrated that RB mutant (S249A/T252A) plasmids markedly disturbed RB-p65 interaction (Supplementary Figure 16M). When LKB1 was overexpressed or CDK4/6 inhibitors were used, RB-p65 interaction was observably decreased even when wild type RB was overexpressed (Supplementary Figure 16M). CDK4 overexpression prominently enhanced RB-p65 interaction in RB wild-type group, but no such effect was observed for RB S249A/T252A group (Supplementary Figure 16M). Together, LKB1 blocks CDK4-phosphorylated RB (S249/T252 phosphorylation), p65 is thus released and transactivates ICAM1.

References:

1. Li H, *et al.* AXL targeting restores PD-1 blockade sensitivity of STK11/LKB1 mutant NSCLC through expansion of TCF1(+) CD8 T cells. *Cell Rep Med* **3**, 100554 (2022).
2. Macaluso M, *et al.* Cytoplasmic and nuclear interaction between Rb family proteins and PAI-2: a physiological crosstalk in human corneal and conjunctival epithelial cells. *Cell Death Differ* **13**, 1515-1522 (2006).
3. Luo Q, *et al.* ARID1A prevents squamous cell carcinoma initiation and chemoresistance by antagonizing pRb/E2F1/c-Myc-mediated cancer stemness. *Cell Death Differ* **27**, 1981-1997 (2020).
4. Jin X, *et al.* Phosphorylated RB Promotes Cancer Immunity by Inhibiting NF-kappaB Activation and PD-L1 Expression. *Mol Cell* **73**, 22-35 e26 (2019).
5. Kim SJ, Asfaha S, Dick FA. CDK4 Inhibitors Thwart Immunity by Inhibiting Phospho-RB-NF-kappaB Complexes. *Mol Cell* **73**, 1-2 (2019).
6. Topacio BR, *et al.* Cyclin D-Cdk4,6 Drives Cell-Cycle Progression via the Retinoblastoma Protein's C-Terminal Helix. *Mol Cell* **74**, 758-770.e754 (2019).

REVIEWER COMMENTS

Reviewer #5 (Remarks to the Author):

The authors have provided additional experimental evidence to support their major conclusions during the last round of revision. However, the following concerns should be addressed.

- 1). Figure 2f: The molecular mechanism of how loss of LKB1 suppresses ICAM1 expression should be further strengthened. In current model, LKB1 can compete with Cdk4 to interact with Rb, and Rb phosphorylation status affects its binding to p65, which is an upstream transcriptional activator of ICAM1. It will be nice for the authors to devote a cartoon to clarify it.
- 2). Figure 2g: the authors showed that LKB1 can affect ICAM1 transcription in a kinase dependent manner, thus it will be important for the authors to show if LKB1 can phosphorylate Rb and whether LKB1 downstream target AMPK involves in regulating ICAM1 transcription? Will depleting AMPK affect ICAM1 mRNA levels?
- 3) Figure 5a, it will be important for the authors to show if other validated CDK4/6 inhibitors can also cause similar phenotypes, to avoid possible off-target effects of Palbociclib.
- 4). Figure 5g: the effects of upregulation in ICAM1 abundance by Palbociclib is moderate and not convincing.
- 5). Figure 5i: authors should provide additional insights into the molecular mechanism of how LKB1 can regulate pRB. It is known that Rb phosphorylation status affects its ability to interact with its interactors. It will be nice for the authors to examine i) if phosphor-deficient Rb (13A mutant) or phosphor-mimetic (13E mutant) Rb display differences in interacting with LKB1, and ii). If Cdk4/6 inhibitor can affect LKB1 binding with Rb and iii) If LKB1 or its downstream kinase can phosphorylate Rb?
- 6). Supplemental Fig S16M, it will be critical to provide solid evidence to validate p65 as an upstream transcriptional activator for ICAM1. i) the authors should show that depleting p65 can reduce ICAM1 transcription, and ii) using ChIP assay to show p65 localize in the promoter region of ICAM1 and iii) ICAM1 possess p65 binding motifs in its promoter region.

Reviewer #5 (Remarks to the Author):

The authors have provided additional experimental evidence to support their major conclusions during the last round of revision. However, the following concerns should be addressed.

1). Figure 2f: The molecular mechanism of how loss of LKB1 suppresses ICAM1 expression should be further strengthened. In current model, LKB1 can compete with Cdk4 to interact with Rb, and Rb phosphorylation status affects its binding to p65, which is an upstream transcriptional activation of ICAM1. It will be nice for the authors to devote a cartoon to clarify it.

Response: Thank you so much for your professional and pertinent questions, which inspire us a lot. For the molecular mechanism of how loss of LKB1 suppresses ICAM1 expression, we first demonstrated that ICAM1 protein levels were either significantly downregulated or completely undetectable in *LKB1* mutant cell lines (Supporting Figure 1A). We also observed that the introduction of LKB1 into *LKB1*-deficient cell lines (A549, H460, H1944) caused an upregulation of ICAM1 (Supporting Figure 1B-C), while knockdown of LKB1 in *LKB1*-intact cell lines (H1299, Calu1, LLC1) led to reduced expression of ICAM1 (Supporting Figure 1D). These findings confirmed that loss of LKB1 suppresses ICAM1 expression.

Moreover, overexpression of LKB1 was found to increase the occupancy of p65, a critical transcription factor for ICAM1, in the promoter of ICAM1, thus demonstrating that LKB1 regulates the transcriptional activity of p65 for the expression of ICAM1 (Supporting Figure 1E). Further, we observed that LKB1 impacts p65 through interacting with RB. Specifically, LKB1 bound with RB and interfered with the RB-CDK4 interaction (Supporting Figure 1F-G), promoting p65-mediated upregulation of ICAM1. When RB-LKB1 binding was impaired by overexpression of CDK4, the occupancy of p65 on the ICAM1 promoter was reduced, which was rescued by palbociclib (Supporting Figure 1E). Intriguingly, we found that ectopic expression of kinase-dead LKB1 protein failed to upregulate the expression of ICAM1 (Supporting Figure 1B-C). It was due to the weakened binding between LKB1 and RB protein (Supporting Figure 1F-G).

Taken together, we demonstrated that LKB1 competes with CDK4 to bind RB, and subsequently decreases CDK4-induced phosphorylation of RB, thus hindering the binding between pRB and p65. Released p65 functions as a transcription factor and positively regulates the expression of ICAM1. In addition to that, we also designed a cartoon to clarify it (Supporting Figure 1H).

Supporting figure 1. (A) Immunoblot of the indicated proteins in lung cancer cells with LKB1-loss or LKB1-intact. (B) A549 and H460 lung cancer cells were transfected with lentivirus expressing the indicated genes (Ctrl, LKB1-WT, and LKB1-Mut). ICAM1 expression was analyzed by immunoblot. (C) Immunoblot analysis of ICAM1 expression in H1944 cells expressing WT (LKB1-WT), mutated (kinase-dead mutation; LKB1-MUT), or no LKB1 (LKB1 control). (D) Immunoblot showing ICAM1 expression of lung cancer cell lines (CALU1, H1299 and LLC1) treated with or without small interference or short hairpin RNA mediated knocking down of LKB1/Lkb1. (E) ChIP assay was performed with cell lysates from A549/vector, A549/LKB1-OE, A549/CDK4-OE and A549/vector + Palbociclib cells. A pair of primers flanking the p65 binding site within the ICAM1 promoter were used in PCR. (F) Immunoprecipitation analysis of the

relationship among LKB1, RB, and CDK4, using anti-LKB1 or anti-CDK4 or anti-RB antibodies in A549 cells with or without ectopic LKB1-WT and LKB1-MUT overexpression. (G) Immunoprecipitation analysis of the relationship among LKB1, RB, and CDK4, using anti-flag or anti-myc magnetic beads in H1299 cells with or without ectopic flag-LKB1 and myc-CDK4 overexpression. (H) A schematic of the mechanism through which mutant LKB1 downregulates ICAM1 transcription and expression in a phosphorylated RB-dependent manner. (Results are presented as mean \pm SEM. Student's t-test or one-way ANOVA was performed for comparisons. n.s., not significant; ** $p < .01$, **** $p < .0001$)

2). Figure 2g: the authors showed that LKB1 can affect ICAM1 transcription in a kinase dependent manner, thus it will be important for the authors to show if LKB1 can phosphorylate Rb and whether LKB1 downstream target AMPK involves I regulating ICAM1 transcription? Will depleting AMPK affect ICAM1 mRNA levels?

Response: Thank you so much for your kind and pertinent suggestions. Here we have shown that LKB1 overexpression decreased phosphorylation level of RB. Analysis of the reverse-phase protein arrays (RPPA) data of TCGA-LUAD cohort demonstrated that LKB1 mutation was correlated with higher (instead of lower) phosphorylated level of RB protein (Supporting Figure 2A). Western blotting further demonstrated that LKB1 deficiency, either via complete loss (+vector) or a kinase-dead mutation (+LKB1-MUT), resulted in an upregulation of phosphorylated RB (pRB) (Supporting Figure 2B). Additionally, previous research has revealed that the wide-type *Lkb1*-expressing squamous cell carcinoma cells had augmented levels of the hypo-phosphorylated RB and reduced levels of the hyper-phosphorylated RB^[1], corroborating our findings. Therefore, our results showed that LKB1 did not directly phosphorylate RB. Instead, LKB1 decreased the level of phosphorylated RB.

To gain further insight into how LKB1 reduces RB phosphorylation, we conduct an additional investigation. Our MS analysis identified RB as one of the immunoprecipitated candidates (Supporting Figure 2C), prompting us to focus on the LKB1-RB interaction. Immunofluorescent staining assay displayed that LKB1 and RB were mainly colocalized in the cytoplasm in lung cancer cells (Supporting Figure 2D). Subsequent Co-IP analysis demonstrated that LKB1 can bind with RB. Moreover, CDK4, a known interactor of RB^[2, 3], was also pulled down by LKB1, although to a lesser extent than that by RB, further corroborating the LKB1–RB-CDK4 interaction (Supporting Figure 2E).

It was reported that CDK4/Cyclin D specifically docked in the C-terminal helix motif of RB, which is a major driver of RB phosphorylation^[2, 3]. Bioinformatic analysis revealed a statistically significant ($p < 0.05$) likelihood of LKB1 docking on the same site of RB as CDK4, suggesting potential competition for binding (Supporting Figure 2F). Since CDK4 was responsible for phosphorylating RB and LKB1 and CDK4 potentially share the same binding site on RB, we hypothesized that LKB1 competed with CDK4 for RB binding, thus leading to a decrease in RB phosphorylation. Subsequent IP experiments supported this hypothesis; specifically, LKB1 overexpression interfered with the interaction of CDK4 with RB. However, CDK4-RB interaction recovered in LKB1 kinase dead mutation group (Supporting Figure 1F). Inspired by this finding, we further constructed a plasmid to generate an RB protein variant with truncation of the RB C terminus (RB Δ C-term) to cover the possible binding site comprehensively. We utilized this RB variant (Δ C-term) to

investigate whether LKB1 competitively docks on the C domain of RB with CDK4. Immunoprecipitation assay of LKB1 and CDK4 with RB variant were both decreased (Supporting Figure 2G). In conclusion, our analysis indicates that LKB1 may be able to prevent the phosphorylation of RB by CDK4 by competitively occupying the binding site of CDK4. Either loss of LKB1 or LKB1 kinase-dead mutation reduced LKB1-RB interaction and increased RB phosphorylation.

Then we investigated whether the LKB1 downstream target AMPK was involved in regulating *ICAM1* transcription. LKB1 kinase phosphorylates AMPK^[4, 5], therefore we adopted the AMPK inhibitors in the LKB1 restored groups. It was observed that inhibition of AMPK phosphorylation did not affect *ICAM1* mRNA levels. Moreover, activation of AMPK using AICAR did not influence *ICAM1* mRNA expression either (Supporting Figure 2H, newly-added data).

Supporting figure 2. (A) Analysis of the protein level of total RB and pRB(S807/S811) in patients with or without *LKB1* mutation using TCGA-LUAD reverse-phase protein arrays (RPPA) dataset. (B) Immunoblot analysis of the indicated proteins performed in A549 or H460 cells expressing the indicated genes in plasmids. (C) Immunoprecipitation (IP) analysis in H1299 cells with ectopic expression of flag-LKB1 or transduced with parental flag-NC plasmids. The result was visualized by Coomassie blue staining and further analyzed by Nano LC-ESI-MS/MS Analysis. RB was detected as the selected MS/MS spectrum shown. (D) Multiple IF labeling using LKB1 and RB antibodies in H1299 cells. (E) Immunoprecipitation analysis of the interaction among LKB1, RB, and CDK4 performed in H1299 cells expressing intact LKB1. (F) Bioinformatics tool (Pepsite) was utilized to predict the possibility of LKB1 docking on the C-terminal helix motif of RB (SKFQQKLAEMTSTRTRMQKQK). (G) RB protein variant with truncation of the RB C terminus (Rb Δ C-term) was constructed (RB-Del). RB-WT and RB-Del immunoprecipitated with LKB1 and CDK4, respectively. (H) Quantitative RT-qPCR of relative *ICAM1* expression in A549 cells treated with the indicated drugs (dorsomorphin, AMPK inhibitor; AICAR, AMPK activator). (Results are presented as mean \pm SEM. Student's t-test was performed for comparisons. n.s., not significant; * p <.05)

3) Figure 5a, it will be important for the authors to show if other validated CDK4/6 inhibitors can also cause similar phenotypes, to avoid possible off-target effects of Palbociclib.

Response: Thank you so much for your valuable and professional suggestion. We repeated the experiments in Figure 5a by using another CDK4/6 inhibitor, ribociclib. We found that ribociclib increased expression of *ICAM1* remarkably in both A549 and H460

cells, which precludes possible off-target effects of Palbociclib (Supporting Figure 3A, newly-added data).

Corresponding changes and explanations we have added in the revised manuscript in the results part were as follows:

Page 18: To rule out potential off-target effects of palbociclib, we used another validated CDK4/6 inhibitor, ribociclib, to repeat the experiments. Results showed that ribociclib could effectively increase the expression of ICAM1 in both A549 and H460 cells (Supplementary Fig. 15A), corroborating the efficacy of CDK4/6 inhibitor.

Supporting figure 3. Quantitative RT-qPCR of relative *ICAM1* expression in A549 and H460 cells treated with ribociclib (10nM).

4). Figure 5g: the effects of upregulation in ICAM1 abundance by Palbociclib is moderate and not convincing.

Response: Thank you so much for your kind and pertinent suggestions. We repeated the experiment in Figure 5g, to demonstrate the effect of palbociclib on the upregulation of ICAM1 (Supporting Figure 4). We have substituted the images in Figure 5g of the revised manuscript.

Supporting figure 4. A549 cells transfected with lentivirus expressing the indicated genes (Ctrl, LKB1-WT, LKB1-Mut), treated \pm 500 nM palbociclib, transfected \pm siRB were harvested for immunoblot.

5). Figure 5i: authors should provide additional insights into the molecular mechanism of how LKB1 can regulate pRB. It is known that Rb phosphorylation status affects its ability to interact with its interactors. It will be nice for the authors to examine i) if phosphor-deficient Rb (13A mutant) or phosphor-mimetic (13E mutant) Rb display differences in interacting with LKB1, and ii). If Cdk4/6 inhibitor can affect LKB1 binding with Rb and iii) If LKB1 or its downstream kinase can phosphorylate Rb?

Response: Thank you so much for your professional and pertinent questions which inspire us a lot. It is interesting to investigate into whether RB phosphorylation status affects its ability to interact with LKB1. We therefore constructed phosphor-mimetic RB (13E mutant). Further IP analysis demonstrated that phosphor-mimetic Rb (13E mutant) display no significant differences in interacting with LKB1 (Supporting Figure 5A, newly-added data).

As CDK4/6 inhibitors could block CDK4, we treated cells with palbociclib to determine whether the binding between LKB1 and RB increased. Palbociclib unambiguously enhanced such binding (Supporting Figure 5B, newly-added data).

And as we have demonstrated in supporting figure 4, LKB1 could decreased phosphorylated RB. AMPK, downstream kinase of LKB1, does not affect RB phosphorylation.

Corresponding changes and explanations we have added in the revised manuscript in the results part were as follows:

Page 20: We constructed phosphor-mimetic RB (13E mutant). Co-IP analysis demonstrated that phosphor-mimetic RB display no significant differences in interacting with LKB1 (Supplementary Fig. 15M).

Supporting figure 5. (A) Wildtype RB or phosphor-mimetic RB (13E mutant) were introduced into A549 cells expressing HA-LKB1. Cells were harvested for immunoprecipitation analysis of the indicated proteins. (B) A549 cells expressing HA-LKB1 were treated with or without 500nM palbociclib, followed by immunoprecipitation analysis of the indicated proteins.

6). Supplemental Fig S16M, it will be critical to provide solid evidence to validate p65 as an upstream transcriptional activator for ICAM1. I) the authors should show that depleting p65 can reduce ICAM1 transcription, and ii) using ChIP assay to show p65 localize in the promoter region of ICAM1 and iii) ICAM1 possess p65 binding motifs in its promoter region.

Response: Thank you so much for your valuable and professional suggestion. Our RT-qPCR and flow cytometry results confirmed that overexpression of LKB1 remarkably increased the mRNA and protein expression of ICAM1, and knocking down p65 in LKB1 restored cells appreciably reduced ICAM1 expression (Supporting Figure 6A-B). Further ChIP-Seq analysis identified *ICAM1* gene as the target of p65 (Supporting Figure 6C, newly-added data). Analysis of the *ICAM1* promoter displayed a predicted p65 binding site (<http://jaspar.genereg.net/>) (Supporting Figure 6D), which was verified by chromatin immunoprecipitation (ChIP) analysis, demonstrating that overexpressed p65 bound to the *ICAM1* promoter in HEK-293T cells (Supporting Figure 6E, newly-added data). To investigate whether the binding sequences within the promoters of ICAM1 genes control transcription, we introduced the binding sites into a luciferase reporter system. The activity of the promoter-driven luciferase was enhanced by p65 overexpression. Moreover, the mutation of the p65 binding site in the ICAM1 promoter significantly reduced the p65-driven expression of luciferase (Supporting Figure 6F, newly-added data). Finally, Chromatin Immunoprecipitation (ChIP) analysis demonstrated that p65 occupancy in the promoter of ICAM1 was increased in LKB1-WT group compared to LKB1-loss group (Supporting Figure 1E). Taken together, p65 localizes in the promoter region of ICAM1 and ICAM1 possesses

p65 binding motifs in its promoter region. LKB1 regulates the transcriptional activity of p65 for the expression of ICAM1.

Corresponding changes and explanations we have added in the revised manuscript in the results part were as follows:

Page 21: Chromatin Immunoprecipitation (ChIP) analysis demonstrated that p65 occupies in the promoter of ICAM1 (Supplementary Fig. 16G).

To ascertain whether the binding sequences in ICAM1 gene promoters regulate transcription, we incorporated the binding sites into a luciferase reporter system. The results indicated that the activity of the promoter-driven luciferase was increased when p65 was overexpressed. Moreover, the mutation of the p65 binding site in the ICAM1 promoter significantly reduced the p65-driven expression of luciferase (Supplementary Fig. 16H).

We treated cells with palbociclib to determine whether blocking CDK4 interfere the binding between LKB1 and RB. Results showed that palbociclib unambiguously enhanced such binding (Supplementary Fig. 16L).

Supporting figure 6. (A-B) RT-qPCR (A) and flow cytometry (B) analysis of ICAM1 expression. (C) ChIP-Seq profiles for p65 (in different cell lines) at ICAM1 promoter loci. (D) The genomic loci for ICAM1 and the putative p65 binding sequence in the promoter of ICAM1. (E) ChIP analysis of p65 binding to ICAM1 promoter in 293T cells over-expressing p65. (F) Luciferase reporter assays were conducted in the

indicated cells expressing wild-type or mutated p65 binding site of ICAM1. (Results are presented as mean \pm SEM. Student's t-test or one-way ANOVA was performed for comparisons. n.s., not significant; **p<.01, **p<0.001, ****p<.0001)

References

1. Gurumurthy S, Hezel AF, Sahin E, Berger JH, Bosenberg MW, Bardeesy N. LKB1 deficiency sensitizes mice to carcinogen-induced tumorigenesis. **Cancer Res** **2008**, 68(1): 55-63.
2. Topacio BR, Zatulovskiy E, Cristea S, Xie S, Tambo CS, Rubin SM, *et al.* Cyclin D-Cdk4,6 Drives Cell-Cycle Progression via the Retinoblastoma Protein's C-Terminal Helix. **Mol Cell** **2019**, 74(4): 758-770 e754.
3. Wallace M, Ball KL. Docking-dependent regulation of the Rb tumor suppressor protein by Cdk4. **Mol Cell Biol** **2004**, 24(12): 5606-5619.
4. Jansen M, Ten Klooster JP, Offerhaus GJ, Clevers H. LKB1 and AMPK family signaling: the intimate link between cell polarity and energy metabolism. **Physiol Rev** **2009**, 89(3): 777-798.
5. Shackelford DB, Shaw RJ. The LKB1-AMPK pathway: metabolism and growth control in tumour suppression. **Nat Rev Cancer** **2009**, 9(8): 563-575.

REVIEWERS' COMMENTS

Reviewer #5 (Remarks to the Author):

The authors have addressed most of the raised concerns.

Reviewer #5 (Remarks to the Author):

The authors have addressed most of the raised concerns.

Response: Thank you for your comment on our manuscript, which encourages us a lot.
We appreciate your contribution to the advancement of our research.